# On the Universality and Complexity of GNN Solving Second-order Cone Programs

**Ruizhe Li**[1], **Enming Liang**[2,*], **Minghua Chen**[2,3,*]

[1]Department of Mathematics, Southern University of Science and Technology
[2]Department of Data Science, City University of Hong Kong
[3]School of Data Science, The Chinese University of Hong Kong, Shenzhen

## Abstract

Graph Neural Networks (GNNs) have demonstrated both empirical efficiency and universal expressivity for solving constrained optimization problems such as linear and quadratic programming. However, extending this paradigm to more general convex problems with universality guarantees, particularly Second-Order Cone Programs (SOCPs), remains largely unexplored. We address this challenge by proposing a novel graph representation that captures the inherent structure of conic constraints. We then establish a key universality theorem: *there exist GNNs that can provably approximate essential SOCP properties, such as instance feasibility and optimal solutions*. We further derive the sample complexity for GNN generalization based on Rademacher complexity, filling an important gap for Weisfeiler-Lehman-based GNNs in learning-to-optimize paradigms. Our results provide a rigorous foundation linking GNN expressivity and generalization power to conic optimization structure, opening new avenues for scalable, data-driven SOCP solvers. The approach extends naturally to $p$-order cone programming for any $p \geq 1$ while preserving universal expressivity and requiring no structural modifications to the GNN architecture. Numerical experiments on randomly generated SOCPs and real-world power grid problems demonstrate the effectiveness of our approach, achieving superior prediction accuracy with significantly fewer parameters than fully connected neural networks.

## 1 Introduction:

Second Order Cone Programming (SOCP) represents a fundamental class of convex optimization problems with numerous real-world applications (Lobo et al., 1998), including optimal power flow (Gan et al., 2014), trajectory planning (Liu et al., 2016), image restoration (Goldfarb & Yin, 2005), signal processing (Shi et al., 2014), and network localization (Tseng, 2007). However, traditional algorithms, such as primal-dual interior point methods, face computational limitations in large-scale applications, particularly in real-time scenarios where rapid response is crucial.

Recent advances in machine learning, such as the learn-to-optimize (L2O) paradigm (Chen et al., 2022a; Li & Malik, 2016), have enabled solving optimization problems in real-time. Specifically, graph neural networks (GNNs) have been proven efficient in training by leveraging the inherent graph structures of the problem. For instance, linear programs (LP) can be modeled as bipartite graphs with variable and constraint nodes (Chen et al., 2022b), enabling efficient learning with a parameter sharing mechanism over GPUs. Beyond empirical success,

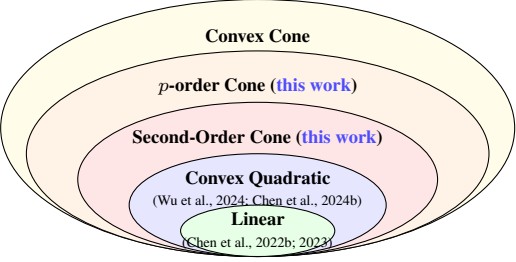

Figure 1: GNN expressivity for convex programs.

theoretical foundations, including *universal approximation capabilities*, have been established for GNN applications in (mixed-integer) LP (Chen et al., 2022b; 2023), quadratic programming (QP) (Chen et al., 2024b), and convex quadratically constrained QP (Chen et al., 2024b; Wu et al., 2024).

---
*Corresponding authors: Enming Liang (enming.cityu@gmail.com); Minghua Chen (minghua@cuhk.edu.cn)

Despite these advances, extending GNNs to more general convex programs like SOCP remains an open challenge. A key difficulty lies in the hybrid structure of second-order cone constraints, which involve both linear parts and non-linear norms. Effectively modeling the interplay between them and encoding constraints into graphs remains largely open. This paper proposes a novel GNN architecture with universal approximation capabilities for SOCPs, making the following contributions:

▷ We propose a novel graph representation for SOCPs, which exploits linear relationships within the non-linear conic constraint and decomposes it into separated nodes for efficient graph representations.

▷ Based on proposed graph representations, we design SOCP-GNNs, to predict the key properties of SOCPs, including instance feasibility and optimal solutions, with universal expressivity guarantees. Our GNN design and expressivity guarantees can be extended to $p$-order conic programming for $p \geq 1$ without GNN structural modifications.

▷ We further derive the sample complexity of the SOCP-GNNs for generalization. Such analysis is general and can also be extended to other Weisfeiler-Lehman-based GNN approaches in the L2O community (Chen et al., 2022b; 2023; Wu et al., 2024).

▷ Our experiments demonstrate that the expressivity of designed GNNs, which use fewer parameters to achieve better prediction accuracy compared to fully connected NNs, in both the synthetic SOCP dataset and the real-world power grid optimization.

To the best of our knowledge, this is the first GNN design for SOCP with universal expressivity guarantees, and also the first work to analyze the generalization ability of Weisfeiler-Lehman-based GNNs designed in the L2O paradigm.

## 2 RELATED WORK

**GNN Expressivity in L2O Paradigms:** We review two primary paradigms for analyzing GNN expressivity for optimization problems: the *Weisfeiler-Lehman* (WL)-based and *Algorithm-Unrolling* (AU)-based frameworks.

The WL-based framework models optimization problems as graphs, where nodes represent variables and constraints, with edges modeling their interactions. It then links the GNN's expressive power with WL tests on graphs. Building on established foundations for (mixed-integer) linear programs (LP) (Chen et al., 2022b; 2023), researchers have extended this framework to more complex problems such as quadratic programs (QP) (Chen et al., 2024b) and quadratically constrained QP (QCQP) (Wu et al., 2024). A key challenge is representing non-linear constraints, as encoding complex interactions into nodes and edges is non-trivial. Recent work has addressed convex quadratic constraints through dynamic edge updates (Chen et al., 2024b) or augmented quadratic variable nodes (Wu et al., 2024). However, extending existing frameworks to represent general conic constraints like second-order cones remains an open question (see Appendix A.1.1 for details).

The AU-based paradigm maps iterative steps of specialized algorithms (e.g., primal-dual methods) onto GNN layers. By aligning GNN layers with known algorithms for specific problems, such as LP (Qian et al., 2024; Li et al., 2024a;b; 2025a), QP (Qian & Morris, 2025a; Yang et al., 2024a), and combinatorial problems (Yau et al., 2025; He & Vitercik, 2025), universality and parameter complexity can be naturally established through existing algorithmic convergence properties. However, representing more complex algorithmic steps involving non-linear operations (e.g., factorization or projection) is non-trivial. Furthermore, the GNN's expressivity is inherently limited by the underlying capability of the algorithm itself (see Appendix A.1.2 for details).

**Generalization of GNNs and L2O:** We briefly review several studies on the generalization ability of both *GNNs* and *L2O* paradigms (see Appendix A.2 for details).

To study the generalization capability of GNN and its variants, researchers have leveraged multiple ways, such as Vapnik–Chervonenkis(VC) dimension (Scarselli et al., 2018; Morris et al., 2023; Franks et al., 2024; D'Inverno et al., 2025), Rademacher complexity (Garg et al., 2020; Pellizzoni et al., 2024), PAC-Bayes bound (Ju et al., 2023; Liao et al., 2020), and stochastic optimization (Tang & Liu, 2023). However, these works cannot be directly applied to WL-based GNN frameworks under the L2O paradigm due to the continuous feature space of optimization problems and the difference in GNN structures. The generalization performance of L2O or data-driven methods has also been studied from many perspectives, including VC dimension (and pseudo dimension) (Balcan et al.,

2021), loss landscape (Yang et al., 2023), and PAC-Bayes bound (Sucker & Ochs, 2025; Sambharya & Stellato, 2024). However, these works are not specifically designed for WL-based GNNs in L2O paradigms.

In this work, we extend the WL-based framework to optimization problems with second-order cone constraints, a general class that encompasses LP, QP, and convex QCQP, with extensive real-world applications. Our specialized GNN achieves universal expressivity capabilities while maintaining computational efficiency, establishing a foundational approach for extending GNNs to broader conic programming domains. Additionally, we provide the first generalization analysis for WL-based GNNs in the learning-to-optimize paradigm, establishing theoretical foundations for sample complexity when applying GNNs to solve optimization problems.

## 3  PROBLEM DEFINITION AND OPEN ISSUES

We consider a general second-order cone programming (SOCP) (Alizadeh & Goldfarb, 2003) as:

$$\min_{l \leq x \leq r} \quad e^\top x \quad \text{s.t.} \quad Fx \leq g, \quad \|A_i x + b_i\|_2 \leq c_i^T x + d_i, \ i \in [m] \tag{1}$$

where decision variables are $x \in \mathbb{R}^n$ and the problem parameters are $e \in \mathbb{R}^n$, $A_i \in \mathbb{R}^{k_i \times n}$, $b_i \in \mathbb{R}^{k_i}$, $c_i \in \mathbb{R}^n$, $d_i \in \mathbb{R}$, $F \in \mathbb{R}^{b \times n}$, $g \in \mathbb{R}^b$, $l \in \mathbb{R}^n$, and $r \in \mathbb{R}^n$.

**Open issue**: While GNNs have successfully modeled linear and convex quadratic constraints with expressivity guarantees, handling more general second-order cone (SOC) constraints remains challenging. Additionally, the generalization capacity of GNNs for optimization problems remains largely unexplored. While previous work focused on expressivity, understanding how many training samples are needed for good performance over new instances is critical for trustworthy applications.

## 4  METHODOLOGY

We design the following layered graph representation to address the expressivity of GNN for SOCP:

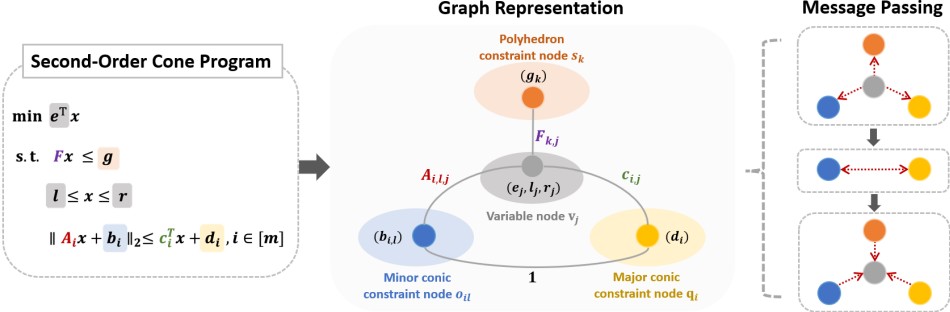

Figure 2: The graph representation of SOCPs and the message passing steps in GNN design. A specific SOCP instance and its corresponding SOCP-graph are included in Fig. 6, Appendix B.4.

### 4.1  GRAPH REPRESENTATION OF SOCPS

As shown in Fig. 2, the graph representation of an SOCP consists of four types of nodes, to represent decision variables ($V_1$), polyhedron constraints ($V_2$), minor conic constraints ($V_3$), and major conic constraints ($V_4$):

- $V_1 := \{v_j\}_{j \in [n]}$ denotes decision variables, where each node $v_j$ is associated with a feature tuple $(e_j, l_j, r_j)$, representing the objective coefficient, variable lower and upper bounds.
- $V_2 := \{s_k\}_{k \in [b]}$ denotes polyhedron constraints equipped with feature $(g_k)$ for each node.
- $V_3 := \{o_{il}\}_{i \in [m]}^{l \in [k_i]}$ denotes the minor conic constraint, where each node $o_{il}$ represents the $i$-th conic constraint's $l$-th component, with feature $(b_{i,l})$.
- $V_4 := \{q_i\}_{i \in [m]}$ denotes the $i$-th conic constraint with feature $(d_i)$.

Meanwhile, the SOCP graph includes four types of edges to model the interactions between the decision variables and different constraint nodes:

- $e_{jk} \in V_1 \times V_2$ denote the edges between the variable node $v_j$ and the polyhedron constraint node $s_k$, with weight $F_{kj}$.

- $e_{j,il} \in V_1 \times V_3$ denote the edge between variable node $v_j$ and minor conic constraint node $o_{il}$, with weight $A_{i,lj}$.

- $e_{ji} \in V_1 \times V_4$ denote the edge between the variable node $v_j$ and major conic constraint node $q_i$, with weight $c_{i,j}$.

- $e_{il,i} \in V_3 \times V_4$ denote the edge between node $o_{il}$ and node $q_i$, with a constant weight 1.

*Remark* 1 (**Insights of Graph Design**). For linear objectives and polyhedral constraints, our structure builds upon foundational works (Chen et al., 2022b). To deal with nonlinear second-order cone constraints, we exploit the linear relationships within the conic constraint, specifically, between $A_i$ and $x$, and between $c_i$ and $x$. By representing the left-hand side and right-hand side as separate constraint nodes ($V_3$ and $V_4$), with linear interactions to decision nodes separately, and connecting $V_3$ and $V_4$ via additional edges, we decompose the challenging nonlinear conic constraint into components amenable to efficient graph representations. Such decomposition and representation are not limited to the second-order cone, and we provide more discussion after Theorem 2

*Remark* 2 (**SOCP → QCQP**[1]). One may note that SOC constraints, $\|Ax + b\|_2 \leq c^\top x + d$, can be transformed into quadratic constraints by squaring both sides, potentially enabling the application of previous work on quadratic constraints (Wu et al., 2024; Chen et al., 2024b). However, this transformation introduces two significant challenges: (i) the resulting quadratic coefficient matrix $A^\top A - cc^\top$ may not be positive semidefinite, rendering previous work theoretically inapplicable for such a non-convex QC; and (ii) the quadratic coefficient matrix $A^\top A - cc^\top$ may be dense, losing the potential sparse/low-rank structure of $A$ and $c$ in the SOC constraint and making the graph representation and message passing inefficient.

*Remark* 3 (**Convex QCQP → SOCP**). Conversely, we may transform convex quadratic constraints of the form $x^\top Q x + c^\top x + d \leq 0$ into SOC constraints for more effective graph representation. For example, we can apply matrix decomposition $Q = LL^\top$ where $L \in \mathbb{R}^{n \times r}$, and reformulate the constraint as $\left\| [(1 + c^\top x + d)/2; \ L^T x] \right\|_2 \leq (1 - c^\top x - d)/2$. Such a transformation is particularly efficient for low-rank matrices $Q$ where $r \ll n$, as it reduces the complexity of the graph representation for original convex quadratic constraints, from quadratic node (Wu et al., 2024) to minor conic constraint node via SOC graph representation. The convex quadratic objective in QCQP can also be converted to a linear objective by adding the epigraph constraint (Alizadeh & Goldfarb, 2003). Thus, a convex QCQP with $n$ variables and $m$ quadratic constraints is equivalent to an SOCP with $n + 1$ variables and $m + 1$ conic constraints (potentially low-rank). We further provide a quantitative comparison in the next section (Table 3).

## 4.2 Message Passing in SOCP-GNNs

Given the established graph representation of SOCPs, we propose message-passing (MP)-GNNs, consisting of an embedding layer, $T$ message-passing layers (each comprised of three sub-layers), and a readout layer, detailed as follows:

- **Embedding Layer**: For all nodes, the input features $h^{0,v}, h^{0,s}, h^{0,o}, h^{0,q}$ are initialized by embedding the node features into a hidden space $\mathbb{R}^{h_0}$, where $h_0$ is the space dimension. Specifically,

$$h^{0,v} \leftarrow \hat{g}_1^0(h^v), \forall v \in V_1, \quad h^{0,s} \leftarrow \hat{g}_2^0(h^s), \forall s \in V_2$$
$$h^{0,o} \leftarrow \hat{g}_3^0(h^o), \forall o \in V_3, \quad h^{0,q} \leftarrow \hat{g}_4^0(h^q), \forall q \in V_4$$

  where $\hat{g}_l^0$ are learnable embedding functions for $l = 1, 2, 3, 4$, and $h^v, h^s, h^o, h^q$ denotes the node features for $v \in V_1, s \in V_2, o \in V_3, q \in V_4$, respectively.

- **Message-Passing Layer:** As shown in Fig. 2, each message-passing layer consists of three sub-layers for updating the features of nodes with learnable functions $f_l^t, g_l^t$. For notation simplicity, $w_{ij}$ represents the weight of edge $e_{ij}$ and $\tau(n) \in \{1, 2, 3, 4\}$ denotes the index of the node set for a node $n$.

---

[1]Please refer to Appendix B.2 for detailed equivalent SOCP formulations.

- **Updating for all constraint nodes** ($V_1 \to V_2 + V_3 + V_4$): $\forall s \in V_2$ and $\forall n \in V_3 \cup V_4$, we update the embedding as:

$$
h^{t+1,s} \leftarrow g_1^t \left( h^{t,s}, \sum_{v \in V_1} w_{v,s} f_1^t(h^{t,v}) \right), \; \bar{h}^{t,n} \leftarrow g_{\tau(n)-1}^t \left( h^{t,n}, \sum_{v \in V_1} w_{v,n} f_{\tau(n)-1}^t(h^{t,v}) \right)
$$

- **Updating between major and minor conic constraint nodes** ($V_3 \to V_4$ and $V_4 \to V_3$): $\forall q \in V_4$ and $\forall o \in V_3$, we update the embedding as:

$$
h^{t+1,q} \leftarrow g_4^t \left( \bar{h}^{t,q}, \sum_{o \in V_3} w_{o,q} f_4^t(\bar{h}^{t,o}) \right), \; h^{t+1,o} \leftarrow g_5^t \left( \bar{h}^{t,o}, \sum_{q \in V_4} w_{q,o} f_5^t(h^{t+1,q}) \right)
$$

- **Updating for variable nodes** ($V_2 + V_3 + V_4 \to V_1$): $\forall v \in V_1$, we update the embedding as:

$$
h^{t+1,v} \leftarrow g_6^t \left( h^{t,v}, \sum_{s \in V_2} w_{s,v} f_6^t(h^{t+1,s}), \sum_{o \in V_3} w_{o,v} f_7^t(h^{t+1,o}), \sum_{q \in V_4} w_{q,v} f_8^t(h^{t+1,q}) \right)
$$

- **Readout layer**: The readout layer leverages a learnable function $f_{\text{out}}$ to map the node embedding $h^{T,v}$ output by the $T$-th (i.e., last) message-passing layer for $v \in V_1 \cup V_2 \cup V_3 \cup V_4$, to a readout $y$ in a desired output space $\mathbb{R}^a$, where $a$ is the output dimension. For example:

  - Graph-level scalar output (e.g., predicting SOCP feasibility with $a = 1$):

  $$
  y = f_{\text{out}}(I_1, I_2, I_3, I_4)
  $$

  - Node-level vector output (e.g., predicting SOCP optimal solutions with $a = n$):

  $$
  y_i = f_{\text{out}}\left( h^{T,v_i}, I_1, I_2, I_3, I_4 \right)
  $$

  where $I_1 = \sum_{v \in V_1} h^{T,v}$, $I_2 = \sum_{s \in V_2} h^{T,s}$, $I_3 = \sum_{o \in V_3} h^{T,o}$, $I_4 = \sum_{q \in V_4} h^{T,q}$.

As mentioned in Remarks 2 and 3, our SOCP-GNN also efficiently handles convex QCQPs by reformulating them into SOCP. Based on the GNN architecture described above, we analyze both the node and message passing complexity compared to previous works on convex QCQP (Wu et al., 2024; Chen et al., 2024b). Our SOCP-GNN achieves the same order of node and message passing complexity as state-of-the-art GNNs designed specifically

|  | Num. of Nodes | MP Complexity |
|---|---|---|
| (Wu et al., 2024) | $\mathcal{O}(n^2 + m)$ | $\mathcal{O}(n^3 + mn^2)$ |
| (Chen et al., 2024b) | $\mathcal{O}(mn)$ | $\mathcal{O}(mn^2)$ |
| Ours | $\mathcal{O}(n + \sum_{i=0}^m r_i)$ | $\mathcal{O}(n \cdot \sum_{i=0}^m r_i)$ |

Figure 3: Complexity comparison of GNNs for **convex QCQP** with $n$ variables, $m$ quadratic constraints, and quadratic coefficient matrix of ranks $r_i \leq n$, $i = 0, \ldots, m$, where $i = 0$ indicate the quadratic matrix from objective.

for QCQP with general parameter coefficients. In practice, the coefficient matrices may exhibit sparse or low-rank structure, resulting in different empirical performance: (i) When the quadratic coefficients $Q_i$ are sparse, previous GNNs benefit from reduced connections and message passing. After reformulating to SOCP via decomposition $Q_i = L_i L_i^T$, the resulting graph may lose this sparsity. However, for structured sparse matrices (e.g., banded or block diagonal (Davis, 2006; Golub & Van Loan, 2013)), sparsity is preserved in the SOCP-graph, and our SOCP-GNN inherits the computational benefits. (ii) When the quadratic matrices exhibit low-rank structure, SOCP-GNN becomes more efficient with reduced graph size and message passing complexity.

Therefore, *SOCP-GNN not only extends theoretical applicability to the broader class of SOCP beyond convex QCQPs, but also maintains competitive computational complexity when restricted to the convex QCQP subclass*. See detailed discussion in Appendix B.5.

## 5 UNIVERSALITY OF SOCP-GNN

With the established graph representation and corresponding GNN, we formally prove the universality of the GNN for predicting key properties of SOCPs, like the instance feasibility and optimal solutions.

## 5.1 BASIC DEFINITIONS

**Definition 5.1** (Spaces of SOCP-Graphs). Let $\mathcal{G}_{\text{SOCP}}^{n,m,k_1,...,k_m,b}$ denote the set of graph representations for all SOCPs with $n$ variables, $m$ conic constraints with dimension $k_1, ..., k_m$, and $b$ polyhedron constraints.

**Definition 5.2** (Spaces of SOCP-GNNs). Let $\mathcal{F}_{\text{SOCP}}^{n,m,k_1,...,k_m,b}(\mathbb{R}^a)$ be the set of message passing GNNs proposed in Sec. 4.2 that map the input graph in $\mathcal{G}_{\text{SOCP}}^{n,m,k_1,...,k_m,b}$ to a target output in $\mathbb{R}^a$. Each GNN is parameterized by continuous embedding functions $g_{l_1}^0, l_1 \in [4]$, continuous hidden functions in the message passing layers $g_{l_2}^t, l_2 \in [6]$ and $f_{l_3}^t, l_3 \in [8]$, and the continuous readout function $f_{\text{out}}$.

**Definition 5.3** (Target mappings[2]). Let $\mathcal{G}_{\text{SOCP}}$ be a graph representation of a SOCP problem. We define the following target mappings.

- **Feasibility mapping:** $\Phi_{\text{feas}}(\mathcal{G}_{\text{SOCP}}) = 1$ if the SOCP is feasible and $\Phi_{\text{feas}}(\mathcal{G}_{\text{SOCP}}) = 0$ otherwise.

- **Optimal solution mapping:** $\Phi_{\text{sol}}(\mathcal{G}_{\text{SOCP}}) = x^*$, where $x^*$ is the optimal solution of the SOCP[3].

## 5.2 SEPARATION POWER OF THE SOCP WEISFEILER–LEHMAN TEST

To investigate the relationship between target properties and SOCP-GNNs, we first analyze the separation power of SOCP-GNNs. The separation power of traditional GNNs is closely related to the Weisfeiler-Lehman (WL) test (Weisfeiler & Leman, 1968), a classical algorithm to identify whether two given graphs are isomorphic. To apply the WL test on SOCP-graphs, we design a modified WL test, called the SOCP-WL test, in Algorithm 1. Below, we provide the main theoretical result about the separation power of the SOCP-WL test.

**Theorem 1.** *Let $\mathcal{I}, \hat{\mathcal{I}}$ (with given sizes $n, m, k_1, ..., k_m, b$, encoded by $G, \hat{G} \in \mathcal{G}_{SOCP}^{n,m,k_1,...,k_m,b}$) be two SOCP instances. If the $G$ and $\hat{G}$ cannot be distinguished by the SOCP-WL test, then: For any target mapping $\Phi : \mathcal{G}_{SOCP}^{n,m,k_1,...,k_m,b} \to \mathbb{R}^a, \Phi(G) = \Phi(\hat{G})$ always holds up to permutations.*

The detailed proof can be found in Appendix C. By Theorem 1, we can see that: *any two instances which the SOCP-WL test cannot separate share the same target property we want (up to permutations).* Hence, demonstrating SOCP-GNN is equivalent to SOCP-WL guarantees its sufficient separation power, as shown in Appendix C.

## 5.3 UNIVERSAL APPROXIMATION OF SOCP-GNNS

Beyond separation power, expressive power (i.e., approximation capability) is also critical. Here, we provide the main theoretical results to validate the SOCP-GNN's universal expressivity for SOCP, i.e., *there always exists an SOCP-GNN that can universally approximate target mappings in Def. 5.3 within given error tolerance*:

**Theorem 2.** *For any Borel regular probability measure $\mathbb{P}$ on the space of SOCPs $\mathcal{G}_{SOCP}^{n,m,k_1,...,k_m,b}$, any target mapping $\Phi : \mathcal{G}_{SOCP}^{n,m,k_1,...,k_m,b} \to \mathbb{R}^a$ defined in Def. 5.3, and any $\delta, \epsilon > 0$, there exists $F \in \mathcal{F}_{SOCP}^{n,m,k_1,...,k_m,b}(\mathbb{R}^a)$ such that:*

$$\mathbb{P}\{||F(\mathcal{G}_{SOCP}) - \Phi(\mathcal{G}_{SOCP})|| > \delta\} < \epsilon. \tag{2}$$

The detailed proof is provided in Appendix C. This Theorem formally establishes the universal expressivity of the proposed SOCP-GNNs. The high-level proof structure follows established foundations for LP in (Chen et al., 2022b). However, previous graph design and expressivity proof can not directly be extended to the challenging non-linear SOC constraints. To this end, we leverage the *equivariance*, *convexity*, and *separability*[4] of the $\ell_2$ norm in SOC, and then establish the expressive power of proposed SOCP-GNNs. We further extend the universal expressivity of the proposed GNN to $p$-order cone programming in Appendix C.6, since the core lemmas in our proof are also satisfied for the $\ell_p$ norm.

---

[2]For more target mappings, please refer to Def. B.1. Theorem 2 also holds for these target mappings.

[3]Since SOCP may admit multiple optimum, we choose the one with minimum $l_2$ norm (Chen et al., 2022b).

[4]Please refer to those definitions in Definition C.2 and C.3.

# 6 GENERALIZATION ABILITY OF SOCP-GNNS

Beyond expressivity, the generalization capability of GNNs, i.e., *how many samples are needed for training to achieve good performance on unseen instances*, is critical for real-world trustworthy applications. While previous foundational works have focused on expressivity (Chen et al., 2022b; 2024b), the generalization ability of GNNs designed for optimization remains largely unexplored. Addressing this gap, our work takes an initial step towards formally analyzing the generalization properties of these models.

We consider a subclass of SOCP problems denoted as $\mathcal{X} \subset \mathcal{G}_{\text{SOCP}}^{n,m,k_1,\ldots,k_m,b}$ with **bounded** parameters $(e, \{A_i\}_{i=1}^m, \{b_i\}_{i=1}^m, \{c_i\}_{i=1}^m, \{d_i\}_{i=1}^m, F, g, l, r)$, and denote $N$ as the problem size, i.e., the total dimension of all parameters. Without loss of generality, it is sufficient to consider the problem parameters lie in a ball $\mathcal{B}_{r_i} = \{x \,|\|x\|_2 \leq r_i\}$ with some positive radius $r_i$.

**Definition 6.1** (Lipschitz GNN). A SOCP-GNN $f \in \mathcal{F}_{\text{SOCP}}^{n,m,k_1,\ldots,k_m,b}(\mathbb{R}^a)$ is said to be Lipschitz with respect to the input domain $\mathcal{X}$ if and only if $\exists L > 0$ such that for each output component $f_i$, $|f_i(x) - f_i(x')| \leq L\|x - x'\|$ holds for all $x, x' \in \mathcal{X}$. And $L$ denotes the Lipschitz constant of $f$.

We also assume the GNN is **Lipschitz** (Def. 6.1) and denote all the SOCP-GNNs whose Lipschitz constant is no more than $L$ with respect to the input domain $\mathcal{B}_{r_i} \subseteq \mathbb{R}^N$ by $\mathcal{A}_{L,N}$. We remark that this Lipschitz assumption is widely adopted in related works on the sample complexity/generalization ability of graph neural networks (Pellizzoni et al., 2024; Garg et al., 2020; Tang & Liu, 2023; Huang et al., 2024a). The Lipschitz condition holds in general if both the input domain and the parameter space are bounded, while the GNNs are differentiable with respect to the inputs and parameters. We then present the main theorem for the generalization capability of our GNN:

**Theorem 3** (Generalization Bound for SOCP-GNNs). *Consider the hypothesis class $\mathcal{A}_{L,N}$ of SOCP-GNNs with outputs in $\mathcal{Y}$ and input SOCP instances $\mathcal{X}$ with parameters in $\mathcal{B}_{r_i}$. Let $D$ be the uniform distribution over $\mathcal{X}$. Assume the loss function $\ell : \mathcal{Y} \times \mathcal{Y} \to \mathbb{R}$ is bounded by $p$ and $q$-Lipschitz with respect to the first parameter when the second parameter is fixed. For a training set $S$ of $m$ samples drawn i.i.d. from $D$, let $h^* = \arg\min_{h \in \mathcal{A}_{L,N}} L_D(h)$[5] be the population risk minimizer and $\hat{h}_S = \arg\min_{h \in \mathcal{A}_{T,N}} \hat{L}_S(h)$ be the empirical risk minimizer. Then with probability at least $1 - \delta$:*

$$L_D(\hat{h}_S)) - L_D(h^*) \leq \mathcal{C}_{task} \cdot \mathcal{B}(m, N, L, r) + 2p\sqrt{2\log(1/\delta)/m}$$

*where the complexity term is $\mathcal{B}(m, N, L, r) = \inf_{\epsilon \in [0, r/2]} \left[ 4\epsilon + \frac{12}{\sqrt{m}} \int_\epsilon^{r/2} C(v)\, dv \right]$, with $C(v) = \sqrt{(\frac{4Lr_i + v}{v})^N (1 - (1 - \min((\frac{v}{2Lr_i})^N, 1))^m) \log\left(\frac{2r}{v} + 2\right)}$. The task-dependent constant are defined as: $\mathcal{C}_{task} = 4q$ for graph-level predictions with outputs in $[-r, r]$; and $\mathcal{C}_{task} = 4\sqrt{2}nq$ for node-level predictions with outputs in $[-r, r]^n$. Here, $n$ denotes the number of decision variables.*

The detailed proof can be found in Appendix D.3. The conditions in Theorem 3 are satisfied by many common loss functions, including margin loss and MSE loss under mild regularity conditions. Theorem 3 provides the first sample complexity analysis for WL-test based GNNs, particularly SOCP-GNNs, establishing a solid theoretical foundation for task-specific sample complexity research. As demonstrated in Theorem 3, sample complexity deteriorates as GNN complexity or problem dimension increases (i.e., as $L$ or $N$ grows larger). This relationship is directly evident from the proofs in Theorems 13 and 14.

*Remark* 4. We also analyze the VC dimension and pseudo dimension of SOCP-GNNs with scalar outputs in Appendices D.1 and D.2, respectively, for SOCPs whose parameters can be encoded into discrete labels (e.g., problems where all coefficients are binary-valued). However, these theoretical results reveal practical limitations: for continuous problem parameters typical in real-world SOCPs, the resulting VC and pseudo dimensions are often infinite. This necessitates more powerful analytical tools capable of handling continuous feature spaces, such as Theorem 3. Under the same assumptions as Theorem 3, our theoretical framework extends directly to other distributions over different SOCP problems and other WL-based approaches in L2O paradigms, since the application of Tonelli's theorem, Jensen's inequality, and contraction lemmas all remain valid as proven in Appendix D.3.

---

[5]Here, the population(true) risk is defined as $L_D(h) = \mathbb{E}_{x \sim D}(l(h(x), y(x)))$ and the empirical risk for training set $S$ is defined as $\frac{\sum_{x \in S} l(h(x), y(x))}{|S|}$, where $y(x)$ is the true label of SOCP instance $x$, e.g. objective value.

## 7 NUMERICAL EXPERIMENTS

In this section, we demonstrate the efficiency of the proposed SOCP-GNN on both synthetic SOCP instances and real-world power grid optimization problems.

To validate the empirical advantages of SOCP-Graphs (Sec. 4.1), we employ a fully-connected neural network (FCNN) as one baseline, where the FCNN receives the same problem parameters as input in vectorized form. This comparison isolates the benefits of the graph structure inherent in SOCP-GNN relative to a standard neural network approach. To validate the designed message-passing mechanisms (Sec. 4.2), we compare our SOCP-GNN with vanilla Message Passing Neural Network (MPNN) (Gilmer et al., 2017) and Graph Isomorphism Network (GIN) (Xu et al., 2019). We note that no existing graph representation has been specifically designed for SOCP problems with non-linear constraints. Therefore, we compare all of those GNN-based baselines to the same graph structure proposed in this work. Notably, vanilla MPNN and GIN perform message passing based on the adjacency relationships, whereas our approach incorporates well-designed constraint-to-variable and variable-to-constraint message passing. While SOCP instances can be reformulated as QCQP, previous QCQP-GNN (Wu et al., 2024; Chen et al., 2024b) do not provide publicly available implementations, and more importantly, they lack theoretical universal approximation guarantees for SOCP instances since the associated quadratic constraints are not necessarily positive semidefinite. Consequently, we focus our experimental comparison on vanilla MPNN, GIN and FCNN baselines.

### 7.1 SYNTHETIC SOCP INSTANCE

For dataset generation, we randomly sample coefficient matrices and constraint parameters following the CVXPY example code structure and parameter settings (Chen et al., 2024b). Each instance is solved in CVXPY to obtain ground truth solutions, forming our training dataset. We then train SOCP-GNN using regular supervised learning procedures for optimal solution predictions. We also test the feasibility classification in Appendix F.

As shown in Fig. 4(a), 4(b) and 4(c), we compare the solution relative error [6] of our SOCP-GNN against the FCNN, Vanilla MPNN, and GIN baselines across three different problem scales over 100 training epochs. SOCP-GNN demonstrates superior performance across all scales, achieving substantially lower error on both training and validation sets. For the large-scale 500-dim SOCP with input dimension `452,400`, our GNN achieves better prediction accuracy while using only `0.35Mb` parameters compared to `110Mb` for the FCNN baseline (shown in Fig. 5(c))—a $300\times$ reduction in model complexity. This demonstrates SOCP-GNN's parameter efficiency and its ability to effectively learn target mappings in SOCPs by leveraging the natural sparse graph structure of these problems. All graph-based neural networks outperform FCNN significantly on synthetic datasets, further validating the effectiveness of our graph representation. Notably, SOCP-GNN substantially surpasses other GNN baselines, demonstrating the advantage of SOCP-GNN's three-sublayer message passing mechanism over methods relying solely on adjacency relationships.

### 7.2 SOC-BASED OPTIMAL POWER FLOW

Optimal power flow (OPF) is the fundamental problem in power systems optimization, determining the most economical operating point while satisfying all constraints. The second-order cone (SoC) relaxation transforms the non-convex AC power flow equations into tractable convex conic forms (see formulations in Appendix E). This relaxation is exact for radial networks and provides near-optimal solutions for meshed transmission systems (Gan et al., 2014; Madani et al., 2014), making it preferred for real-time operations[7]

We evaluate SOCP-GNN on IEEE test systems ranging from 118 to 500-bus power grids (Babaeine-jadsarookolaee et al., 2019). For each grid, we generate problem instances by randomly varying load

---

[6]The solution relative error (Chen et al., 2024b) between prediction $\hat{x}$ and ground truth $x^*$ is as $\frac{\|\hat{x}-x^*\|_2^2}{\max(1,\|x^*\|_2^2)}$

[7]We note that GNN-based methods have been directly applied to non-convex AC-OPF problems using the physical graph structure of power grids (Yang et al., 2024b; Owerko et al., 2020; Varbella et al., 2024). However, we do not compare with these methods directly because: (1) the SOC relaxation provides a lower bound for the original AC-OPF problem, making direct performance comparison unfair, and (2) our focus is on demonstrating GNN universality for convex SOC-relaxed problems. Nevertheless, extending our theoretical framework to establish universality guarantees for non-convex AC-OPF remains an important direction for future work with significant practical value.

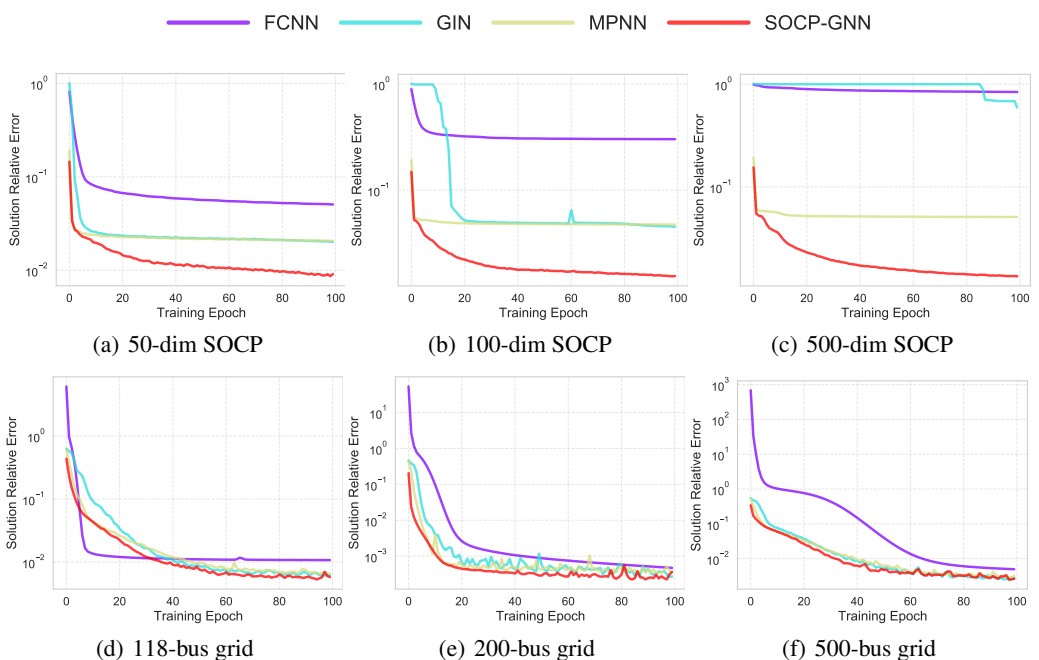

Figure 4: **(a)-(c)**: Performance comparison in predicting solutions of synthetic SOCP instances. The SOCP size are $(50,10,10)$, $(100,50,50)$, and $(500,100,100)$, respectively. **(d)-(f)**: Performance comparison in predicting solutions of SoC-OPF. The SOCP size are $(596,854,555)$, $(764,1288,732)$, and $(2182,3454,2181)$, respectively. Here, we denote the size of an SOCP instance by a tuple $(n, b, m)$, where $n$ represents the number of decision variables, $b$ denotes the number of polyhedral constraints, and $m$ indicates the number of second-order cone constraints. The total input parameters for an SOCP $(n, b, m)$ are of dimension $\mathcal{O}(n \cdot (b + m))$.

demands and generator costs, and introducing random line outages (i.e., loss of line connection) to simulate realistic operational scenarios. We compare against CVXPY with MOSEK solver (on CPU) and learning-based approaches, including various GNNs and FCNN (on both CPU and GPU for a fair comparison). Results in Fig.4(d),4(e), and 4(f) show that SOCP-GNN achieves lower errors across all problem scales with significantly fewer parameters than the FCNN baseline in real-world scenarios. For real-world OPF problems with sparse structures, SOCP-GNN performs better than on randomly generated instances in both prediction errors and inference time (shown in Fig. 5(a) and Fig. 5(b)), highlighting the potential for real-world applications. Consistent with the results obtained from the synthetic dataset, experiments on real-world OPF problems further demonstrate the effectiveness of our graph representation and the benefits of our three-layer message passing mechanism.

## 7.3 EMPIRICAL STUDY ON SAMPLE/MODEL COMPLEXITY AND SIZE GENERALIZATION

In this section, we investigate the performance of SOCP-GNNs under different model sizes and training samples. The detailed experiment settings can be found in Appendix F.4. As shown in Fig. 5(a)-5(d), SOCP-GNNs are both scalable and fast at solving SOCPs with superior accuracy. Since the learnable functions are applied feature-wise, independent of the number of nodes and edges, the memory cost of SOCP-GNNs remains constant across different problem sizes. We then analyze the sensitivity of SOCP-GNN to different hidden sizes and training samples as shown in Fig. 5(e). Both training and validation losses decrease as hidden layer size or number of training samples increases, demonstrating the model's capacity to benefit from additional parameters and data while validating Theorem 3.

To further validate the Lipschitz assumption in Theorem 3, we use projected optimization method (Gouk et al., 2020) to control the Lipschitz coefficient of SOCP-GNN. The results can be found in Appendix F.5. From the result, we can see that: the generalization gap decreases as the model becomes less complex (i.e. we decrease the Lipschitz constant $L$ of SOCP-GNN) as the train error

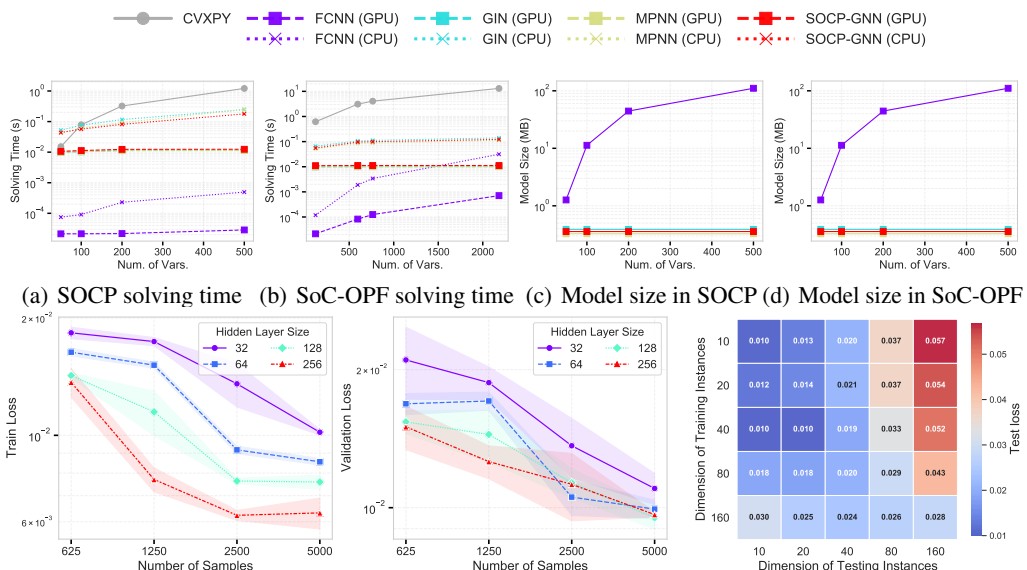

(a) SOCP solving time (b) SoC-OPF solving time (c) Model size in SOCP (d) Model size in SoC-OPF

(e) SOCP-GNN training loss (Left) and validation loss (Right) under different hidden-layer embedding sizes and number of training samples.

(f) Training and Testing of SOCP-GNN on different problem sizes.

Figure 5: **(a)-(d)**: inference time and model size comparison between GNN and FCNN in SOCP and SoC-OPF problems. **(e)**: sensitivity analysis of GNN on hidden-layer embedding sizes and number of training samples. **(f)**: generalization ability analysis of GNN on SOCP problems of different sizes.

increases. This enhances the tradeoff between the expressive power and generalization ability of SOCP-GNNs, as indicated in Theorem 3.

We also investigate the size generalization capability from small to large-scale problems, with results shown in Fig. 5(f). Models trained on larger training samples perform well on smaller testing instances, while those trained on small samples generalize less effectively to larger SOCP instances. This observation motivates further research to theoretically characterize the size generalization ability of GNN training (Huang et al., 2024b).

## 8 CONCLUSIONS, LIMITATIONS, AND FUTURE WORKS

This paper introduces a novel graph representation for SOCP, a fundamental class of convex optimization problems covering LP, QP, and convex QCQP. We design a novel GNN architecture that exploits inherent SOC structure for predicting key properties, including feasibility and optimal solutions, with established universal expressivity guarantees. Our framework extends to $p$-order cone programming, broadening GNN applicability in a subclass of conic and polynomial optimization. We establish the first general framework for analyzing the generalization capability of SOCP-GNN or other WL-based GNNs, bridging an important research gap. Comprehensive experiments validate both our theoretical predictions and practical performance.

While our work establishes universality and sample complexity guarantees, several important limitations suggest promising future directions. The parameter complexity of GNNs for optimization problems remains a significant challenge shared by prior WL-test-based frameworks. One promising avenue involves combining algorithm-unrolling approaches with the WL-based framework to develop a unified theoretical analysis of GNNs for optimization. Another important direction is extending the GNN paradigm beyond convex settings to handle semidefinite programs and general polynomial optimization problems. Such extensions would require developing new graph representations and theoretical frameworks capable of capturing the more complex variable-constraint relations. Furthermore, exploring metrics beyond the naive $\ell_2$ norm for the SOCP parameter space is crucial. An optimization-property-aware distance could significantly lower the covering number by better fitting the problem's intrinsic structure, directly yielding a tighter generalization bound.

## ACKNOWLEDGMENT

This work is supported in part by a General Research Fund from Research Grants Council, Hong Kong (Project No. 11200223), a Collaborative Research Fund from Research Grants Council, Hong Kong (Project No. C1049-24G), an InnoHK initiative, The Government of the HKSAR, Laboratory for AI-Powered Financial Technologies, a Shenzhen-Hong Kong-Macau Science & Technology Project (Category C, Project No. SGDX20220530111203026), and a Start-up Research Grant from The Chinese University of Hong Kong, Shenzhen (Project No. UDF01004086). The authors would also like to thank the anonymous reviewers for their helpful comments. This work has previously been presented at the NeurIPS 2025 ScaleOPT workshop (Li et al., 2025b).

## LLM USAGE

Large Language Models (LLMs) were used to aid in the writing and polishing of the manuscript.

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

# Contents

# A   DISCUSSION OF RELATED WORK

Machine learning has been increasingly integrated with optimization problems to enable efficient decision-making. Several paradigms have emerged for this integration, each addressing different aspects of the optimization process but sharing common architectural challenges. Learning to optimize trains models to iteratively refine solutions or to learn update rules that accelerate convergence (Li & Malik, 2016; Chen et al., 2022a). Decision-focused learning leverages downstream optimization tasks to refine upstream prediction models (Elmachtoub & Grigas, 2022; Mandi et al., 2024; Yang et al., 2025; Guo et al., 2026). End-to-end learning trains neural networks to directly predict solutions to constrained optimization problems, serving either as warm starts for solvers or as approximate optima (Owerko et al., 2020; Pan et al., 2020; 2022; Liang et al., 2023; 2024; Liang & Chen, 2025). Generative modeling approaches learn solution distributions for hard optimization problems (Sun & Yang, 2023; Liang & Chen, 2024; Li et al., 2023; 2025c; 2026).

While these paradigms differ in their formulations, they share a critical challenge: selecting an appropriate neural network architecture for the predictor or iterative component is essential for algorithmic performance.

## A.1   GNNs FOR CONSTRAINED OPTIMIZATION

Among various neural architectures, Graph Neural Networks (GNNs) have demonstrated particular effectiveness for problems with inherent graph structures, leveraging the natural correspondence between problem formulations and graph representations (Chen et al., 2022b; Wu et al., 2024; Chen et al., 2023; Varbella et al., 2024; Shamseldein, 2026; Liu et al., 2025).

To understand the fundamental capabilities of GNNs in optimization contexts, research has examined their expressivity (Chen et al., 2022b; 2024c), generalization properties (Balcan et al., 2021; 2024), and symmetry preservation (Chen et al., 2024a; Matthew Morris, 2024; Chen et al., 2025; Qian & Morris, 2025b). The analysis of GNN expressive power in optimization is primarily guided by two complementary theoretical paradigms: the **Weisfeiler-Lehman (WL)-test-based framework**, which characterizes what optimization properties GNNs can theoretically distinguish, and the **Algorithm-Unrolling (AU)-based framework**, which establishes connections between classical optimization algorithms and GNN architectures through direct algorithmic simulation.

### A.1.1   WL-BASED FRAMEWORKS

This section reviews optimization problems where GNNs have been proven to achieve universal approximation capabilities through theoretical frameworks based on Weisfeiler-Leman (WL) tests.

**Linear Programming (LP) (Chen et al., 2022b):** It establishes a foundational theoretical framework for analyzing GNN expressivity in solving LPs through WL-tests. Building upon the bipartite graph representation introduced by (Gasse et al., 2019), they demonstrate a formal connection between GNN expressivity and WL-tests on graph structures. Their key theoretical contribution proves that GNNs achieve universality over the parameter space of LPs. Specifically, they show the existence of message-passing GNNs capable of reliably approximating fundamental LP properties, including feasibility, optimal objective value, and optimal solutions.

**Mixed-Integer Linear Programming (MILP) (Chen et al., 2023):** The extension to MILP presents significant theoretical challenges not encountered in the continuous LP setting. The fundamental limitation arises from the discrete nature of integer variables, where GNN expressivity remains constrained by the discriminative power of WL-tests. A critical issue emerges: two MILP instances that are indistinguishable under WL-tests may exhibit fundamentally different properties regarding feasibility and optimal solutions. To address these challenges, the authors identify a restricted class of MILPs satisfying the "unfoldable" property, for which universality guarantees can be established. Additionally, they demonstrate that augmenting the graph representation with random node features enables GNNs to achieve universality over the complete class of MILP problems, effectively circumventing the limitations imposed by deterministic WL-tests.

**Linearly Constrained Quadratic Programming (LCQP) (Chen et al., 2024b):** While modeling linear constraints through a bipartite graph is relatively straightforward, extending graph-based approaches to handle quadratic objective functions presents challenges. It addresses this by introducing

self-connections within variable nodes to capture quadratic interactions in the objective function. Their framework extends a broader class of mixed-integer LCQP problems satisfying the MP-tractable property, establishing universality results for GNNs on specific computational tasks within this class.

The authors further extend their approach to convex quadratically constrained quadratic programming (QCQP) through dynamic edge update mechanisms, as detailed in their supplementary materials, demonstrating the framework's adaptability to more complex constraint structures.

**Convex Quadratically Constrained Quadratic Programming (QCQP) (Wu et al., 2024):** It provides a comprehensive treatment of convex QCQPs, addressing the significant complexity introduced by multiple convex quadratic constraints. The key innovation lies in their sophisticated design of edge weights and specialized GNN architecture, which together ensure that the resulting message-passing framework achieves universality for the complete class of convex QCQP problems. This represents a significant advancement in handling optimization problems with complex constraint structures through GNNs.

### A.1.2    AU-based Frameworks

Algorithm unrolling represents a fundamental approach in learning-based optimization, enhancing interpretability by directly simulating classical algorithmic procedures through neural network architectures. This section reviews successful applications of GNNs in unrolling established optimization algorithms.

**Interior Point Method :** The unrolling of Interior Point Methods (IPM) establishes a direct and interpretable correspondence between classical optimization algorithms and GNNs. (Qian et al., 2024) first provides theoretical foundations demonstrating that standard IPM iterations for LPs can be precisely simulated through sequences of GNN message-passing operations. This framework was extended to the broader class of LCQPs (Qian & Morris, 2025a), maintaining the fundamental correspondence between algorithmic steps and neural computations.

**Primal-Dual Hybrid Gradient:** The unrolling of Primal-Dual Hybrid Gradient (PDHG) algorithms provides a scalable framework for accelerating first-order optimization methods through learning-based approaches. (Li et al., 2024a) introduces PDHG-Net for large-scale LPs, demonstrating that optimal LP solutions can be approximated using polynomial-sized neural networks. This foundational work establishes both theoretical guarantees and practical scalability for the unrolled PDHG framework. The extension to QP represents another advancement (Yang et al., 2024a), which introduces an innovative unsupervised training methodology that directly incorporates Karush-Kuhn-Tucker (KKT) optimality conditions into the loss function.

**Specialized Algorithms for Structured Problems:** For optimization problems with specialized structures, researchers have developed tailored algorithmic approaches that leverage problem-specific properties for effective GNN unrolling.

For covering and packing LPs, (Li et al., 2024b) design variants of the Awerbuch-Khandekar algorithm, successfully unrolling these through careful exploitation of activation function properties. Specifically, they utilize ELU and sigmoid activation functions to simulate exponential operations and Heaviside step functions, respectively, enabling reproduction of the classical algorithm's behavior within the GNN framework.

In the context of sparse binary LPs, (Li et al., 2025a) proposes a constant-round distributed algorithm that applies to almost all sparse binary LP instances. This algorithm naturally aligns with constant-depth, constant-width GNN architectures, providing theoretical justification for the empirical success of shallow networks in this domain.

(Yau et al., 2025) demonstrates that polynomial-sized GNNs can effectively learn powerful approximation algorithms for Maximum Constraint Satisfaction Problems (Max-CSP). Their approach leverages the equivalence between projected gradient descent on low-rank vector formulations of relaxed semidefinite programs and local message-passing operations inherent in GNN architectures.

Additionally, (He & Vitercik, 2025) aligns GNN architectures with primal-dual algorithmic reasoning for minimum hitting set problems, achieving empirical success in generalization across problem sizes and out-of-distribution scenarios.

## A.2 GENERALIZATION ANALYSIS OF GNNs AND L2O PARADIGMS

**Generalization of GNNs:** Graph Neural Networks (GNNs) (Gori et al., 2005; Scarselli et al., 2008) are state-of-the-art architectures proposed for graph learning. They leverage neighborhood information to capture the structured properties of a graph. To ensure effective learning, several approaches have been introduced to study their sample complexity, which is defined as the number of data required to generalize well to unseen data from the same underlying distribution. (Scarselli et al., 2018) connects the VC dimension to network parameters, activation functions (like piecewise polynomial activation functions), and graph size. Furthermore, (D'Inverno et al., 2025) derives upper bounds for the VC dimension of Graph Neural Networks using more general Pfaffian activation functions (like sigmoid and tanh), relating generalization capacity to network hyperparameters and the number of colors determined by the Weisfeiler-Lehman test. (Morris et al., 2023) then tightly link the generalization ability to GNN expressivity via the Weisfeiler-Leman(WL) test. Further, (Franks et al., 2024) uses margin theory to show that greater expressivity only improves generalization if it also increases the margin between classes. Based on Rademacher complexity, (Garg et al., 2020) gives the first data-dependent generalization bounds for GNNs using Rademacher complexity, which are significantly tighter than previous VC-dimension-based guarantees. For node individualization schemes emerging these days, (Pellizzoni et al., 2024) uses both VC dimension and Rademacher complexity to give the generalization bound via WL-test and covering number bounds. Moreover, several researches (Ju et al., 2023; Liao et al., 2020; Tang & Liu, 2023) give the generalization bound from other perspectives, like the PAC-Bayes bound and stochastic optimization.

**Generalization of L2O paradigms:** There are two main parts of researches in the study of generalization ability of L2O paradigms: optimizer generalization, i.e., the performance gap between trained optimizees(tasks) and unseen optimizees, and optimizee generalization, i.e., the performance gap between training data and unseen test data of the same underlying optimizees (Yang et al., 2023). We only review the optimzee generalization part and the data-driven method generalization studies below.

(Balcan et al., 2021) first proposes a unified sample complexity framework for the algorithm parameter configuration based on pseudo-dimension. (Yang et al., 2023) shows that local entropy measures loss landscape flatness, similar to the Hessian. It then uses both metrics as regularizers to meta-train optimizers that provably learn to find generalizable models. (Sucker & Ochs, 2025) combines PAC-Bayesian generalization theory with variational analysis to show that a learned algorithm's trajectory will converge to a critical point with high probability on unseen problems. (Sambharya & Stellato, 2024) develops a general data-driven framework using PAC-Bayes theory to provide probabilistic performance guarantees for both classical and learned optimizers over a fixed number of iterations.

# B  PRELIMINARY AND BASIC CONCEPTS

## B.1  BASIC CONCEPTS OF SOCPs

For problem 1, we denote all the feasible solution by:

$$\mathcal{X}_{\text{feasible}} := \left\{ x \in \mathbb{R}^n \mid Fx \leq g;\ l \leq x \leq r;\ \|A_i x + b_i\|_2 \leq c_i^T x + d_i,\ \forall i \in [m] \right\}. \quad (3)$$

If $\mathcal{X}_{\text{feasible}}$ is not empty, problem 1 is said to be **feasible**; otherwise, it is said to be **infeasible**. A feasible SOCP is **bounded** if and only if the objective function is bounded from below in $\mathcal{X}_{\text{feasible}}$, i.e., $\exists a \in \mathbb{R}$ such that

$$e^T x \geq a, \forall x \in \mathcal{X}_{\text{feasible}}$$

Otherwise, the SOCP instance is **unbounded**.

For a feasible and bounded SOCP, its optimal value is defined as: $\inf \{e^T x, \forall x \in \mathcal{X}_{\text{feasible}}\}$. Moreover, $x^*$ is said to be an **optimal solution** if it's feasible and

$$e^T x^* \leq e^T x, \forall x \in \mathcal{X}_{\text{feasible}}$$

Unlike convex QCQP, an SOCP instance may not admit an optimal solution even when it's feasible and bounded (see corollary 4). Moreover, an SOCP instance can also have multiple solutions.

### B.2 EQUIVALENT FORMULATIONS OF SOCP

**Dimension Reduction of SOC Constraints**: Consider a second-order cone (SOC) constraint of the form $\|Ax + b\|_2 \leq c^T x + d$, where $A \in \mathbb{R}^{k \times n}$ has rank $r \leq \min(k, n)$. Let the singular value decomposition of $A$ be $A = U \Sigma V^T$, where $U \in \mathbb{R}^{k \times r}$ has orthonormal columns, $\Sigma \in \mathbb{R}^{r \times r}$ is diagonal with positive entries, and $V \in \mathbb{R}^{n \times r}$ has orthonormal columns.

Since $U$ has orthonormal columns, we have $U^T U = I_r$ and $U U^T$ is the orthogonal projection onto the column space of $A$. We can decompose the vector $b$ as

$$b = b_\| + b_\perp, \quad \text{where} \quad b_\| = U U^T b, \quad b_\perp = (I_k - U U^T) b \tag{4}$$

where $b_\|$ lies in the column space of $A$ and $b_\perp$ is orthogonal to it.

Define $A' = \Sigma V^T \in \mathbb{R}^{r \times n}$ and $b' = U^T b_\| \in \mathbb{R}^r$. Then:

$$A = U \Sigma V^T = U A' \tag{5}$$

and

$$Ax + b = U A'x + U U^T b + (I_k - U U^T) b = U(A'x + U^T b_\|) + b_\perp \tag{6}$$

Since $U$ has orthonormal columns and $b_\perp$ is orthogonal to the column space of $U$, we have:

$$\|Ax + b\|_2 = \|U(A'x + U^T b_\|) + b_\perp\|_2 = \left\| \begin{pmatrix} A'x + U^T b_\| \\ \|b_\perp\|_2 \end{pmatrix} \right\|_2 \tag{7}$$

This reformulation reduces the constraint to at most $r + 1$ rows, which is beneficial when $k \gg r$.

**Reformulation of SOCP to QCQP**: A SOC constraint $\|Ax + b\|_2 \leq c^T x + d$ can be equivalently written as the quadratic constraint (may be non-convex) by squaring both sides as:

$$(Ax + b)^T (Ax + b) \leq (c^T x + d)^2$$
$$x^T A^T A x + 2b^T Ax + \|b\|_2^2 \leq x^T c c^T x + 2d c^T x + d^2$$

provided that $c^T x + d \geq 0$. Rearranging terms yields:

$$x^T (A^T A - c c^T) x + 2(b^T A - d c^T) x + (\|b\|_2^2 - d^2) \leq 0 \tag{8}$$

This transformation is valid only when the right-hand side of the original SOC constraint is non-negative, which must be enforced as an additional linear constraint $c^T x + d \geq 0$.

**Reformulation of Convex QCQP to SOCP**: Conversely, we may transform convex quadratic constraints of the form $x^\top Q x + c^\top x + d \leq 0$ into SOC constraints. Since $Q \in \mathbb{S}_+^n$ is positive semidefinite, we can apply matrix decomposition $Q = L L^\top$ where $L \in \mathbb{R}^{n \times r}$ with $r = \operatorname{rank}(Q)$. This decomposition can be obtained through Cholesky factorization when $Q$ is positive definite, or through eigenvalue decomposition in the general case.

The quadratic constraint can then be reformulated as:

$$x^\top Q x + c^\top x + d \leq 0$$
$$x^\top L L^\top x + c^\top x + d \leq 0$$
$$\|L^\top x\|_2^2 + c^\top x + d \leq 0$$

Using the rotated second-order cone representation, we can reformulate the constraint as:

$$\left\| \begin{pmatrix} \frac{1 + c^\top x + d}{2} \\ L^\top x \end{pmatrix} \right\|_2 \leq \frac{1 - c^\top x - d}{2} \tag{9}$$

This formulation is valid when $1 - c^\top x - d \geq 0$, which ensures that the right-hand side is non-negative. The constraint $c^\top x + d \leq 0$ from the original quadratic form is automatically satisfied when the SOC constraint holds.

For the convex quadratic objective function $\min_x\ x^\top Q x + c^\top x + d$, we can reformulate it using an epigraph variable $\tau$:

$$\min_{x,\tau}\ \ \tau$$
$$\text{s.t.}\ \ x^\top Q x + c^\top x + d \le \tau$$

Using the matrix decomposition $Q = LL^\top$, this becomes:

$$\min_{x,\tau}\ \ \tau$$
$$\text{s.t.}\ \ \left\| \begin{pmatrix} \frac{1-\tau+c^\top x+d}{2} \\ L^\top x \end{pmatrix} \right\|_2 \le \frac{1+\tau-c^\top x - d}{2}$$

### B.3 Target Mappings for SOCP

Then, we propose all our target mappings.

**Definition B.1** (Target mappings). Let $\mathcal{G}_{\text{SOCP}}$ be a graph representation of a SOCP problem. We define the following target mappings.

- **Feasibility mapping:** We define $\Phi_{\text{feas}}(\mathcal{G}_{\text{SOCP}}) = 1$ if the SOCP problem is feasible and $\Phi_{\text{feas}}(\mathcal{G}_{\text{SOCP}}) = 0$ otherwise.

- **Boundedness mapping:** For a feasible SOCP problem, we define $\Phi_{\text{bound}}(\mathcal{G}_{\text{SOCP}}) = 1$ if the SOCP problem is bounded and $\Phi_{\text{bound}}(\mathcal{G}_{\text{SOCP}}) = 0$ otherwise.

- **Optimal value mapping:** For a feasible and bounded SOCP problem, we set $\Phi_{\text{opt}}(\mathcal{G}_{\text{SOCP}})$ to be its optimal objective value.

- **Solution Attainability Mapping :** For a feasible and bounded SOCP problem, its optimal value (infimum) is finite, but this value is not necessarily attained by a feasible point. Therefore, we introduce a mapping $\Phi_{\text{attain}}(\mathcal{G}_{\text{SOCP}})$ which equals 1 if an optimal solution exists, and 0 otherwise.

- **Optimal solution mapping:** For an SOCP problem that admits a solution, its optimal solution might not be unique. Therefore, we define the optimal solution mapping to be $\Phi_{\text{sol}}(\mathcal{G}_{\text{SOCP}}) = x^*$, where $x^*$ is the solution with the smallest $l_2$ norm of the corresponding SOCP

### B.4 An Example for SOCP graphs

Figure 6 is an example of a toy SOCP and its corresponding graph representation:

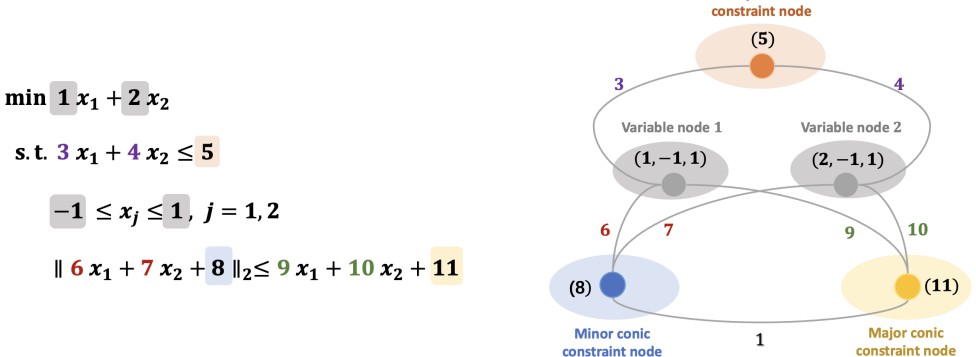

Figure 6: A toy SOCP instance with its graph representation

### B.5 COMPLEXITY COMPARISON WITH SOTA WORKS:

**Complexity for representing convex QCQP**: We discuss further about what we mentioned in remark 1, 2, 3. For a convex QCQP instance with $m$ quadratic constraints and $n$ variables, where the $i$-th constraint matrix has rank $r_i \leq n$, our graph representation requires $n + m + 2 + \sum_{i=0}^{m}(r_i + 1)$ nodes while the architecture in (Wu et al., 2024) requires $n + m + \frac{1}{2}n(n+1)$ nodes and architecture in (Chen et al., 2024b) requires $m + n + mn$ "nodes" that need to be updated dynamically. It's noteworthy that our graph representation only uses sparse connections between these nodes via using minor conic nodes as a **sparse intermediate information passing layer** between variables and conic constraints. As a result, our SOCP-GNN requires only $\mathcal{O}(n(\sum_{i=0}^{m} r_i))$ messages per iteration. This is in sharp contrast to the architecture by (Wu et al., 2024), which models each quadratic term explicitly and thus incurs a much higher per-iteration cost of $\mathcal{O}(n^3 + mn^2)$. And result in (Chen et al., 2024b) use $\mathcal{O}(mn^2)$ messages each iteration.

**Reducing the Node Complexity of SOCP-GNNs**: One may note that for SOC constraints $\|Ax + b\|_2 \leq c^\top x + d$ with $A \in \mathbb{R}^{k \times n}$ of a large $k \gg n$, the GNN need $k$ minor conic constraint nodes to represent it. However, as shown in Appendix B.1, we can reduce the complexity to $\mathcal{O}(n)$ by reformulating it into another equivalent SOC constraint with corresponding $A' \in \mathbb{R}^{k' \times n}$ of $k' \leq n + 1$. This reformulation makes SOCP-GNN more scalable for the large and structured problems in real-world applications.

## C PROOF OF MAIN THEOREM

### C.1 SOCP WL-TEST

The separation power of traditional GNNs is closely related to the Weisfeiler-Lehman (WL) test, a classical algorithm to identify whether two given graphs are isomorphic. To apply the WL test on SOCP-graphs, we design a modified WL test in Algorithm 1.

We denote Algorithm 1 by $\mathrm{WL}_{\mathrm{SOCP}}(\cdot)$ and we assume that there is no collision of Hash functions and their linear combination in the following proof (Chen et al., 2024b; Wu et al., 2024). We say that two SOCP-graphs $G, \hat{G}$ can be distinguished by Algorithm 1 if and only if there exist a positive integer $L$ and injective hash functions mentioned above such that the output multisets of $G, \hat{G}$ are different.

### C.2 THE CONNECTION BETWEEN THE WL-INDISTINGUISHABLITY AND TARGET PROPERTY

Here, we analyze the WL test's convergence and corresponding stable properties to lead to the core lemma

**Lemma 1.** *Assume all hash functions satisfy conditions in Appendix C.1, and we terminate the SOCP WL-test when the number of distinct colors no longer changes in an iteration. Then the SOCP WL-test terminates in finite iterations.*

*Proof.* Here, notice that the SOCP WL-test satisfies the following two properties:

- If two nodes $v, w$ have different colors in one (sub)iteration, then they will always have different colors in the following (sub)iterations.

- If after one full iteration, the nodes' color doesn't change under some one-to-one color mapping, then for all iterations after this iteration, the algorithm will always return the same result.

These two facts have shown that, after one iteration, the color collections either get strictly finer or remain unchanged for all following iterations. Since the number of nodes is finite, the algorithm terminates in finite iterations. $\qquad \square$

And now, we study the convergence properties of the SOCP-WL test

---

**Algorithm 1** The WL test for SOCP-Graphs (denoted by $\text{WL}_{\text{SOCP}}$)

---

1: **Require:** A graph instance $\mathcal{G} = (V, E)$ with node sets $V_1, V_2, V_3, V_4$, initial node features $h^v, h^s, h^o, h^q$, and an iteration limit $L > 0$.

2: **Initialize** initial colors for all nodes:

3: $C^{0,v} \leftarrow \text{HASH}_{0,V}(h^v), \quad \forall v \in V_1$

4: $C^{0,s} \leftarrow \text{HASH}_{0,S}(h^s), \quad \forall s \in V_2$

5: $C^{0,o} \leftarrow \text{HASH}_{0,O}(h^o), \quad \forall o \in V_3$

6: $C^{0,q} \leftarrow \text{HASH}_{0,Q}(h^q), \quad \forall q \in V_4$

7: **for** $l = 1$ **to** $L$ **do**

8:     **Update colors for polyhedron constraint nodes ($V_2$):**

9:     $C^{l,s} \leftarrow \text{HASH}\left(C^{l-1,s}, \sum_{v \in V_1} w_{s,v}\text{HASH}(C^{l-1,v})\right)$

10:     **Update colors for minor conic constraint nodes ($V_3$):**

11:     $\bar{C}^{l-1,o} \leftarrow \text{HASH}\left(C^{l-1,o}, \sum_{v \in V_1} w_{o,v}\text{HASH}(C^{l-1,v})\right)$

12:     **Update colors for major conic constraint nodes ($V_4$):**

13:     $\bar{C}^{l-1,q} \leftarrow \text{HASH}\left(C^{l-1,q}, \sum_{v \in V_1} w_{q,v}\text{HASH}(C^{l-1,v})\right)$

14:     **Update colors for major conic constraint nodes ($V_4$):**

15:     $C^{l,q} \leftarrow \text{HASH}\left(\bar{C}^{l-1,q}, \sum_{o \in V_3} w_{q,o}\text{HASH}(\bar{C}^{l-1,o})\right)$

16:     **Update colors for minor conic constraint nodes ($V_3$):**

17:     $C^{l,o} \leftarrow \text{HASH}\left(\bar{C}^{l-1,o}, \sum_{q \in V_4} w_{o,q}\text{HASH}(C^{l,q})\right)$

18:     **Update colors for variable nodes ($V_1$):**

19:     $C^{l,v} \leftarrow \text{HASH}\left(C^{l-1,v}, M_1, M_2, M_3\right)$,where:

$$M_1 = \sum_{s \in V_2} w_{v,s}\text{HASH}(C^{l,s})$$

$$M_2 = \sum_{o \in V_3} w_{v,o}\text{HASH}(C^{l,o})$$

$$M_3 = \sum_{q \in V_4} w_{v,q}\text{HASH}(C^{l,q})$$

20: **end for**

21: **Return** The multisets of final colors: $\{\{C^{L,v}\}\}_{v \in V_1}, \{\{C^{L,s}\}\}_{s \in V_2}, \{\{C^{L,o}\}\}_{o \in V_3}, \{\{C^{L,q}\}\}_{q \in V_4}$

---

**Lemma 2.** *Given the SOCP graph $G$, assume the SOCP WL-test stabilizes after $T \geq 0$ iterations. The sum of weights from a certain node of one color to all nodes of another color only depends on the color of the given node. Specifically, the sum (taking $W_1$ for variable nodes and $W_2$ for polyhederon constraint nodes as an example) is:*

$$S(W_2, W_1; G) := \sum_{C^{T,v}=W_1} w_{s,v}$$

*and is well-defined for all $s$, such that $C^{T,s} = W_2$*

*Similarly, for any color of variables $W_1$, color of polyhedron constraints $W_2$, color of minor conic constraints $W_3$ and color of major conic constraints $W_4$, the following sums are well-defined:*

$$S(W_3, W_1; G) := \sum_{C^{T,v}=W_1} w_{o,v}, \quad C^{T,o} = W_3$$

$$S(W_4, W_3; G) := \sum_{C^{T,o}=W_3} w_{q,o}, \quad C^{T,q} = W_4$$

$$S(W_1, W_2; G) := \sum_{C^{T,s}=W_2} w_{v,s}, \quad C^{T,v} = W_1$$

$$S(W_1, W_3; G) := \sum_{C^{T,o}=W_3} w_{v,o}, \quad C^{T,v} = W_1$$

$$S(W_1, W_4; G) := \sum_{C^{T,q}=W_4} w_{v,q}, \quad C^{T,v} = W_1$$

$$S(W_4, W_1; G) := \sum_{C^{T,v}=W_1} w_{q,v}, \quad C^{T,q} = W_4$$

*Proof.* Let $v_1, v_2$ be two nodes with color $W_1 = C^{T,v_1} = C^{T,v_2}$. Since the SOCP WL-test has stabilized, the node pairs won't be finer, i.e.

$$\sum_s w_{v_1,s}\text{HASH}(C^{T,s}) = \sum_s w_{v_2,s}\text{HASH}(C^{T,s}).$$

Rearranging according to $W_2 = C^{T,s}$, we get:

$$\sum_{W_2} \sum_{C^{T,s}=W_2} w_{v_1,s}\text{HASH}(W_2) = \sum_{W_2} \sum_{C^{T,s}=W_2} w_{v_2,s}\text{HASH}(W_2).$$

Assuming that the hash function is collision-free and maps different colors into different linearly independent vectors, we conclude that:

$$\sum_{C^{T,s}=W_2} w_{v_1,s} = \sum_{C^{T,s}=W_2} w_{v_2,s},$$

i.e., $S(W_1, W_2; G) := \sum_{C^{T,s}=W_2} w_{v,s}, \quad C^{T,v} = W_1$ is well-defined.

Other proofs are similar.

$\square$

An immediate conclusion is listed following.

**Corollary 1.** *If all the SOCP WL-test cannot separate the two instances: $\mathcal{I}, \hat{\mathcal{I}}$ (with given sizes $n, m, , k_1, ..., k_m, b$, encoded by $G, \hat{G} \in \mathcal{G}_{SOCP}^{n,m,k_1,...,k_m,b}$), then: All the sum in lemma 3 is well defined for $G$ and $\hat{G}$ and equal each other respectively.*

Meanwhile, we define: $W_{ij}$ to be the collection of nodes with node type i and color j. By summing the cross terms and rearranging the sum, we have:

$$S(W_{1j}, W_{2k}; G)|W_{1j}| = S(W_{2k}, W_{1j}; G)|W_{2k}|$$

$$S(W_{1j}, W_{3l}; G)|W_{1j}| = S(W_{3l}, W_{1j}; G)|W_{3l}|$$

$$S(W_{1j}, W_{4m}; G)|W_{1j}| = S(W_{4m}, W_{1j}; G)|W_{4m}|$$

.

Now, we begin to prove the following lemma.

**Lemma 3.** *Let $\mathcal{I}, \hat{\mathcal{I}}$ (with given sizes $n, m, k_1, ..., k_m, b$, encoded by $G, \hat{G} \in \mathcal{G}_{SOCP}^{n,m,k_1,...,k_m,b}$) be two SOCP instances. If the following holds:*

- *The SOCP WL-test cannot separate the two instances;*

- *$x$ is a feasible solution of $\mathcal{I}$.*

*Then there exists a feasible solution $\hat{x}$ for $\hat{\mathcal{I}}$ whose objective and $\ell_2$-norm are controlled by $x$, such that:*

$$\hat{e} \cdot \hat{x} \leq e \cdot x$$
$$||\hat{x}||_2 \leq ||x||_2$$

*Proof.* The key to this proof is to take the average among the nodes in the same node pair. Formally, we define $\hat{x}_v = \frac{1}{|W_{1j}|}(\sum_{C^{T,v'}=W_{1j}} x_{v'})$ for all $v$, such that: $C^{T,v} = W_{1j}$

By the Cauchy-Schwarz inequality, we have:

$$\sum_{C^{T,v'}=W_{1j}} x_{v'}^2 \geq |W_{1j}|[\frac{1}{|W_{1j}|}(\sum_{C^{T,v'}=W_{1j}} x_{v'})]^2$$

Summing over all possible $W_{1j}$, we get: $||\hat{x}||_2 \leq ||x||_2$

Meanwhile, notice that: for all $v'$, such that: $C^{T,v'} = W_{1j}$, their corresponding $e_{v'}, l_{v'}, r_{v'}$ and $\hat{e}_{v'}, \hat{l}_{v'}, \hat{r}_{v'}$ are the same, respectively.

Hence, we have:

$$\sum_{C^{T,v'}=W_{1j}} e_{v'} x_{v'} = \sum_{C^{T,v'}=W_{1j}} \hat{e}_{v'} \hat{x}_{v'}$$

$$\hat{x}_v \in [\hat{l}_v, \hat{r}_v]$$

for all possible variable node v and color $W_{1j}$.

Summing over $W_{1j}$ yields:

$$\sum_{v'} e_{v'} x_{v'} = \sum_{v'} \hat{e}_{v'} \hat{x}_{v'}$$

Further, consider the edge properties brought by the above lemma, we get:

For the l-th and t-th polyhedron constraint both with color $W_{2k}, l, t \in \{1, 2, \cdots, b\}$ (Here, we assume in both $G$ and $\hat{G}$, the l-th and t-th polyhedron constraint are both with color $W_{2k}$ respectively) the following inequality holds:

$$\sum_{j=1}^{n} F_{l,j} x_j \leq g_t, \quad \Rightarrow \quad \frac{1}{|W_{2k}|} \sum_{l \in W_{2k}} \sum_{W_{1j}} \sum_{v \in W_{1j}} F_{l,v} x_v \leq g_t,$$

Exchange the order of the sum, we get:

$$\frac{1}{|W_{2k}|} \sum_{W_{1j}} \sum_{v \in W_{1j}} \sum_{l \in W_{2k}} F_{l,v} x_v \leq g_t,$$

Notice that:

$$\frac{1}{|W_{2k}|} \sum_{W_{1j}} \sum_{v \in W_{1j}} \sum_{l \in W_{2k}} F_{l,v} x_v = \frac{1}{|W_{2k}|} \sum_{W_{1j}} \sum_{v \in W_{1j}} \sum_{l \in W_{2k}} F_{l,v} \hat{x}_v$$

$$= \frac{1}{|W_{2k}|} \sum_{W_{1j}} \sum_{v \in W_{1j}} \sum_{l \in W_{2k}} \hat{F}_{l,v} \hat{x}_v$$

$$= \sum_{W_{1j}} \sum_{v \in W_{1j}} \hat{F}_{l,v} \hat{x}_v$$

Thus, $\sum_{W_{1j}} \sum_{v \in W_{1j}} \hat{F}_{l,v} \hat{x}_v \leq g_t = \hat{g}_t = \hat{g}_l$, which shows that the polyhedron constraint is satisfied.

Similarly, for the u-th and r-th (major) conic constraints both with color $W_{4m}$, we have:

- After proper rearranging of nodes $o_{us_u}$ and $o_{rs_r}$, where $s_u = 1, 2, ..., k_u; s_r = 1, 2, ..., k_r$, the color of $o_{us_u}$ and $o_{rs_r}$, where $s_u = 1, 2, ..., k_u; s_r = 1, 2, ..., k_r$, are the same regarding the order, i.e. $C^{T,o_{ui}} = C^{T,o_{ri}}, \forall i = 1, 2, ..., k_u$ (Notice that :$k_u = k_r$).

- $d_u = d_r$.

- For any node $o_{hi}$ and $o_{jk} \in V_3$ with the same stable color, either $h = j$ or the color of node $q_h$ and $q_j$ are the same.

Now let's prove the conic part. For the u-th and r-th major conic constraint node both with color $W_{4m}$ in both $G$ and $\hat{G}$ and the minor conic node $j_1$ corresponds to r-th major conic constraint node in both $G$ and $\hat{G}$, we have:

**Right constraint:**

$$\frac{1}{|W_{4m}|} \sum_{C^{T,u}=W_{4m}} c_u^T x + d_u$$

$$= \left( \frac{1}{|W_{4m}|} \sum_{C^{T,u}=W_{4m}} \sum_{W_{1j}} \sum_{v \in W_{1j}} c_{uv} x_v \right) + \hat{d}_r$$

$$= \left( \frac{1}{|W_{4m}|} \sum_{W_{1j}} \sum_{v \in W_{1j}} \sum_{C^{T,u}=W_{4m}} c_{uv} x_v \right) + \hat{d}_r$$

$$= \left( \frac{1}{|W_{4m}|} \sum_{W_{1j}} \sum_{v \in W_{1j}} S(W_{1j}, W_{4m}; G) x_v \right) + \hat{d}_r$$

$$= \left( \frac{1}{|W_{4m}|} \sum_{W_{1j}} \sum_{v \in W_{1j}} S(W_{1j}, W_{4m}; G) \hat{x}_v \right) + \hat{d}_r$$

$$= \left( \frac{1}{|W_{4m}|} \sum_{W_{1j}} |W_{1j}| S(W_{1j}, W_{4m}; G) \hat{x}_v \right) + \hat{d}_r$$

$$= \left( \frac{1}{|W_{4m}|} \sum_{W_{1j}} |W_{4m}| S(W_{4m}, W_{1j}; G) \hat{x}_v \right) + \hat{d}_r$$

$$= \left( \sum_{W_{1j}} S(W_{4m}, W_{1j}; G) \hat{x}_v \right) + \hat{d}_r = \left( \sum_{W_{1j}} \sum_{v \in W_{1j}} \hat{c}_{rv} \hat{x}_v \right) + \hat{d}_r = \hat{c}_r^T \hat{x} + \hat{d}_r$$

**Left Constraint:** Recall: two nodes in $V_3$ has the same stable color $W_{3l}$ if their corresponding major conic constraint node's stable color is the same. So for each stable color $W_{3l}$ in $V_3$, the corresponding major conic node's stable colors are all the same, denoted by $W_{4m}$. And each major conic node have $\frac{|W_{3l}|}{|W_{4m}|}$ minor nodes with stable color $W_{3l}$. And we use $j \in u$ denotes a minor conic node $j$ corresponds to node $u$

$$\frac{1}{|W_{4m}|} \sum_{C^{T,u}=W_{4m}} \sum_{j \in W_{3l}, j \in u} \frac{|W_{4m}|}{|W_{3l}|} (A_u x + b_u)_j$$

$$= (\hat{b}_r)_{j_1} + \frac{1}{|W_{3l}|} \sum_{C^{T,u}=W_{4m}} \sum_{j \in W_{3l}, j \in u} (A_u x)_j$$

$$= (\hat{b}_r)_{j_1} + \frac{1}{|W_{3l}|} \sum_{C^{T,u}=W_{4m}} \sum_{j \in W_{3l}, j \in u} \sum_{W_{1h}} \sum_{v \in W_{1h}} (A_u)_{jv} x_v$$

$$= (\hat{b}_r)_{j_1} + \frac{1}{|W_{3l}|} \sum_{W_{1h}} \sum_{v \in W_{1h}} \sum_{C^{T,u}=W_{4m}} \sum_{j \in W_{3l}, j \in u} (A_u)_{jv} x_v$$

$$= (\hat{b}_r)_{j_1} + \frac{1}{|W_{3l}|} \sum_{W_{1h}} \sum_{v \in W_{1h}} S(W_{1h}, W_{3l}; G) x_v$$

$$= (\hat{b}_r)_{j_1} + \frac{1}{|W_{3l}|} \sum_{W_{1h}} \sum_{v \in W_{1h}} S(W_{1h}, W_{3l}; G) \hat{x}_v$$

$$= (\hat{b}_r)_{j_1} + \frac{1}{|W_{3l}|} \sum_{W_{1h}} |W_{1h}| S(W_{1h}, W_{3l}; G) \hat{x}_v$$

$$= (\hat{b}_r)_{j_1} + \frac{1}{|W_{3l}|} \sum_{W_{1h}} |W_{3l}| S(W_{3l}, W_{1h}; G) \hat{x}_v$$

$$= (\hat{b}_r)_{j_1} + \sum_{W_{1h}} S(W_{3l}, W_{1h}; G) \hat{x}_v$$

$$= (\hat{b}_r)_{j_1} + \sum_{W_{1h}} \sum_{v \in W_{1h}} \hat{(A_r)}_{j_1 v} \hat{x}_v$$

$$= (\hat{b}_r)_{j_1} + (\hat{A}_r \hat{x})_{j_1} = (\hat{b}_r + \hat{A}_r \hat{x})_{j_1}$$

**Conic feasibility for $\hat{I}$**

For the u-th conic constraint with stable color $W_{4m}$ and its corresponds minor conic node $j$ with color $W_{3l}$, without loss of generality, we assume $(A_u x + b_u)$ 's first $N = \frac{|W_{3l}|}{|W_{4m}|}$ rows' corresponding to all the minor conic nodes with color $W_{3l}$ for such u and $j$, Thus, we have:

$$\sum_{j_1 \in r, j_1 \in W_{3l}} ||(\hat{b}_r + \hat{A}_r \hat{x})_{j_1}||_2^2 = \frac{|W_{3l}|}{|W_{4m}|} ||\frac{1}{|W_{4m}|} \sum_{C^{T,u}=W_{4m}} \sum_{j=1}^{N} \frac{|W_{4m}|}{|W_{3l}|} (A_u x + b_u)_j||_2^2$$

$$\leq \sum_{j=1}^{N} ||\frac{1}{|W_{4m}|} \sum_{C^{T,u}=W_{4m}} (A_u x + b_u)_j||_2^2$$

Hence, summing over all possible $W_{3l}$ for fixed $W_{4m}$, we have:

$$||\hat{b}_r + \hat{A}_r \hat{x}||_2^2 = \sum_{W_{3l}} \sum_{j_1 \in r, j_1 \in W_{3l}} ||(\hat{b}_r + \hat{A}_r \hat{x})_{j_1}||_2^2 \leq ||\frac{1}{|W_{4m}|} \sum_{C^{T,u}=W_{4m}} (A_u x + b_u)||_2^2$$

This yields that:

$$||\hat{b}_r + \hat{A}_r \hat{x}||_2 \leq ||\frac{1}{|W_{4m}|} \sum_{C^{T,u}=W_{4m}} (A_u x + b_u)||_2 \leq \frac{1}{|W_{4m}|} \sum_{C^{T,u}=W_{4m}} ||(A_u x + b_u)||_2$$

$$\leq \frac{1}{|W_{4m}|} \sum_{C^{T,u}=W_{4m}} c_u^T x + d_u = \hat{c}_r^T \hat{x} + \hat{d}_r$$

Since r is arbitrarily chosen from $W_{4m}$, the conic feasibility holds obviously. $\qquad \square$

**Corollary 2.** *If two SOCP instances $\mathcal{I}, \hat{\mathcal{I}}$ with their graph representations are indistinguishable by the SOCP-WL test, then the two problems share the same feasibility.*

*Proof.* Let $x$ be a feasible solution for $\mathcal{I}$, then by lemma 3, there exists a feasible solution $\hat{x}$ for $\hat{\mathcal{I}}$. $\qquad \square$

**Corollary 3.** *If two SOCP instances $\mathcal{I}, \hat{\mathcal{I}}$ with their graph representations are indistinguishable by the SOCP-WL test, then the two problems share the same boundness.*

*Proof.*
- If one instance is infeasible, by corollary 2, the other instance is infeasible as well, i.e., they are not bounded.

- If one instance is not bounded from below, denoted by $\mathcal{I}$. Since we can always find a feasible solution of $\hat{\mathcal{I}}$ which has a smaller objective value than any fixed feasible solution of $\mathcal{I}$ by Lemma 3, the conclusion is obvious.

$\qquad \square$

**Corollary 4.** *If two SOCP instances $\mathcal{I}, \hat{\mathcal{I}}$ with their graph representations are indistinguishable by the SOCP-WL test, then the two problems share the same optimal objective value.*

*Proof.* By corollary 3, we only need to consider the case when both instances are feasible and bounded.

Notice that the feasibility with boundness may not lead to the existence of an optimal solution for SOCP problems, for example:

$$\min_{x_1, x_2} \quad x_1 \text{ s.t.} \quad ||(2, x_1 - x_2)||_2 \leq x_1 + x_2 \,, x_1 \geq 0, x_2 \geq 0$$

So, we prove by "infimum" argument, let $p$ and $\hat{p}$ be the optimal value of $\mathcal{I}$ and $\hat{\mathcal{I}}$ respectively. Then for any $\epsilon > 0$, there exists feasible solution $x$, s.t. $e^T x \leq p + \epsilon$. By lemma 4, there exists a feasible solution $\hat{x}$ of $\hat{\mathcal{I}}$, such that $\hat{p} \leq \hat{e}^T \hat{x} \leq e^T x \leq p + \epsilon$. Let $\epsilon \to 0$ yields $\hat{p} \leq p$. Similarly, we have: $\hat{p} \geq p$, which finishes the proof. $\square$

**Corollary 5.** *If two SOCP instances $\mathcal{I}, \hat{\mathcal{I}}$ with their graph representations are indistinguishable by the SOCP-WL test and one of these instances admits an optimal solution, then the other instance has an optimal solution as well.*

*Proof.* See the proof of corollary 6. $\square$

**Corollary 6.** *If two SOCP instances $\mathcal{I}, \hat{\mathcal{I}}$ with their graph representations are indistinguishable by the SOCP-WL test, then the two problems share the same optimal solution with the smallest Euclidean norm if one instance admits an optimal solution up to permutation.*

*Proof.* Without loss of generality, we assume that for each variable j, its corresponding stable color in $\mathcal{I}, \hat{\mathcal{I}}$ after the SOCP-WL test is the same, and $\mathcal{I}$ has an optimal solution $x$ with the smallest Euclidean norm. By using lemma 3 twice, we can construct a feasible solution $\hat{x}$ for $\hat{\mathcal{I}}$ and construct a feasible solution $\hat{\hat{x}}$ for $\mathcal{I}$ again with

$$e^T x \geq \hat{e}^T \hat{x} \geq e^T \hat{\hat{x}} \quad and \quad ||x||_2 \geq ||\hat{x}||_2 \geq ||\hat{\hat{x}}||_2$$

Hence, $x = \hat{\hat{x}}$. Recall the proof of the lemma 3, the variables in $\hat{x}$ with the same stable color after SOCP-WL test already have the same value, so averaging them again won't change it anymore, i.e. $\hat{x} = \hat{\hat{x}}$. Hence, $x = \hat{x} = \hat{\hat{x}}$. By corollary 4, $\hat{x}$ is an optimal solution of $\hat{\mathcal{I}}$

Now, if there exists an optimal solution $y$ of $\hat{\mathcal{I}}$ with $||y||_2 < ||\hat{x}||_2 = ||x||_2$, by similar proof above, we can get: $y$ is also an optimal solution of $\mathcal{I}$, which contradicts the fact that: $x$ is the optimal solution of $\mathcal{I}$ with the smallest Euclidean norm. Hence, $\hat{x}$ is an optimal solution of $\hat{\mathcal{I}}$ with the smallest Euclidean norm $\square$

### C.3 THE MEASURABLE PROPERTY OF TARGET MAPPING

**Definition C.1.** For an SOCP instance G:

$$\begin{aligned}
\text{minimize} \quad & e^T x \\
\text{subject to} \quad & ||A_i x + b_i||_2 \leq c_i^T x + d_i, \quad i = 1, \ldots, m \\
& Fx \leq g \\
& l_i \leq x_i \leq r_i, i = 1, \ldots, n
\end{aligned}$$

Its parameter is defined as $(e, \{A_i\}_{i=1}^m, \{b_i\}_{i=1}^m, \{c_i\}_{i=1}^m, \{d_i\}_{i=1}^m, F, g, l, r)$, and all the parameter form the parameter space $\mathcal{P}$

Notice that: For an SOCP instance, there exists a bijective mapping $\mathbf{I} : \mathcal{G}_{SOCP}^{n,m,k_1,\ldots,k_m,b} \to \mathcal{P}$ with $\mathbf{I}(G) = (e, \{A_i\}_{i=1}^m, \{b_i\}_{i=1}^m, \{c_i\}_{i=1}^m, \{d_i\}_{i=1}^m, F, g, l, r)$ for any SOCP instance G parametrized by $(e, \{A_i\}_{i=1}^m, \{b_i\}_{i=1}^m, \{c_i\}_{i=1}^m, \{d_i\}_{i=1}^m, F, g, l, r)$. And we equip both $\mathcal{G}_{SOCP}^{n,m,k_1,\ldots,k_m,b}$ and $\mathcal{P}$ with the standard Euclidean topology and product topology in its feature space. Then $\mathbf{I}$ is a homeomorphism.

**Remark:** If we can prove that $\Phi_{target} : \mathcal{P} \to \mathbb{R}$ is measurable, then $\Phi_{target} \circ \mathbf{I} : \mathcal{G}_{SOCP}^{n,m,k_1,\ldots,k_m,b} \to \mathbb{R}$ is measurable as well.

**Theorem 4.** *For any Borel regular measure $\mu$ defined on $\mathcal{P}$, $\Phi_{feas} : \mathcal{P} \to \{0, 1\}$ is $\mu-$measurable.*

*Proof.* Below, we use measurable to denote $\mu-$measurable for simplicity.

To prove that $\Phi_{\text{feas}}$ is measurable, it suffices to show that the preimage of $\{1\}$, denoted $\mathcal{P}_{\text{feas}} = \{P \in \mathcal{P} \mid \Phi_{\text{feas}}(P) = 1\}$, is a measurable set.

First, we define a **feasibility violation function** $V_{\text{feas}} : \mathcal{P} \times \mathbb{R}^n \to \mathbb{R}_{\geq 0}$. Let $(y)_+ = \max(0, y)$.

$$V_{\text{feas}}(P, x) = \sum_{i=1}^{m} \left( \|A_i x + b_i\|_2 - (c_i^T x + d_i) \right)_+ + \sum_{j=1}^{p} ((Fx)_j - g_j)_+ + \sum_{k=1}^{n} ((l_k - x_k)_+ + (x_k - r_k)_+)$$

This function $V_{\text{feas}}(P, x)$ is continuous with respect to both $P$ and $x$, as it is a sum and composition of continuous functions (norms, linear maps, max function). Furthermore, $V_{\text{feas}}(P, x) = 0$ if and only if $x$ is a feasible point for the problem instance $P$.

A problem $P$ is feasible if and only if there exists an $x \in \mathbb{R}^n$ such that $V_{\text{feas}}(P, x) = 0$. This is equivalent to the condition $\exists R \in \mathbb{N}^+, s.t. \inf_{x \in \mathbb{R}^n \cap B_R} V_{\text{feas}}(P, x) = 0$.

We can now express the set of feasible problems $\mathcal{P}_{\text{feas}}$ by restricting the infimum to a countable dense set. Let $B_R$ be the closed ball of radius $R$ centered at the origin. By continuity of $V_{\text{feas}}$ in $x$, we have:

$$\mathcal{P}_{\text{feas}} = \bigcup_{R \in \mathbb{N}^+} \left\{ P \in \mathcal{P} \mid \inf_{x \in \mathbb{R}^n \cap B_R} V_{\text{feas}}(P, x) = 0 \right\}$$

$$= \bigcup_{R \in \mathbb{N}^+} \bigcap_{k \in \mathbb{N}^+} \left\{ P \in \mathcal{P} \mid \exists x \in \mathbb{R}^n \cap B_R, s.t. V_{\text{feas}}(P, x) < \frac{1}{k} \right\}$$

So, $\mathcal{P}_{\text{feas}}$ can be written as:

$$\mathcal{P}_{\text{feas}} = \bigcup_{R \in \mathbb{N}^+} \bigcap_{k \in \mathbb{N}^+} \bigcup_{x \in B_R \cap \mathbb{Q}^n} \left\{ P \in \mathcal{P} \mid V_{\text{feas}}(P, x) < \frac{1}{k} \right\}$$

For any fixed $x \in \mathbb{Q}^n$, the function $P \mapsto V_{\text{feas}}(P, x)$ is continuous. Thus, for each tuple $(R, k, x)$, the set $\{P \mid V_{\text{feas}}(P, x) < 1/k\}$ is a Borel set. Since $\mathcal{P}_{\text{feas}}$ is formed by countable unions and intersections of measurable sets, it is itself a measurable (Borel) set. Therefore, $\Phi_{\text{feas}}$ is a measurable function.

$\square$

**Theorem 5.** *For any Borel regular measure $\mu$ defined on $\mathcal{P}$, $\Phi_{bound} : \mathcal{P} \to \{0, 1\}$ is $\mu-$measurable.*

*Proof.* Below, we use measurable to denote $\mu-$measurable for simplicity.

Let $\mathcal{P}_{\text{feas}} = \Phi_{\text{feas}}^{-1}(1)$, which is a measurable set. We only need to show that the set $\mathcal{P}_{\text{bdd}} = \{P \in \mathcal{P}_{\text{feas}} \mid \Phi_{\text{bound}}(P) = 1\}$ is measurable.

A problem $P \in \mathcal{P}_{\text{feas}}$ is bounded if and only if there exists $M \in \mathbb{Z}$ such that for all feasible solutions $x$ of $P$, $e^T x \geq M$. This can be stated as:

$$\mathcal{P}_{\text{bdd}} = \bigcup_{M \in \mathbb{Z}} \left\{ P \in \mathcal{P}_{\text{feas}} \mid \forall x \in \mathbb{R}^n, s.t. V_{\text{feas}}(P, x) = 0 \Rightarrow e^T x \geq M \right\}$$

$$= \bigcup_{M \in \mathbb{Z}} \left\{ P \in \mathcal{P}_{\text{feas}} \mid \inf_{x \in \mathbb{R}^n, s.t. V_{\text{feas}}(P, x) = 0} e^T x \geq M \right\}$$

Let's define the **boundness violation function**:

$$V_{\text{bdd}}(P, x) = \inf_{x \in \mathbb{R}^n, s.t. V_{\text{feas}}(P, x) = 0} e^T x$$

Now, we have:

$$\mathcal{P}_{\text{bdd}} = \bigcup_{M \in \mathbb{Z}} \{P \in \mathcal{P}_{\text{feas}} \mid V_{\text{bdd}}(P, x) \geq M\}$$

So it suffices to prove $V_{\text{bdd}}(P, x)$ is measurable and we only need to show that: for any $M \in \mathbb{R}$, the sublevel set $\{P \in \mathcal{P}_{\text{feas}} \mid V_{\text{bdd}}(P) < M\}$ is a measurable set.

The condition $V_{\text{bdd}}(P) < M$ is equivalent to the existence of a feasible point $z$ such that $e^\top z < M$. This can be expressed as:

$$\{P \in \mathcal{P}_{\text{feas}} \mid V_{\text{bdd}}(P) < M\} = \bigcup_{k \in \mathbb{N}_+} \left\{P \in \mathcal{P}_{\text{feas}} \mid \exists z \in \mathbb{R}^n \text{ s.t. } e^\top z \le M - \frac{1}{k} \text{ and } z \text{ is feasible}\right\}.$$

Let us define an auxiliary violation function $V_{\text{bdd\_viol}} : \mathcal{P} \times \mathbb{R}^n \times \mathbb{R} \to \mathbb{R}_{\ge 0}$:

$$V_{\text{bdd\_viol}}(P, z, M) = \max\left((e^\top z - M)_+, V_{\text{feas}}(P, z)\right).$$

This function is continuous in $(P, z, M)$. The condition $V_{\text{bdd\_viol}}(P, z, M') = 0$ holds if and only if $z$ is a feasible point and its objective value satisfies $e^\top z \le M'$.

Thus, similar to the proof of feasibility, the condition $V_{\text{bdd}}(P) < M$ is equivalent to:

$$\bigcup_{k \in \mathbb{N}_+} \bigcup_{R \in \mathbb{N}^+} \left\{P \in \mathcal{P}_{\text{feas}} \mid \inf_{z \in \mathbb{R}^n \cap B_R} V_{\text{bdd\_viol}}\left(P, z, M - \frac{1}{k}\right) = 0\right\}.$$

By continuity of $V_{\text{bdd\_viol}}$ in $z$, we can restrict the infimum to the countable dense set $\mathbb{Q}^n$:

$$\bigcup_{k \in \mathbb{N}_+} \bigcup_{R \in \mathbb{N}^+} \left\{P \in \mathcal{P}_{\text{feas}} \mid \inf_{z \in \mathbb{Q}^n \cap B_R} V_{\text{bdd\_viol}}\left(P, z, M - \frac{1}{k}\right) = 0\right\}.$$

For any fixed $z \in \mathbb{Q}^n$, $R \in \mathbb{N}^+$ and $M' \in \mathbb{R}$, the function $P \mapsto V_{\text{bdd\_viol}}(P, z, M')$ is continuous, hence measurable. The infimum of a countable collection of such measurable functions is also measurable. Therefore, the set $\{P \mid \inf_{z \in \mathbb{Q}^n} V_{\text{bdd\_viol}}(P, z, M') = 0\}$ is measurable for any fixed $M'$.

Since the sublevel set $\{P \in \mathcal{P}_{\text{feas}} \mid V_{\text{bdd}}(P) < M\}$ is a countable union of such measurable sets, it is measurable. This holds for all $M \in \mathbb{R}$, so $V_{\text{bdd}}$ is a measurable function. $\qquad\square$

**Theorem 6.** *For any Borel regular measure $\mu$ defined on $\mathcal{P}$, $\Phi_{obj} : \mathcal{P} \to \mathbb{R}$ is $\mu-$measurable.*

*Proof.* Below, we use measurable to denote $\mu-$measurable for simplicity.

To prove that $\Phi_{\text{obj}}$ is measurable, we only need to show that for any $\phi \in \mathbb{R}$, the sublevel set $\{P \in \mathcal{P} \mid \Phi_{\text{obj}}(P) < \phi\}$ is measurable.

Let us define an **objective violation function** $V_{\text{obj}} : \mathcal{P} \times \mathbb{R}^n \times \mathbb{R} \to \mathbb{R}_{\ge 0}$:

$$V_{\text{obj}}(P, x, \phi) = \max\left((e^T x - \phi)_+, V_{\text{feas}}(P, x)\right)$$

This function is continuous in $(P, x, \phi)$. $V_{\text{obj}}(P, x, \phi) = 0$ if and only if $x$ is a feasible point and its objective value satisfies $e^T x \le \phi$.

The condition $\Phi_{\text{obj}}(P) < \phi$ is equivalent to the existence of a feasible point $x$ such that $e^T x < \phi$. This can be expressed as:

$$\{P \in \mathcal{P} \mid \Phi_{\text{obj}}(P) < \phi\} = \bigcup_{k \in \mathbb{N}^+} \{P \in \mathcal{P} \mid \exists x \in \mathbb{R}^n \text{ s.t. } e^T x \le \phi - \frac{1}{k} \text{ and } x \text{ is feasible}\}$$

Similar to the previous proof, this is equivalent to:

$$\bigcup_{k \in \mathbb{N}^+} \bigcup_{R \in \mathbb{N}^+} \left\{P \in \mathcal{P} \mid \inf_{x \in \mathbb{R}^n \cap B_R} V_{\text{obj}}\left(P, x, \phi - \frac{1}{k}\right) = 0\right\}$$

$$= \bigcup_{k \in \mathbb{N}^+} \bigcup_{R \in \mathbb{N}^+} \left\{P \in \mathcal{P} \mid \inf_{x \in \mathbb{Q}^n \cap B_R} V_{\text{obj}}\left(P, x, \phi - \frac{1}{k}\right) = 0\right\}$$

For any fixed $x \in \mathbb{Q}^n$, the function $P \mapsto V_{\text{obj}}(P, x, \phi')$ is continuous, hence measurable. The infimum of a countable collection of measurable functions is measurable. Hence, the set $\{P \mid \inf_{x \in \mathbb{Q}^n} V_{\text{obj}}(P, x, \phi') = 0\}$ is measurable for any fixed $\phi'$. Since the sublevel set $\{P \mid \Phi_{\text{obj}}(P) < \phi\}$ is a countable union of such measurable sets, it is measurable. This holds for all $\phi \in \mathbb{R}$, so $\Phi_{\text{obj}}$ is a measurable function. $\qquad\square$

**Theorem 7.** *For any Borel regular measure $\mu$ defined on $\mathcal{P}$, $\Phi_{attain} : \mathcal{P} \to \{0, 1\}$ is $\mu-$measurable.*

*Proof.* Below, we use measurable to denote $\mu-$measurable for simplicity.

Let $\mathcal{P}_{\text{fin}} = \Phi_{\text{obj}}^{-1}(\mathbb{R})$, which is a measurable set. We only need to show that the set $\mathcal{P}_{\text{sol}} = \{P \in \mathcal{P}_{\text{fin}} \mid \Phi_{\text{attain}}(P) = 1\}$ is measurable.

A problem $P \in \mathcal{P}_{\text{fin}}$ attains its optimal solution if and only if there exists a point $x \in \mathbb{R}^n$ such that $x$ is feasible and its objective value is equal to the optimal value, $\Phi_{\text{obj}}(P)$. This can be stated as:

$$\mathcal{P}_{\text{sol}} = \left\{ P \in \mathcal{P}_{\text{fin}} \mid \exists x \in \mathbb{R}^n \text{ s.t. } V_{\text{feas}}(P, x) = 0 \text{ and } e^T x = \Phi_{\text{obj}}(P) \right\}$$

Let's define the **optimality violation function**:

$$V_{\text{solu}}(P, x) = \max \left( (e^T x - \Phi_{\text{obj}}(P))_+, V_{\text{feas}}(P, x) \right)$$

Notice that:

- For a fixed $x$, the function $P \mapsto V_{\text{solu}}(P, x)$ is measurable because it is a "composition" of continuous functions and the measurable function $\Phi_{\text{obj}}$.

- For a fixed $P$, the function $x \mapsto V_{\text{solu}}(P, x)$ is continuous.

A SOCP instance $P$ attains its solution if and only if there exists $R \in \mathbb{N}^+$, s.t. the infimum of $V_{\text{solu}}(P, x)$ over $x \in B_R$ is zero, i.e. :

$$\mathcal{P}_{\text{sol}} = \bigcup_{R \in \mathbb{N}^+} \left\{ P \in \mathcal{P}_{\text{fin}} \mid \inf_{x \in \mathbb{R}^n \cap B_R} V_{\text{solu}}(P, x) = 0 \right\}$$

Following the same logic used for $\Phi_{\text{feas}}$, we can write:

$$\mathcal{P}_{\text{sol}} = \bigcup_{R \in \mathbb{N}^+} \left\{ P \in \mathcal{P}_{\text{fin}} \mid \inf_{x \in \mathbb{R}^n \cap B_R} V_{\text{solu}}(P, x) = 0 \right\}$$

$$= \bigcup_{R \in \mathbb{N}^+} \bigcap_{k \in \mathbb{N}^+} \left\{ P \in \mathcal{P}_{\text{fin}} \mid \inf_{x \in B_R \cap \mathbb{Q}^n} V_{\text{solu}}(P, x) < \frac{1}{k} \right\}$$

For any fixed $x$, $P \mapsto V_{\text{solu}}(P, x)$ is measurable. The infimum over a countable set of measurable functions is measurable. Therefore, the set

$$\left\{ P \in \mathcal{P}_{\text{fin}} \mid \inf_{x \in B_R \cap \mathbb{Q}^n} V_{\text{solu}}(P, x) < \frac{1}{k} \right\}$$

is a measurable subset of $\mathcal{P}_{\text{fin}}$. Since $\mathcal{P}_{\text{sol}}$ is formed by countable unions and intersections of measurable sets, it is measurable. Thus, $\Phi_{\text{attain}}$ is a measurable function. $\square$

**Theorem 8.** *For any Borel regular measure $\mu$ defined on $\mathcal{P}$, $\Phi_{solu} : \mathcal{P} \to \mathbb{R}^n$ is $\mu-$measurable.*

*Proof.* Below, we use measurable to denote $\mu-$measurable for simplicity.

For any $P \in \mathcal{P}_{sol} = \Phi_{attain}^{-1}(1)$, $\Phi_{solu}$ is well-defined. And if suffices to prove that:$(\Phi_{\text{solu}})_i$ is measurable for any $i \in [n]$, i.e. for any $\phi \in \mathbb{R}$, the set: $\{P \in \mathcal{P}_{sol} \mid (\Phi_{solu})_i < \phi\}$ is measurable.

Notice that: the followings are equivalent for $P \in \mathcal{P}_{sol}$:

- $P \in \{P \in \mathcal{P}_{sol} \mid (\Phi_{solu})_i < \phi\}$.

- There exists $x \in \mathbb{R}^n$ with $x_i < \phi$, such that $V_{\text{solu}}(P, x) = 0$ and $V_{\text{solu}}(P, x') > 0, \forall x' \in B_{\|x\|}, x_i' \geq \phi$.

- There exists $R \in \mathbb{Q}_+$, $r \in \mathbb{N}_+$, and $x \in B_R$ with $x_i \leq \phi - 1/r$, such that $V_{\text{solu}}(P, x) = 0$ and $V_{\text{solu}}(P, x') > 0, \forall x' \in B_R, x_i' \geq \phi$.

- There exists $R \in \mathbb{Q}_+$ and $r \in \mathbb{N}_+$, such that for all $r' \in \mathbb{N}_+$, $\exists x \in B_R \cap \mathbb{Q}^n$, $x_i \leq \phi - 1/r$, s.t. $V_{\text{solu}}(P, x) < 1/r'$ and that $\exists r'' \in \mathbb{N}_+$, s.t., $V_{\text{solu}}(P, x') \geq 1/r''$, $\forall x' \in B_R \cap \mathbb{Q}^n$, $x_i' \geq \phi$.

Hence, we can rewrite $\{P \in \mathcal{P}_{sol} \mid (\Phi_{solu})_i < \phi\}$ as:

$$\bigcup_{R \in \mathbb{Q}_+} \bigcup_{r \in \mathbb{N}_+} \left( \begin{array}{l} \left( \bigcap_{r' \in \mathbb{N}_+} \bigcup_{x \in B_R \cap \mathbb{Q}^n, \, x_i \leq \phi - \frac{1}{r}} \left\{ P \in \mathcal{P}_{sol} \mid V_{\text{solu}}(P, x) < \frac{1}{r'} \right\} \right) \\ \cap \left( \bigcup_{r'' \in \mathbb{N}_+} \bigcap_{x' \in B_R \cap \mathbb{Q}^n, \, x_i' \geq \phi} \left\{ P \in \mathcal{P}_{sol} \mid V_{\text{solu}}(P, x') \geq \frac{1}{r''} \right\} \right) \end{array} \right)$$

, which is measurable. $\qquad\square$

### C.4 Relation between SOCP-GNN's separation power and SOCP-WL test's separation power

*Remark* 5. Thanks to the universality of MLPs, it's noteworthy that we can assume all learnable functions in SOCP-GNN are continuous in the following proof without loss of generality, since they are always parametrized by MLPs.

**Theorem 9.** *SOCP-GNN has the same separation power as the SOCP-WL test.*

*Proof.* We only need to show: For any SOCP instance $I$ and $\hat{I}$, encoded by $G, \hat{G}$, respectively, the following holds:

- For graph-level output, two instances can't be separated by $\mathcal{F}_{\text{SOCP}}^{n,m,k_1,\dots,k_m,b}(\mathbb{R})$, i.e.,

$$F(G) = F(\hat{G}), \quad \forall F \in \mathcal{F}_{\text{SOCP}}^{n,m,k_1,\dots,k_m,b}(\mathbb{R})$$

  if and only if the two instances can't be separated by the SOCP-WL test either.

- For node-level output, the two instances can't be separated by $\mathcal{F}_{\text{SOCP}}^{n,m,k_1,\dots,k_m,b}(\mathbb{R}^n)$, i.e.,

$$F(G) = F(\hat{G}), \quad \forall F \in \mathcal{F}_{\text{SOCP}}^{n,m,k_1,\dots,k_m,b}(\mathbb{R}^n)$$

  if and only if the two instances can't be separated by the SOCP-WL test either, with $\mathcal{C}^{T,v_j} = \hat{\mathcal{C}}^{T,\hat{v}_j}$ hold for all $j \in [n]$, i.e. the variables are reindexed according to the SOCP-WL test for both instances.

We first prove that SOCP-GNN can simulate the SOCP WL-test for any fixed SOCP instance. This can be proved by showing that: For any special SOCP-WL test and given graph $G$, there exists an SOCP-GNN that can simulate arbitrary iterations of this test given the same input for $G$ under the one-hot encoding.

Let $\mathcal{F}$ denote the set of all the initial features for all nodes in $G$. Then we select $\hat{g}_i^0$, $i = 1, 2, 3, 4$ to map these features in $\mathcal{F}$ to their one-hot encoding respectively by theorem 3.2 of (Yun et al., 2019). So for any initial round in the SOCP-WL test, there exists an SOCP-GNN that can simulate it.

Assume now, we already have: we get an SOCP-GNN which can simulate the first t rounds of a special SOCP-WL test, so that: $h^{t,n}$ is just the one-hot encoding of $C^{t,n}$ for all nodes n. For the first refinement round for the polyhedron constraint node $s$, we choose $f_1^t$ as an identity mapping, so that: if $\left( C^{t,s}, \sum_{v \in V_1} w_{s,v} \text{HASH}(C^{t,v}) \right)$ and $\left( C^{t,s'}, \sum_{v \in V_1} w_{s',v} \text{HASH}(C^{t,v}) \right)$ are different, then $\left( h^{t,s}, \sum_{v \in V_1} w_{s,v} f_1^t(h^{t,v}) \right)$ and $\left( h^{t,s'}, \sum_{v \in V_1} w_{s',v} f_1^t(h^{t,v}) \right)$ are different. Then, by Theorem 3.2 of (Yun et al., 2019), there exists 4-layered MLP $g_1^t(\cdot)$ with ReLU activation can map these inputs: $\left( h^{t,s}, \sum_{v \in V_1} w_{s,v} f_1^t(h^{t,v}) \right)$ to their corresponding output in SOCP-WL test's one-hot encoding.

Similarly, we can prove that: there exists $\{g_i^t(\cdot)\}$ and $\{f_j^t(\cdot)\}$, such that the corresponding SOCP-GNN can simulate the t+1 round of the SOCP-WL test for $G$. By mathematical induction, for any possible output of $G$ for SOCP-WL test, there exists SOCP-GNNs can output the corresponding one-hot encoding of the stable color, respectively. Consider the two possible outputs:

- Graph-level scalar output. In this case, we set

$$y = f_{\text{out}}\left(I_1, I_2, I_3, I_4\right)$$

- Node-level vector output. In this case, we only consider the output associated with the variable nodes in $V_1$, given by

$$y_i = f_{\text{out}}\left(h^{T,v_i}, I_1, I_2, I_3, I_4\right), i \in [n]$$

where, $I_1 = \sum_{v \in V_1} h^{T,v}$, $I_2 = \sum_{s \in V_2} h^{T,s}$, $I_3 = \sum_{o \in V_3} h^{T,o}$, and $I_4 = \sum_{q \in V_4} h^{T,q}$.

If two instances $\mathcal{I}$ and $\hat{\mathcal{I}}$ can't be separated by any SOCP-GNNs but can be separated by some SOCP-WL test $\mathcal{W}$. By applying the results discussed above to the disjoint union of these two instances' corresponding graphs, we get: $h^{T,\cdot}$ is just one-hot encoding of $C^{T,\cdot}$, respectively. Then we can conclude that their output multisets under $\mathcal{W}$ are the same, which causes a contradiction. Hence, if two instances $\mathcal{I}$ and $\hat{\mathcal{I}}$ can't be separated by any SOCP-GNNs, then they can't be separated by any SOCP-WL test $\mathcal{W}$ as well. Similarly, we have:

For any node $n', n''$ in SOCP instance $\mathcal{I}, \hat{\mathcal{I}}$ respectively, if $h^{t,n'} = \hat{h}^{t,n''}, \forall F \in \mathcal{F}_{\text{SOCP}}^{n,m,,k_1,\ldots,k_m,b}(\mathbb{R})$ holds for any $t \in \mathbb{N}$, then $n', n''$ have the same stable color for any possible SOCP-WL test.

Now, assume two instances $\mathcal{I}$ and $\hat{\mathcal{I}}$ can't be separated by any SOCP-WL test. Now, we show that:

$$C^{t,s} = \hat{C}^{t,s'} \implies h^{t,s} = \hat{h}^{t,s'}, \quad \forall \text{ polyhedron constraint } s, s' \quad \text{and} \quad F \in \mathcal{F}_{\text{SOCP}}^{n,m,,k_1,\ldots,k_m,b}(\mathbb{R}),$$

while a similar result can be derived for other sublayer-iterations using the same method.

When $t = 0$, the conclusion holds obviously.

When $t \geq 1$, assume the conclusion for all nodes holds for $t - 1$, then we have: $\left(C^{t-1,s}, \sum_{v \in V_1} w_{s,v}\text{HASH}(C^{t-1,v})\right) = \left(\hat{C}^{t-1,s'}, \sum_{v \in V_1} \hat{w}_{s',v}\text{HASH}(\hat{C}^{t-1,v})\right)$

Hence, we have:

- $C^{t-1,s} = \hat{C}^{t-1,s'} \Rightarrow h^{t-1,s} = \hat{h}^{t-1,s'}$

- For any color $W_{1j}$ in the collection of colors at the t-1 th iteration for varaible nodes, $\sum_{v \in W_{1j}} w_{s,v} = \sum_{v \in W_{1j}} \hat{w}_{s',v}$. This can be shown by assuming Hash function maps different colors to linearly independent vectors.

- For any color $W_{1j}$, $\sum_{v \in W_{1j}} w_{s,v} f_1^{t-1}(h^{t-1,v}) = \sum_{v \in W_{1j}} \hat{w}_{s',v} f_1^{t-1}(\hat{h}^{t-1,v})$ (By inductive assumption for node $v$ at iteration $t - 1$)

- $\sum_{W_{1j}} \sum_{v \in W_{1j}} w_{s,v} f_1^{t-1}(h^{t-1,v}) = \sum_{W_{1j}} \sum_{v \in W_{1j}} \hat{w}_{s',v} f_1^{t-1}(\hat{h}^{t-1,v})$.

Therefore, $h^{t,s} = \hat{h}^{t,s'}$, which finishes the proof. $\square$

An immediate corollary is:

**Corollary 7.** *For any node $n, n'$ in SOCP instance $\mathcal{I}, \hat{\mathcal{I}}$ respectively, $C^{t,n} = \hat{C}^{t,n'}$ holds for all possible SOCP-WL test and any $t \in \mathbb{N}$ if and only if $h^{t,n} = \hat{h}^{t,n'}, \forall F \in \mathcal{F}_{SOCP}^{n,m,,k_1,\ldots,k_m,b}(\mathbb{R})$ holds for any $t \in \mathbb{N}$.*

By the proof of lemma 3, you can see that:

**Corollary 8.** *For any node $n, n'$ in SOCP instance $\mathcal{I}, \hat{\mathcal{I}}$ respectively, $C^{t,n} = \hat{C}^{t,n'}$ holds for all possible SOCP-WL test and any $t \in \mathbb{N}$ if and only if $h^{t,n} = \hat{h}^{t,n'}, \forall F \in \mathcal{F}_{SOCP}^{n,m,,k_1,\ldots,k_m,b}(\mathbb{R})$ holds for any $t \in \mathbb{N}$. Under such assumption, $(\Phi_{solution}(\mathcal{I}))_n = (\Phi_{solution}(\hat{\mathcal{I}}))_{n'}$ if $n, n'$ are variable nodes.*

C.5 Main theorem's proof

Consider the following theorems, which play an important role in real analysis:

**Lusin theorem:** Let $\mu$ be a Borel regular measure on $\mathbb{R}^n$ and let $f : \mathbb{R}^n \to \mathbb{R}^m$ be $\mu$-measurable. Then for any $\mu$-measurable $X \subset \mathbb{R}^n$ with $\mu(X) < \infty$ and any $\epsilon > 0$, there exists a compact set $E \subset X$ with $\mu(X \setminus E) < \epsilon$, such that $f|_E$ is continuous.

By this fundamental but important theorem, we get $\forall \epsilon > 0$, $\exists$ compact $X \subset \mathcal{G}_{\text{SOCP}}^{n,m,k_1,\ldots,k_m,b}$ with $\mu(\mathcal{G}_{\text{SOCP}}^{n,m,k_1,\ldots,k_m,b} \setminus X) < \epsilon$, such that $\Phi_{target}|_X$ is continuous holds for any $\Phi_{target}$ mentioned in Definition B.1.

Moreover, using similar tricks in (Chen et al., 2022b), we can assume that: $X$ remains the same under the action of the permutation group $S_n$ without loss of generality.

**Generalized Stone-Weierstrass theorem:**[Theorem 22 of (Azizian & Lelarge, 2020)] Let $X$ be a compact topology space and let $\mathbf{G}$ be a finite group that acts continuously on $X$ and $\mathbb{R}^n$. Define the collection of all equivariant continuous functions from $X$ to $\mathbb{R}^n$ as follows:

$$C_E(X, \mathbb{R}^n) = \{F \in C(X, \mathbb{R}^n) : F(g * x) = g * F(x), \forall x \in X, g \in \mathbf{G}\}.$$

Consider any $\mathcal{F} \subset C_E(X, \mathbb{R}^n)$ and any $\Phi \in C_E(X, \mathbb{R}^n)$. Suppose the following conditions hold:

(i) $\mathcal{F}$ is a subalgebra of $C(X, \mathbb{R}^n)$ and $\mathbf{1} \in \mathcal{F}$.

(ii) For any $x, x' \in X$, if $f(x) = f(x')$ holds for any $f \in C(X, \mathbb{R})$ with $f\mathbf{1} \in \mathcal{F}$, then for any $F \in \mathcal{F}$, there exists $g \in \mathbf{G}$ such that $F(x) = g * F(x')$.

(iii) For any $x, x' \in X$, if $F(x) = F(x')$ holds for any $F \in \mathcal{F}$, then $\Phi(x) = \Phi(x')$.

(iv) For any $x \in X$, it holds that $\Phi(x)_j = \Phi(x)_{j'}, \forall (j, j') \in J(x)$, where

$$J(x) = \{\{1, 2, \ldots, n\}^n : F(x)_j = F(x)_{j'}, \forall F \in \mathcal{F}\}.$$

Then for any $\epsilon > 0$, there exists $F \in \mathcal{F}$ such that

$$\sup_{x \in X} \|\Phi(x) - F(x)\| < \epsilon.$$

Now we leverage the theorems listed above to give a proof of the main theorem. And we let the group $\mathbf{G}$ to be permutation group $S_n$. Since our SOCP-GNNs are permutation-equivariant, they are obviously $\mathbf{G} -$ equivariant continuous functions. (The following $a$ refers to 1 or n)

**Property (i):** $\mathcal{F}_{\text{SOCP}}^{n,m,k_1,\ldots,k_m,b}(\mathbb{R}^a)$ is a subalgebra of $C_E(X, \mathbb{R}^a)$ and $\mathbf{1} \in \mathcal{F}_{\text{SOCP}}^{n,m,k_1,\ldots,k_m,b}(\mathbb{R}^a)$

*Proof.* If suffices to prove this by using similar channel expansion techniques mentioned in (Chen et al., 2022b). $\square$

**Property (ii):** For any $x, x' \in X$, if $f(x) = f(x')$ holds for any $f \in C(X, \mathbb{R})$ with $f\mathbf{1} \in \mathcal{F}_{\text{SOCP}}^{n,m,k_1,\ldots,k_m,b}(\mathbb{R}^a)$, then for any $F \in \mathcal{F}_{\text{SOCP}}^{n,m,k_1,\ldots,k_m,b}(\mathbb{R}^a)$, there exists $g \in \mathbf{G}$ such that $F(x) = g * F(x')$.

*Proof.* First notice that: $\mathcal{F}_{\text{SOCP}}^{n,m,,k_1,\ldots,k_m,b}(\mathbb{R}) \in C(X, \mathbb{R})$ with $f\mathbf{1} \in \mathcal{F}_{\text{SOCP}}^{n,m,k_1,\ldots,k_m,b}(\mathbb{R}^a), \forall f \in \mathcal{F}_{\text{SOCP}}^{n,m,k_1,\ldots,k_m,b}(\mathbb{R})$. Then applying theorem 9 and corollary 7 is enough. $\square$

**Property (iii) and (iv):**

- For any $x, x' \in X$, if $F(x) = F(x')$ holds for any $F \in \mathcal{F}_{\text{SOCP}}^{n,m,k_1,\ldots,k_m,b}(\mathbb{R}^a)$, then $\Phi(x) = \Phi(x')$.

- For any $x \in X$, it holds that $\Phi(x)_j = \Phi(x)_{j'}, \forall (j, j') \in J(x)$, where

$$J(x) = \{\{1, 2, \ldots, a\}^2 : F(x)_j = F(x)_{j'}, \forall F \in \mathcal{F}_{\text{SOCP}}^{n,m,k_1,\ldots,k_m,b}(\mathbb{R}^a)\}$$

.

*Proof.* Applying theorems in Appendix C.2, theorem 9, and corollary 8 is enough. □

Applying the generalized Stone-Weierstrass theorem gives us Theorem 2 immediately.

## C.6 EXTENSION TO $p$-ORDER CONE PROGRAMMING

A general $p$-order cone programming can be stated as:

$$
\begin{aligned}
\text{minimize} \quad & e^\top x \\
\text{subject to} \quad & Fx \leq g, \quad l \leq x \leq r, \\
& \|A_i x + b_i\|_p \leq c_i^T x + d_i, \quad i \in [m]
\end{aligned}
\tag{10}
$$

where decision variables are $x \in \mathbb{R}^n$ and the problem parameters are $e \in \mathbb{R}^n$, $A_i \in \mathbb{R}^{k_i \times n}$, $b_i \in \mathbb{R}^{k_i}$, $c_i \in \mathbb{R}^n$, $d_i \in \mathbb{R}$, $F \in \mathbb{R}^{b \times n}$, $g \in \mathbb{R}^b$, $l_j \in (\{-\infty\} \cup \mathbb{R})^n$, and $r \in (\{+\infty\} \cup \mathbb{R})^n$. Here, we only consider the case: $p \in [1, +\infty]$.

Here, we formally define some concepts that are helpful to the extension of $p$-order cone programming.

**Definition C.2.** A function $f : \mathbb{R}^n \to \mathbb{R}$ is said to be **separable** if $f(x)$ can be expressed as a sum $f(x) = \sum_{j=1}^n f_j(x_j)$, where each function $f_j$ only depends on the scalar $x_j$. (This definition is stricter than traditional "block separable".)

**Definition C.3.** A function $f : \mathbb{R}^n \to \mathbb{R}$ is said to be equivalent (w.r.t. permutation group $S_n$) if for any rearranging $\{\sigma(1), \sigma(2), \cdots, \sigma(n)\}$ of $\{1, 2, \cdots, n\}$ and any $x \in \mathbb{R}^n$, $f(x_1, x_2, \cdots, x_n) = f(x_{\sigma(1)}, x_{\sigma(2)}, \cdots, x_{\sigma(n)})$

For $p \in [1, +\infty)$, we have: $\|x\|_p^p = \sum_{i=1}^n |x_i|^p$, which is separable and equivalent according to Definition C.2 and C.3.

**Situation 1:Use p as a fixed parameter:** We don't need to make any modifications to our architectures. As for the proof of the universality, we just need to change $\|.\|_2$ to $\|.\|_p$ for $p \geq 1$ in our proof of lemma 3 and other theorems in Appendix C, since our proof only uses the convexity, permutation-invariant property, continuous property, and separability of the $l_2$ norm, which holds for the $l_p$ norm as well when $p \in [1, +\infty)$. As for $p = +\infty$, lemma 3 can be directly validated by noticing that:

$$
\begin{aligned}
|(\hat{b}_r + \hat{A}_r \hat{x})_{j_1}| &= |\frac{1}{|W_{4m}|} \sum_{C^{T,u} = W_{4m}} \sum_{j \in W_{3l}, j \in u} \frac{|W_{4m}|}{|W_{3l}|} (A_u x + b_u)_j| \\
&\leq \frac{1}{|W_{4m}|} \sum_{C^{T,u} = W_{4m}} \sum_{j \in W_{3l}, j \in u} \frac{|W_{4m}|}{|W_{3l}|} |(A_u x + b_u)_j| \\
&\leq \frac{1}{|W_{4m}|} \sum_{C^{T,u} = W_{4m}} \sum_{j \in W_{3l}, j \in u} \frac{|W_{4m}|}{|W_{3l}|} (c_u x + d_u) = \frac{1}{|W_{4m}|} \sum_{C^{T,u} = W_{4m}} (c_u x + d_u) \\
&\leq \hat{c}_r^T \hat{x} + \hat{d}_r
\end{aligned}
$$

, where the notions follow the settings in lemma 3. Since the above equation holds for all $j_1$, we can see that: lemma 3 still holds. Since $\| \cdot \|_\infty$ is continuous, the measurability holds as well.

**Situation 2:Use p as a continuous parameter:** Here, we need a little modification on our architectures and proofs, while we only consider $p \in [1, +\infty)$ since $\|x\|_p$ is continuous in $p$ when $p \in [1, +\infty)$.

For the graph representation, we only need to augment our variable features from $(e_i, l_i, r_i)$ to $(e_i, l_i, r_i, p)$. And the GNN and related WL test don't need any modification. As for the proof, it suffices to notice that:

- To prove lemma 3, we just need to observe that: If two instances $\mathcal{I}$ and $\hat{\mathcal{I}}$ can't be distinguished by the WL test, then their corresponding p must be the same. Then what remains is just the situation one's proof mentioned above. Other related results hold as well, like the equivalence of the WL test and GNN in separation power.

- As for the measurability, we just need to repeat what we do in Appendix C.3 while taking p as a parameter in the new parameter space.

**Situation 3:Mix order conic programming:** Here, similar to situation 2, we need to augment features for minor constraint nodes. For the constraint $\|A_i x + b_i\|_p \leq c_i^T x + d_i$, we reset the minor conic node j's feature to be $((b_i)_j, p)$ [8]. Then we can prove Lemma 3 by noticing that: two major conic constraints have the same color if and only if their corresponding p are the same. The measurability holds as well, similar to situation 1.

# D   PROOF OF THEOREM 3

## D.1   VC-DIMENSION BASED APPROACHES FOR BINARY CLASSIFICATION

**Definition D.1** (Growth function). For binary classification, the growth function of a hypothesis class $\mathcal{A}$ over the domain $\mathcal{X}$ is defined as:

$$\tau_{\mathcal{A}}(n) = \sup_{\mathbf{x} \in \mathcal{X}^n} |\mathcal{A} \circ \mathbf{x}|$$

, where $\mathcal{A} \circ \mathbf{x} = \{(a(x_1), a(x_2), \cdots, a(x_n)) \in \{0,1\}^n \mid a \in \mathcal{A}\}$

**Definition D.2** (VC-dimension). The Vapnik-Chervonenkis dimension, or VC-dimension, of $\mathcal{A}$ is the largest integer n such that: $\tau_{\mathcal{A}}(n) = 2^n$. If $\tau_{\mathcal{A}}(n) = 2^n$ for all possible n, then $\mathcal{A}$'s VC dimension is $+\infty$.

Below, we use $\mathcal{VC}(\mathcal{A})$ to denote the VC dimension of the hypothesis class $\mathcal{A}$ for simplicity.

**Definition D.3** (WL equivalence relation). We define the equivalence relation in $\mathcal{G}_{\text{SOCP}}^{n,m,k_1,\ldots,k_m,b}$ as: two graphs $G_1$ and $G_2$ are equivalent if and only if they can't be distinguished by all possible SOCP-WL tests. Given a space of graphs $\mathcal{G} \subset \mathcal{G}_{\text{SOCP}}^{n,m,k_1,\ldots,k_m,b}$, let $\mathcal{G}/_{WL}$ denotes collections of the equivalence class of $\mathcal{G}$ under such equivalence relation.

**Theorem 10** (VC dimension of $\mathcal{F}_{\text{SOCP}}^{n,m,k_1,\ldots,k_m,b}(\mathbb{R})$ over $\mathcal{G}$). *For hypothesis class* $\mathcal{F}_{SOCP}^{n,m,k_1,\ldots,k_m,b}(\mathbb{R})$, $\mathcal{VC}(\mathcal{F}_{SOCP}^{n,m,k_1,\ldots,k_m,b}(\mathbb{R})) = |\mathcal{G}/_{WL}|$. *Here,* $\mathcal{F}_{SOCP}^{n,m,k_1,\ldots,k_m,b}(\mathbb{R})$ *do binary classification in the following way: any function* $f \in \mathcal{F}_{SOCP}^{n,m,k_1,\ldots,k_m,b}(\mathbb{R})$ *maps x to 1 if* $f(x) \geq 0.5$. *Otherwise, it maps x to 0.*

*Proof.* First, we show that: $\mathcal{VC}(\mathcal{F}_{\text{SOCP}}^{n,m,k_1,\ldots,k_m,b}(\mathbb{R})) \leq |\mathcal{G}/_{WL}|$. We prove by contradiction, if $\mathcal{VC}(\mathcal{F}_{\text{SOCP}}^{n,m,k_1,\ldots,k_m,b}(\mathbb{R})) > |\mathcal{G}/_{WL}|$, then there exists two graphs $G_1, G_2 \in \mathcal{G}$ which can't be distinguished by SOCP-WL test but have different output under some $f \in \mathcal{F}_{\text{SOCP}}^{n,m,k_1,\ldots,k_m,b}(\mathbb{R})$. This contradicts with theorem 9.

Now, we show that: $\mathcal{VC}(\mathcal{F}_{\text{SOCP}}^{n,m,k_1,\ldots,k_m,b}(\mathbb{R})) \geq |\mathcal{G}/_{WL}|$. Let $u = |\mathcal{G}/_{WL}|$. Take representative elements $G_1, G_2, \cdots, G_u$ of $\mathcal{G}/_{WL}$ respectively. Consider $u < +\infty$ first, from theorem 9, we know that there exists a SOCP-GNN that can simulate the SOCP-WL test for $\cup_{i=1}^u G_i$. Hence, $G_i$'s output $(I_1, I_2, I_3, I_4)$ must be different respectively under this GNN after enough iterations. By theorem 3.1 of (Yun et al., 2019), we can output all possible results for $G_i$ respectively by using a 3-layer ReLU-like FNN as the output layer. Hence, $\mathcal{VC}(\mathcal{F}_{\text{SOCP}}^{n,m,k_1,\ldots,k_m,b}(\mathbb{R})) \geq |\mathcal{G}/_{WL}|$ when $u < +\infty$, which indicates $\mathcal{VC}(\mathcal{F}_{\text{SOCP}}^{n,m,k_1,\ldots,k_m,b}(\mathbb{R})) = |\mathcal{G}/_{WL}|$. In case where $u = +\infty$, we have: $\mathcal{VC}(\mathcal{F}_{\text{SOCP}}^{n,m,k_1,\ldots,k_m,b}(\mathbb{R})) = +\infty$ as well. Similar to the proof when $u < +\infty$, we can see that: $\forall n \in \mathbb{N}, \tau_{\mathcal{F}_{\text{SOCP}}^{n,m,k_1,\ldots,k_m,b}(\mathbb{R})}(n) = 2^n$, which finishes the proof.

$\square$

## D.2   PSEUDO-DIMENSION BASED APPROACHES FOR REAL-VALUED SCALAR PREDICTION

**Definition D.4.** Let $\mathcal{G}$ be a family of real-valued functions $g : \mathcal{X} \to \mathbb{R}$. We say that a set of points $S = \{x_1, x_2, \ldots, x_N\} \subset \mathcal{X}$ is **pseudo-shattered** by $\mathcal{G}$ if there exists a vector of thresholds (or targets)

---

[8]Here, we use $p = -1$ to encode $+\infty$ into feature.

$\mathbf{z} = (z_1, z_2, \ldots, z_N) \in \mathbb{R}^N$ such that for any binary vector $\mathbf{b} = (b_1, b_2, \ldots, b_N) \in \{+1, -1\}^N$, there is a function $g \in \mathcal{G}$ satisfying:

$$\forall i \in \{1, \ldots, N\}, \quad \text{sign}(g(x_i) - z_i) = b_i$$

The **pseudo-dimension** of $\mathcal{G}$, denoted as $\text{Pdim}(\mathcal{G})$, is the size of the largest set that can be pseudo-shattered by $\mathcal{G}$. If arbitrarily large sets can be pseudo-shattered, the pseudo-dimension is infinite.

A common result in learning theory shows that: For any family of functions $\mathcal{H}$ mapping from a domain $\mathcal{Y}$ to a bounded interval $[0, H]$, the following generalization guarantee holds: For any $\delta \in (0, 1)$, with probability at least $1 - \delta$ over the draw of a set $S \sim \mathcal{D}^N$ of $N$ samples drawn i.i.d. from an arbitrary distribution $\mathcal{D}$ over $\mathcal{Y}$, the following bound holds uniformly for all $h \in \mathcal{H}$:

$$\left| \frac{1}{N} \sum_{y \in S} h(y) - \mathbb{E}_{y \sim \mathcal{D}}[h(y)] \right| \leq O\left( H \sqrt{\frac{\text{Pdim}(\mathcal{H}) + \ln\left(\frac{1}{\delta}\right)}{N}} \right)$$

Now, we begin to give the pseudo-dimension of SOCP-GNNs for real-valued scalar prediction (e.g. predicting the objective value).

**Theorem 11** (pseudo-dimension of $\mathcal{F}_{\text{SOCP}}^{n,m,k_1,\ldots,k_m,b}(\mathbb{R})$ over $\mathcal{G}$). *For hypothesis class* $\mathcal{F}_{SOCP}^{n,m,k_1,\ldots,k_m,b}(\mathbb{R})$, $Pdim(\mathcal{F}_{SOCP}^{n,m,k_1,\ldots,k_m,b}(\mathbb{R})) = |\mathcal{G}/_{WL}|$.

*Proof.* Similar to the proof above, we prove this theorem from two sides.

First, we show that: $\text{Pdim}(\mathcal{F}_{\text{SOCP}}^{n,m,k_1,\ldots,k_m,b}(\mathbb{R})) \leq |\mathcal{G}/_{WL}|$. Otherwise, if $\text{Pdim}(\mathcal{F}_{\text{SOCP}}^{n,m,k_1,\ldots,k_m,b}(\mathbb{R})) > |\mathcal{G}/_{WL}|$, then there exists two graphs $G_1, G_2 \in \mathcal{G}$ which can't be distinguished by SOCP-WL test but have different output under some $f \in \mathcal{F}_{\text{SOCP}}^{n,m,k_1,\ldots,k_m,b}(\mathbb{R})$. This contradicts with theorem 9. (Assume there exist $z_1, z_2$ such that: for any binary vector $\mathbf{b} = (b_1, b_2) \in \{+1, -1\}^2$, there is a function $g \in \mathcal{F}_{\text{SOCP}}^{n,m,k_1,\ldots,k_m,b}(\mathbb{R})$ satisfying:

$$\forall i \in \{1, 2\}, \quad \text{sign}(g(G_i) - z_i) = b_i$$

Without loss of generality, we assume $z_1 \geq z_2$. Then there is a function $g \in \mathcal{F}_{\text{SOCP}}^{n,m,k_1,\ldots,k_m,b}(\mathbb{R})$ such that $g(G_1) > z_1 \geq z_2 > g(G_2)$.)

Now, we show that: $\text{Pdim}(\mathcal{F}_{\text{SOCP}}^{n,m,k_1,\ldots,k_m,b}(\mathbb{R})) \geq |\mathcal{G}/_{WL}|$. Let $u = |\mathcal{G}/_{WL}|$. Take representative elements $G_1, G_2, \cdots, G_u$ of $\mathcal{G}/_{WL}$ respectively. Consider $u < +\infty$ first, from theorem 9, we know that there exists a SOCP-GNN that can simulate the SOCP-WL test for $\cup_{i=1}^u G_i$. Hence, $G_i$'s output $(I_1, I_2, I_3, I_4)$ must be different respectively under this GNN after enough iterations. By theorem 3.1 of (Yun et al., 2019), we can output all possible results for $G_i$ respectively by using a 3-layer ReLU-like FNN as the output layer. Hence, $\text{Pdim}(\mathcal{F}_{\text{SOCP}}^{n,m,k_1,\ldots,k_m,b}(\mathbb{R})) \geq |\mathcal{G}/_{WL}|$ when $u < +\infty$, which indicates $\text{Pdim}(\mathcal{F}_{\text{SOCP}}^{n,m,k_1,\ldots,k_m,b}(\mathbb{R})) = |\mathcal{G}/_{WL}|$. In case where $u = +\infty$, we have: $\text{Pdim}(\mathcal{F}_{\text{SOCP}}^{n,m,k_1,\ldots,k_m,b}(\mathbb{R})) = +\infty$ as well, since any finite set composed of the representative elements of $\mathcal{G}/_{WL}$ can be pseudo-shattered. $\square$

### D.3 RADEMACHER COMPLEXITY BASED APPROACHES

Before we start, let's recall some basic concepts first.

**Definition D.5.** For a SOCP problem $X \in \mathcal{G}_{\text{SOCP}}^{n,m,k_1,\ldots,k_m,b}$, we define its size $N$ to be its parameter $(e, \{A_i\}_{i=1}^m, \{b_i\}_{i=1}^m, \{c_i\}_{i=1}^m, \{d_i\}_{i=1}^m, F, g, l, r)$'s dimension when equipped with product topology over these Euclidean spaces for predicting *boundedness*, *solution attainability*, *optimal value*, *optimal solution*. And we define its problem size $N$ to be the dimension of its constraints' parameter $(\{A_i\}_{i=1}^m, \{b_i\}_{i=1}^m, \{c_i\}_{i=1}^m, \{d_i\}_{i=1}^m, F, g, l, r)$'s dimension when equipped with product topology over these Euclidean space for predicting *feasibility*.

We make the task-specialized definitions above since predicting feasibility has nothing to do with the objective function, while other tasks are all closely related to the objective function. Here, we focus on the following set of problems and hypotheses:

**Problem class:** The problem class $\mathcal{X}$ we are solving satisfies the following properties:

- The problem size is $N$.
- Its valid parameters lie in the bounded ball $\mathcal{B}_{r_i} = \{x \mid \|x\|_2 \le r_i\}$.

Note that we can always transform a problem class whose valid parameters are bounded into a new problem class whose valid parameters lie in the bounded ball by scaling. Moreover, real-world large-scale problems always have sparsity. So it's reasonable to assume that the $\ell_2$ norm of the valid parameters of the problems is bounded.

**Hypothesis class:** Here, we consider a subclass of SOCP-GNNs $\mathcal{A}_{L,N}$ such that each $a \in \mathcal{A}_{L,N}$ satisfying the following property:

- **Graph-level Output Lipschitz property:** $a$ is $L$-Lipshitz w.r.t. the input problem parameters in $\mathcal{B}_{r_i}$
- **Node-level Output Lipschitz property:** For each variable $i \in [n]$, we have: $a$'s $i$-th output is $L$-Lipshitz w.r.t. the input problem parameters in $\mathcal{B}_{r_i}$ as well.

We remark that this kind of assumption is widely accepted by researchers in the sample complexity/generalization ability of graph neural networks (Pellizzoni et al., 2024; Garg et al., 2020; Tang & Liu, 2023). Now, we introduce some concepts which is helpful to our theory.

**Definition D.6** (Rademacher Complexity of a set). Given a set $A \subseteq \mathbb{R}^m$, the Rademacher complexity of $A$ is defined as follows:

$$Rad(A) := \frac{1}{m}\mathbb{E}_\sigma \left[ \sup_{a \in A} \sum_{i=1}^m \sigma_i a_i \right]$$

where $\sigma_1, \sigma_2, \ldots, \sigma_m$ are independent random variables drawn from the Rademacher distribution, i.e.,

$$\Pr(\sigma_i = +1) = \Pr(\sigma_i = -1) = 1/2 \quad \text{for } i = 1, 2, \ldots, m,$$

and $a = (a_1, \ldots, a_m) \in A$. The expectation $\mathbb{E}_\sigma$ is taken over the random variables $\sigma = (\sigma_1, \ldots, \sigma_m)$.

**Definition D.7** (pseudo metric space). A pseudometric space is an ordered pair $(X, d)$ where $X$ is a set and $d$ is a function $d : X \times X \to \mathbb{R}$, called a pseudometric, satisfying the following conditions for all $x, y, z \in X$:

1. $d(x, y) \ge 0$ (Non-negativity)

2. $d(x, x) = 0$ (Identity of self)

3. $d(x, y) = d(y, x)$ (Symmetry)

4. $d(x, z) \le d(x, y) + d(y, z)$ (Triangle inequality)

Unlike a metric space, a pseudometric space allows $d(x, y) = 0$ for distinct points $x \ne y$.

**Definition D.8** (Covering number). Let $(X, d)$ be a pseudometric space and let $S$ be a subset of $X$. For a given $\epsilon > 0$, an $\epsilon$-**covering** for $S$ is a set of points $\{x_1, \ldots, x_N\} \subseteq X$ such that for every point $s \in S$, there exists some $x_i$ in the set for which $d(s, x_i) \le \epsilon$. The $\epsilon$-covering number of $S$, denoted by $Cov(S, d, \epsilon)$, is the minimum size $N$ of $\epsilon$-coverings for $S$. Formally:

$$Cov(S, d, \epsilon) = \min \{|P| \ : \ P \subseteq X \text{ is an } \epsilon\text{-covering}\}$$

**Definition D.9** (Packing number). Let $(X, d)$ be a pseudometric space. For a given $\epsilon > 0$, an $\epsilon$-**packing** of $X$ is a subset $P \subseteq X$ in which the distance between any two distinct points is strictly greater than $\epsilon$, i.e., $d(x, y) > \epsilon$ for all $x, y \in P$ with $x \ne y$. The $\epsilon$-packing number of $X$, denoted by $Pack(X, d, \epsilon)$, is the maximum possible cardinality of such a set. Formally, it is defined as the supremum over the sizes of all possible $\epsilon$-packings:

$$Pack(X, d, \epsilon) = \max \{|P| \ : \ P \subseteq X \text{ is an } \epsilon\text{-packing}\}.$$

Now, let's define the pseudo metric over $\mathcal{A}_{L,N}$ for both graph-level output and node-level output.
**Pseudo metric for graph-level scalar output:** Given a training set $\mathbf{x} = \{x_1, \cdots, x_m\}$, we define $\|a\|_{p,\mathbf{x}} = |\frac{\sum_{i=1}^m |a(x_i)|^p}{m}|^{\frac{1}{p}}$ for $a \in \mathcal{A}_{L,N}$ with output dimension 1 as a pseudo norm. And define

$\|a - b\|_{p,\mathbf{x}} = |\frac{\sum_{i=1}^{m} |a(x_i) - b(x_i)|^p}{m}|^{\frac{1}{p}}$ as the pseudo metric on $\mathcal{A}_{L,N}$ with scalar output, denoted by $\|.\|_{p,\mathbf{x}}$.

**Pseudo metric for node-level vector output:** Given a training set $\mathbf{x} = \{x_1, \cdots, x_m\}$, we define $\|a\|_{p,\mathbf{x}} = |\frac{\sum_{i=1}^{m} \sum_{j=1}^{n} |(a(x_i))_j|^p}{mn}|^{\frac{1}{p}}$ for $a \in \mathcal{A}_{L,N}$ with output dimension n as a pseudo norm. And define $\|a - b\|_{p,\mathbf{x}} = |\frac{\sum_{i=1}^{m} \sum_{j=1}^{n} |(a(x_i))_j - (b(x_i))_j|^p}{mn}|^{\frac{1}{p}}$ as the pseudo metric on $\mathcal{A}_{L,N}$ with vector output, denoted by $\|.\|_{p,\mathbf{x}}$.

Without loss of generality, we assume our loss function is Lipshitz continuous with coefficient $q$.

**Lemma 4** (Contraction lemma, (Shalev-Shwartz & Ben-David, 2014)'s Lemma 26.9). *For each $i \in [m]$, let $\phi_i : \mathbb{R} \to \mathbb{R}$ be a $\rho$-Lipschitz function, namely for all $\alpha, \beta \in \mathbb{R}$ we have $|\phi_i(\alpha) - \phi_i(\beta)| \leq \rho|\alpha - \beta|$. For $\mathbf{a} \in \mathbb{R}^m$ let $\boldsymbol{\phi}(\mathbf{a})$ denote the vector $(\phi_1(a_1), \ldots, \phi_m(a_m))$. Let $\boldsymbol{\phi} \circ A = \{\boldsymbol{\phi}(\mathbf{a}) : \mathbf{a} \in A\}$. Then,*

$$Rad(\boldsymbol{\phi} \circ A) \leq \rho Rad(A).$$

For the node-level scalar output, we have:

**Lemma 5** (Contraction lemma for node-level output, (Maurer, 2016)). *Let $\mathcal{X}$ be any set, $(x_1, \ldots, x_n) \in \mathcal{X}^n$, let $F$ be a class of functions $f : \mathcal{X} \to \ell_2$ and let $h_i : \ell_2 \to \mathbb{R}$ have Lipschitz norm $L$. Then*

$$\mathbb{E} \sup_{f \in F} \sum_i \sigma_i h_i(f(x_i)) \leq \sqrt{2} L \mathbb{E} \sup_{f \in F} \sum_{i,k} \sigma_{ik} f_k(x_i),$$

*where $\sigma_{ik}$ is an independent doubly indexed Rademacher sequence and $f_k(x_i)$ is the k-th component of $f(x_i)$. And We use $\ell_2$ to denote the Hilbert space of square summable sequences of real numbers.*

By setting the after M-th coordinate of $x \in R^M$ to 0, we can see that any finite dimensional Euclidean space is a special class of $\ell_2$ space.

Now, Let $z_i$ denote $(x_i, y_i)$, where $x_i$ is the i-th socp instance and $y_i \in \mathbb{R}$ is the label of $x_i$. We denote the loss function as $\phi(z) = \phi(a(x), y)$, which is $q$-Lipschitz w.r.t $a(x)$ for all possible y. Let $\phi(\mathbf{z})$ denote the vector $(\phi(z_1), \ldots, \phi(z_m))$. Let $\phi \circ A = \{\phi(\mathbf{z}) : \mathbf{z} \in A\}$. Then, we have:

$$Rad(\phi \circ \{(z_1, ..., z_m) : z_i = (a(x_i), y_i), \forall i \in [m] \text{ for } a \in \mathcal{A}_{L,N}\}) \leq q \, Rad(\mathcal{A}_{L,N} \circ \{(x_1, ..., x_m)\})$$

Meanwhile, Let $z_i$ denote $(x_i, y_i)$, where $x_i$ is the i-th socp instance and $y_i \in \mathbb{R}^n$ is the label of $x_i$ and we denote the loss function as $\phi(z) = \phi(a(x), y)$, which is $q$-Lipshitz w.r.t $a(x) \in \mathbb{R}^n$ for all possible y. by lemma 5, we get:

$$Rad(\phi \circ \{(z_1, ..., z_m) : z_i = (a(x_i), y_i), \forall i \in [m] \text{ for } a \in \mathcal{A}_{L,N}\}) \leq \sqrt{2} qn \, Rad(\mathcal{A}_{L,N} \circ \{(x_1, ..., x_m)\})$$

where $Rad(\mathcal{A}_{L,N} \circ \{(x_1, ..., x_m)\}) = \frac{1}{m} E_\sigma \left[\sup_{a \in \mathcal{A}_{L,N}} \sum_{i=1}^{m} \sigma_i a(x_i)\right]$ for graph-level scalar output and $Rad(\mathcal{A}_{L,N} \circ \{(x_1, ..., x_m)\}) = \frac{1}{mn} E_\sigma \left[\sup_{a \in \mathcal{A}_{L,N}} \sum_{i=1}^{m} \sum_{j=1}^{n} \sigma_{ij}(a(x_i))_j\right]$ for node-level vector output.

So, we only need to focus on $Rad(\mathcal{A}_{L,N} \circ \{(x_1, ..., x_m)\})$ for fixed training sample $(x_1, \cdots, x_m)$. That's just $\hat{\mathcal{R}}_S(\mathcal{A}_{L,N})$ we defined following.

**Lemma 6** (Dudley Entropy Integral for scalar output, chapter 5.3.3 of (Wainwright, 2019)). *Let $\mathcal{A}_{L,N}$ be the hypothesis class of SOCP-GNNs with scalar output as defined above. Let $S = \{x_1, \ldots, x_m\}$ be a fixed set of $m$ SOCP problem instances. The empirical Rademacher complexity of $\mathcal{A}_{L,N}$ on $S$ is defined as*

$$\hat{\mathcal{R}}_S(\mathcal{A}_{L,N}) = \frac{1}{m} E_\sigma \left[\sup_{a \in \mathcal{A}_{L,N}} \sum_{i=1}^{m} \sigma_i a(x_i)\right]$$

*where $\sigma_i$ are independent Rademacher random variables. Let $||\cdot||_{2,S}$ be the empirical $L_2$ pseudo metric on $\mathcal{A}_{L,N}$, given by $||a||_{2,S} = \sqrt{\frac{1}{m} \sum_{i=1}^{m} a(x_i)^2}$ for $a \in \mathcal{A}_{L,N}$. Assume that for some $C_S > 0$,*

*we have $\sup_{a \in \mathcal{A}_{L,N}} ||a||_{2,S} \leq C_S$. Then,*

$$\hat{\mathcal{R}}_S(\mathcal{A}_{L,N}) \leq \inf_{\epsilon \in [0, C_S/2]} \left\{ 4\epsilon + \frac{12}{\sqrt{m}} \int_{\epsilon}^{C_S/2} \sqrt{\log Cov(\mathcal{A}_{L,N}, || \cdot ||_{2,S}, v)} \, dv \right\}$$

*where $Cov(\mathcal{A}_{L,N}, d, \epsilon)$ is the $\epsilon$-covering number of the set $\mathcal{A}_{L,N}$ with respect to the pseudometric d.*

*Proof.* We start by constructing a sequence of coverings for the hypothesis class $\mathcal{A}_{L,N}$ at progressively finer scales. Define $\epsilon_j = C_S/2^j$ for $j = 1, 2, \ldots, K$. For each $j$, let $\mathcal{A}_j$ be a minimal $\epsilon_j$-cover of $\mathcal{A}_{L,N}$ with respect to the $|| \cdot ||_{2,S}$ pseudometric, so that its size is $|\mathcal{A}_j| = Cov(\mathcal{A}_{L,N}, || \cdot ||_{2,S}, \epsilon_j)$.

For any function $a \in \mathcal{A}_{L,N}$, we can define a sequence of approximations $\pi_j(a) \in \mathcal{A}_j$ such that $|| a - \pi_j(a) ||_{2,S} \leq \epsilon_j$ and set $\pi_0(a) = 0$. For any integer $K \geq 0$, any function $a \in \mathcal{A}_{L,N}$ can be decomposed into:

$$a = (a - \pi_K(a)) + \sum_{j=1}^{K} (\pi_j(a) - \pi_{j-1}(a)).$$

By the sub-additivity of the supremum, the empirical Rademacher complexity $\hat{\mathcal{R}}_S(\mathcal{A}_{L,N}) = \frac{1}{m} \mathbb{E}_\sigma \left[ \sup_{a \in \mathcal{A}_{L,N}} \sum_{i=1}^{m} \sigma_i a(x_i) \right]$ can be bounded by:

$$\hat{\mathcal{R}}_S(\mathcal{A}_{L,N}) \leq \frac{1}{m} \mathbb{E}_\sigma \left[ \sup_{a \in \mathcal{A}_{L,N}} \sum_{i=1}^{m} \sigma_i(a(x_i) - \pi_K(a)(x_i)) \right] + \sum_{j=1}^{K} \frac{1}{m} \mathbb{E}_\sigma \left[ \sup_{a \in \mathcal{A}_{L,N}} \sum_{i=1}^{m} \sigma_i(\pi_j(a)(x_i) - \pi_{j-1}(a)(x_i)) \right].$$

The first term, representing the residual error, can be bounded using the Cauchy-Schwarz inequality. For any $a \in \mathcal{A}_{L,N}$, we have $\sum_{i=1}^{m} \sigma_i(a(x_i) - \pi_K(a)(x_i)) \leq m||a - \pi_K(a)||_{1,S} \leq m||a - \pi_K(a)||_{2,S} \leq m\epsilon_K$. Thus, the residual term is bounded by $\epsilon_K$.

For the second terms, we consider each term for $j = 0, \ldots, K$. Let $d_j(a) = \pi_j(a) - \pi_{j-1}(a)$. This difference function belongs to the set $D_j = \{c - c' \mid c \in \mathcal{A}_j, c' \in \mathcal{A}_{j-1}\}$, whose size is at most $|\mathcal{A}_j||\mathcal{A}_{j-1}|$. By the triangle inequality, the norm of any such difference is bounded by:

$$||d_j(a)||_{2,S} = ||\pi_j(a) - \pi_{j-1}(a)||_{2,S} \leq ||\pi_j(a) - a||_{2,S} + ||a - \pi_{j-1}(a)||_{2,S} \leq \epsilon_j + \epsilon_{j-1} = 3\epsilon_j.$$

We apply Massart's Lemma to the Rademacher complexity of the finite set $D_j$:

$$\mathbb{E}_\sigma \left[ \sup_{a \in \mathcal{A}_{L,N}} \sum_{i=1}^{m} \sigma_i d_j(a)(x_i) \right] \leq \sup_{d \in D_j} \sqrt{\sum_{i=1}^{m} d(x_i)^2} \cdot \sqrt{2 \log |D_j|}$$

$$\leq \sup_{a \in \mathcal{A}_{L,N}} \left( \sqrt{m}||d_j(a)||_{2,S} \right) \sqrt{2 \log(|\mathcal{A}_j||\mathcal{A}_{j-1}|)}.$$

Since the covering number is non-increasing with scale, $|\mathcal{A}_{j-1}| \leq |\mathcal{A}_j|$, which gives $\log |D_j| \leq 2 \log |\mathcal{A}_j|$. Therefore, the bound on the $j$-th term of the Rademacher complexity is:

$$\frac{1}{m} \mathbb{E}_\sigma \left[ \sup_{a \in \mathcal{A}_{L,N}} \sum_{i=1}^{m} \sigma_i d_j(a)(x_i) \right] \leq \frac{1}{\sqrt{m}} (3\epsilon_j) \sqrt{4 \log |\mathcal{A}_j|} = \frac{6\epsilon_j}{\sqrt{m}} \sqrt{\log Cov(\mathcal{A}_{L,N}, || \cdot ||_{2,S}, \epsilon_j)}.$$

Summing these bounds from $j = 1$ to $K$, and noting that $\epsilon_j = 2(\epsilon_j - \epsilon_{j+1})$, we obtain a sum that approximates an integral:

$$\sum_{j=1}^{K} \frac{6\epsilon_j}{\sqrt{m}} \sqrt{\log Cov(\mathcal{A}_{L,N}, || \cdot ||_{2,S}, \epsilon_j)} = \frac{12}{\sqrt{m}} \sum_{j=1}^{K} (\epsilon_j - \epsilon_{j+1}) \sqrt{\log Cov(\mathcal{A}_{L,N}, || \cdot ||_{2,S}, \epsilon_j)}$$

$$\leq \frac{12}{\sqrt{m}} \int_{\epsilon_{K+1}}^{\epsilon_1} \sqrt{\log Cov(\mathcal{A}_{L,N}, || \cdot ||_{2,S}, v)} \, dv.$$

Combining all parts, we have for any chosen refinement level $K$:

$$\hat{\mathcal{R}}_S(\mathcal{A}_{L,N}) \leq 2\epsilon_{K+1} + \frac{12}{\sqrt{m}} \int_{\epsilon_{K+1}}^{C_S/2} \sqrt{\log Cov(\mathcal{A}_{L,N}, || \cdot ||_{2,S}, v)} \, dv.$$

Since this holds for any $K$, we can replace the cutoff scale $\epsilon_{K+1}$ with an arbitrary $\epsilon \in [\frac{\epsilon_{K+1}}{2}, \epsilon_{K+1}]$. Taking the infimum over all such $\epsilon$ yields the tightest bound:

$$\hat{\mathcal{R}}_S(\mathcal{A}_{L,N}) \leq \inf_{\epsilon \in [0, C_S/2]} \left\{ 4\epsilon + \frac{12}{\sqrt{m}} \int_\epsilon^{C_S/2} \sqrt{\log \text{Cov}(\mathcal{A}_{L,N}, ||\cdot||_{2,S}, v)} \, dv \right\}.$$

$\square$

**Lemma 7** (Dudley Entropy Integral for vector output). *Let $\mathcal{A}_{L,N}$ be the hypothesis class of SOCP-GNNs as defined above. Let $S = \{x_1, \ldots, x_m\}$ be a fixed set of $m$ SOCP problem instances. The empirical Rademacher complexity of $\mathcal{A}_{L,N}$ with output dimension $n$ on $S$ is defined as*

$$\hat{\mathcal{R}}_S(\mathcal{A}_{L,N}) = \frac{1}{mn} E_\sigma \left[ \sup_{a \in \mathcal{A}_{L,N}} \sum_{i=1}^m \sum_{j=1}^n \sigma_{ij}(a(x_i))_j \right]$$

*where $\sigma_{ij}$ are independent Rademacher random variables. Let $||\cdot||_{2,S}$ be the empirical $L_2$ pseudometric on $\mathcal{A}_{L,N}$, given by $||a||_{2,S} = \sqrt{\frac{1}{mn} \sum_{i=1}^m \sum_{j=1}^n (a(x_i))_j^2}$ for $a \in \mathcal{A}_{L,N}$. Assume that for some $C_S > 0$, we have $\sup_{a \in \mathcal{A}_{L,N}} ||a||_{2,S} \leq C_S$. Then,*

$$\hat{\mathcal{R}}_S(\mathcal{A}_{L,N}) \leq \inf_{\epsilon \in [0, C_S/2]} \left\{ 4\epsilon + \frac{12}{\sqrt{mn}} \int_\epsilon^{C_S/2} \sqrt{\log \text{Cov}(\mathcal{A}_{L,N}, ||\cdot||_{2,S}, v)} \, dv \right\}$$

*where $\text{Cov}(\mathcal{A}_{L,N}, d, \epsilon)$ is the $\epsilon$-covering number of the set $\mathcal{A}_{L,N}$ with respect to the pseudometric $d$.*

*Proof.* Similarly, we define $\epsilon_k = C_S/2^k$ for $k = 1, 2, \ldots, K$. For each $k$, let $\mathcal{A}_k$ be a minimal $\epsilon_k$-cover of $\mathcal{A}_{L,N}$ with respect to the $||\cdot||_{2,S}$ pseudometric, so that its size is $|\mathcal{A}_k| = \text{Cov}(\mathcal{A}_{L,N}, ||\cdot||_{2,S}, \epsilon_k)$.

For any function $a \in \mathcal{A}_{L,N}$, we can define a sequence of approximations $\pi_k(a) \in \mathcal{A}_k$ such that $||a - \pi_k(a)||_{2,S} \leq \epsilon_k$ and set $\pi_0(a) = 0$. For any integer $K \geq 0$, any function $a \in \mathcal{A}_{L,N}$ can be decomposed into:

$$a = (a - \pi_K(a)) + \sum_{k=1}^K (\pi_k(a) - \pi_{k-1}(a)).$$

By the sub-additivity of the supremum, the empirical Rademacher complexity $\hat{\mathcal{R}}_S(\mathcal{A}_{L,N}) = \frac{1}{mn} \mathbb{E}_\sigma \left[ \sup_{a \in \mathcal{A}_{L,N}} \sum_{i=1}^m \sum_{j=1}^n \sigma_{ij}(a(x_i))_j \right]$ can be bounded by:

$$\hat{\mathcal{R}}_S(\mathcal{A}_{L,N}) \leq \frac{1}{mn} \mathbb{E}_\sigma \left[ \sup_{a \in \mathcal{A}_{L,N}} \sum_{i=1}^m \sum_{j=1}^n \sigma_{ij}(a(x_i) - \pi_K(a)(x_i))_j \right]$$

$$+ \sum_{k=1}^K \frac{1}{mn} \mathbb{E}_\sigma \left[ \sup_{a \in \mathcal{A}_{L,N}} \sum_{i=1}^m \sum_{j=1}^n \sigma_{ij}(\pi_k(a)(x_i) - \pi_{k-1}(a)(x_i))_j \right].$$

For any $a \in \mathcal{A}_{L,N}$, we have $\sum_{i=1}^m \sum_{j=1}^n \sigma_{ij}(a(x_i) - \pi_K(a)(x_i))_j \leq mn||a - \pi_K(a)||_{1,S} \leq mn||a - \pi_K(a)||_{2,S} \leq mn\epsilon_K$. Thus, the residual term is bounded by $\epsilon_K$.

For the second terms, we consider each term for $k = 0, \ldots, K$. Let $d_k(a) = \pi_k(a) - \pi_{k-1}(a)$. This difference function belongs to the set $D_k = \{c - c' \mid c \in \mathcal{A}_k, c' \in \mathcal{A}_{k-1}\}$, whose size is at most $|\mathcal{A}_k||\mathcal{A}_{k-1}|$. By the triangle inequality, the norm of any such difference is bounded by:

$$||d_k(a)||_{2,S} = ||\pi_k(a) - \pi_{k-1}(a)||_{2,S} \leq ||\pi_k(a) - a||_{2,S} + ||a - \pi_{k-1}(a)||_{2,S} \leq \epsilon_k + \epsilon_{k-1} = 3\epsilon_k.$$

We apply Massart's Lemma to the Rademacher complexity of the finite set $D_j$:

$$\mathbb{E}_\sigma \left[ \sup_{a \in \mathcal{A}_{L,N}} \sum_{i=1}^m \sum_{j=1}^n \sigma_{ij}(d_k(a)(x_i))_j \right] \leq \sup_{d \in D_k} \sqrt{\sum_{i=1}^m \sum_{j=1}^n (d(x_i))_j^2} \cdot \sqrt{2 \log |D_k|}$$

$$\leq \sup_{a \in \mathcal{A}_{L,N}} \left( \sqrt{mn} \|d_k(a)\|_{2,S} \right) \sqrt{2 \log(|\mathcal{A}_k||\mathcal{A}_{k-1}|)}.$$

Since the covering number is non-increasing with scale, $|\mathcal{A}_{k-1}| \leq |\mathcal{A}_k|$, which gives $\log |D_k| \leq 2 \log |\mathcal{A}_k|$. Therefore, the bound on the $k$-th term of the Rademacher complexity is:

$$\frac{1}{mn} \mathbb{E}_\sigma \left[ \sup_{a \in \mathcal{A}_{L,N}} \sum_{i=1}^m \sum_{j=1}^n \sigma_{ij} (d_k(a)(x_i))_j \right] \leq \frac{1}{\sqrt{mn}} (3\epsilon_k) \sqrt{4 \log |\mathcal{A}_k|} = \frac{6\epsilon_k}{\sqrt{mn}} \sqrt{\log \mathrm{Cov}(\mathcal{A}_{L,N}, || \cdot ||_{2,S}, \epsilon_k)}.$$

Summing these bounds from $k = 1$ to $K$, and noting that $\epsilon_k = 2(\epsilon_k - \epsilon_{k+1})$, we obtain a sum that approximates an integral:

$$\sum_{k=1}^K \frac{6\epsilon_k}{\sqrt{mn}} \sqrt{\log \mathrm{Cov}(\mathcal{A}_{L,N}, || \cdot ||_{2,S}, \epsilon_k)} = \frac{12}{\sqrt{mn}} \sum_{k=1}^K (\epsilon_k - \epsilon_{k+1}) \sqrt{\log \mathrm{Cov}(\mathcal{A}_{L,N}, || \cdot ||_{2,S}, \epsilon_k)}$$

$$\leq \frac{12}{\sqrt{mn}} \int_{\epsilon_{K+1}}^{\epsilon_1} \sqrt{\log \mathrm{Cov}(\mathcal{A}_{L,N}, || \cdot ||_{2,S}, v)} \, dv.$$

Combining all parts, we have for any chosen refinement level $K$:

$$\hat{\mathcal{R}}_S(\mathcal{A}_{L,N}) \leq 2\epsilon_{K+1} + \frac{12}{\sqrt{m}} \int_{\epsilon_{K+1}}^{C_S/2} \sqrt{\log \mathrm{Cov}(\mathcal{A}_{L,N}, || \cdot ||_{2,S}, v)} \, dv.$$

Since this holds for any $K$, we can replace the cutoff scale $\epsilon_{K+1}$ with an arbitrary $\epsilon \in [\frac{\epsilon_{K+1}}{2}, \epsilon_{K+1}]$. Taking the infimum over all such $\epsilon$ yields the tightest bound:

$$\hat{\mathcal{R}}_S(\mathcal{A}_{L,N}) \leq \inf_{\epsilon \in [0, C_S/2]} \left\{ 4\epsilon + \frac{12}{\sqrt{m}} \int_\epsilon^{C_S/2} \sqrt{\log \mathrm{Cov}(\mathcal{A}_{L,N}, || \cdot ||_{2,S}, v)} \, dv \right\}.$$

$\square$

By the above lemma, it suffices to study the bound of $\mathrm{Cov}(\mathcal{A}_{L,N}, || \cdot ||_{2,S}, v)$ now. And we will consider two different situations here, i.e., the output dimension is 1 and n, respectively.

**Lemma 8** (Estimation for covering number bounds in SOCP-parameter space, (Pellizzoni et al., 2024) Lemma 2). *Let $\mathcal{A}_{L,N} \subset \{f : S \to \mathbb{R}\}$ be the hypothesis class of SOCP-GNNs whose output dimension is 1, where $S = \{x_1, \ldots, x_m\}$ be a fixed set of $m$ problem instances. We assume the function outputs lie in the interval $[-r, r]$. For any $\epsilon > 0$, the logarithm of the $\epsilon$-covering number of this class with respect to the empirical $L_2$ pseudometric, $|| \cdot ||_{2,S}$, is bounded by:*

$$\log Cov(\mathcal{A}_{L,N}, || \cdot ||_{2,S}, \epsilon) \leq Cov(S, || \cdot ||_2, \frac{\epsilon}{2L}) \log \left( \frac{2r}{\epsilon} + 2 \right).$$

*Proof.* Take $S_s$ to be the minimal $\frac{\epsilon}{2L}$-covering of S. Consider function class $O = \{f : S_s \to \{\frac{\epsilon}{2} + k\epsilon : k = -([\frac{r}{\epsilon}] + 1), ..., 0, [\frac{r}{\epsilon}]\}\}$. For each $x \in S$, let $\pi(x) \in S_s$ be one of the closet point of $x$ in $S_s$ satisfying: $\|\pi(x) - x\|_2 \leq \frac{\epsilon}{2L}$. Ler $\mathcal{O} = \{f : S \to \mathbb{R} : f(x) = \hat{f}(\pi(x)) \text{ for some } \hat{f} \in O\}$. Then $|\mathcal{O}| = |O| = (2[\frac{r}{\epsilon}] + 2)^{\mathrm{Cov}(S, \|\cdot\|_2, \frac{\epsilon}{2L})}$. So, it suffices to prove that: $\mathcal{O}$ is a $\epsilon$-covering of $\mathcal{A}_{L,N}$ under $|| \cdot ||_{2,S}$.

For any $f \in \mathcal{A}_{L,N}$, take $g \in \mathcal{O}$ such that $|g(x) - f(x)| \leq \frac{\epsilon}{2}$ for all $x \in S_s$. Then we have:

$$\|f - g\|_{2,S}^2 = \frac{1}{m} (\sum_{x \in S} |f(x) - g(x)|^2)$$

$$\leq \frac{1}{m} (\sum_{x \in S} [|f(x) - f(\pi(x))| + |f(\pi(x)) - g(\pi(x))| + |g(\pi(x)) - g(x)|]^2)$$

$$\leq \frac{1}{m} (\sum_{x \in S} [\frac{\epsilon}{2L} \cdot L + \frac{\epsilon}{2} + 0]^2) = \epsilon^2$$

Taking square root of both sides, we get the proof. $\square$

**Lemma 9.** *Let $\mathcal{A}_{L,N} \subset \{f : S \to \mathbb{R}^n\}$ be the hypothesis class of SOCP-GNNs whose output dimension is n, where $S = \{x_1, \ldots, x_m\}$ be a fixed set of m problem instances. We assume that each component of the function output lies in the interval $[-r, r]$. For any $\epsilon > 0$, the logarithm of the $\epsilon$-covering number of this class with respect to the empirical $L_2$ pseudometric, $|| \cdot ||_{2,S}$, is bounded by:*

$$\log Cov(\mathcal{A}_{L,N}, || \cdot ||_{2,S}, \epsilon) \leq nCov(S, || \cdot ||_2, \frac{\epsilon}{2L}) \log\left(\frac{2r}{\epsilon} + 2\right).$$

*Proof.* Take $S_s$ to be the minimal $\frac{\epsilon}{2L}$-covering of S. Consider function class $O = \{f : S_s \to \{\frac{\epsilon}{2} + k\epsilon : k = -([\frac{r}{\epsilon}] + 1), ..., 0, [\frac{r}{\epsilon}]\}^n\}$. For each $x \in S$, let $\pi(x) \in S_s$ be one of the closest point of $x$ in $S_s$ satisfying: $\|\pi(x) - x\|_2 \leq \frac{\epsilon}{2L}$. Let $\mathcal{O} = \{f : S \to \mathbb{R}^n : f(x) = \hat{f}(\pi(x)) \text{ for some } \hat{f} \in O\}$. Then $|\mathcal{O}| = |O| = (2[\frac{r}{\epsilon}] + 2)^{nCov(S,\|\cdot\|_2, \frac{\epsilon}{2L})}$. So, it suffices to prove that: $\mathcal{O}$ is a $\epsilon$-covering of $\mathcal{A}_{L,N}$ under $|| \cdot ||_{2,S}$.

For any $f \in \mathcal{A}_{L,N}$, take $g \in \mathcal{O}$ such that $|g(x)_i - f(x)_i| \leq \frac{\epsilon}{2}$ for all $x \in S_s$ and $i \in [n]$. Then we have:

$$\|f - g\|_{2,S}^2 = \frac{1}{mn}(\sum_{x \in S} \sum_{i=1}^{n} |f(x)_i - g(x)_i|^2)$$

$$\leq \frac{1}{mn}(\sum_{x \in S} \sum_{i=1}^{n} [|f(x)_i - f(\pi(x))_i| + |f(\pi(x))_i - g(\pi(x))_i| + |g(\pi(x))_i - g(x)_i|]^2)$$

$$\leq \frac{1}{mn}(\sum_{x \in S} \sum_{i=1}^{n} [\frac{\epsilon}{2L} \cdot L + \frac{\epsilon}{2} + 0]^2) = \epsilon^2$$

Taking square root of both sides, we get the proof. $\square$

So, what remains is to discuss the covering number bounds of $S$ in its valid parameter space $\mathcal{B}_{r_i}$

**Lemma 10** (Estimation for covering number bounds in socp-parameter space). *For any $\epsilon > 0$ and the uniform distribution $\mathbb{P}$ over the valid parameter space $\mathcal{B}_{r_i}$, we have:*

$$\mathbb{E}_{S \sim \mathbb{P}^m}(Cov(S, || \cdot ||_2, \epsilon)) \leq (\frac{2r_i + \epsilon}{\epsilon})^N (1 - (1 - \min((\frac{\epsilon}{r_i})^N, 1))^m)$$

,

*Proof.* Notice that:

$$Cov(S, || \cdot ||_2, \epsilon) \leq Pack(S, || \cdot ||_2, \epsilon) \leq Pack(\mathcal{B}_{r_i}, || \cdot ||_2, \epsilon)$$

Here, the first inequality holds since any maximum $\epsilon$-packing is a $\epsilon$-covering as well. Let $P$ be the maximum $\epsilon$-packing of $\mathcal{B}_{r_i}$ and let $P = \{x_1, \cdots, x_{|P|}\}$. Then, we have: $\cup_{i=1}^{|P|} B(x_i, \frac{\epsilon}{2}) \subset \mathcal{B}(0, r_i + \frac{\epsilon}{2})$, where $B(x_i, \frac{\epsilon}{2}) = \{x \in \mathbb{R}^N : \|x_i - x\|_2 \leq \frac{\epsilon}{2}\}$ are mutually disjoint. Hence, if let $\alpha_N$ denote the volume of the bounded ball in $\mathbb{R}^N$, we get:$|P| \leq \frac{\alpha_N}{\alpha_N}(\frac{2r_i + \epsilon}{\epsilon})^N$.

Consider the collection of balls $\mathcal{B} = \{B(x_i, \epsilon) : i \in [|P|]\}$. For any ball $b \in \mathcal{B}$ and m socp instances $s_1, \cdots, s_m$ which are sampled i.i.d. from the uniform distribution on $\mathcal{B}_{r_i}$, we have:

$$\mathbb{P}(s_i \notin b, \forall i \in [m]) \geq (1 - \min((\frac{\epsilon}{r_i})^N, 1))^m$$

. For any possible realization of $S = \{s_1, \cdots, s_m\}$, we take all balls $b \in \mathcal{B}$, which contain some points in $S$ to form a new set of balls, denoted by $\mathcal{C}$. Then $\mathcal{C}$'s center forms a $\epsilon$-covering of $S$.

(Here, notice that: $\mathcal{B}_{r_i} \subset \cup_{b \in \mathcal{B}} b$). Let $I_1, ..., I_{|P|}$ denote the indicator variables for the event: $\exists\, y \in S, s.t.\ y \in B(x_i, \epsilon)$ respectively. Then, we have:

$$\mathbb{E}_{S \sim \mathbb{P}^m}(\text{Cov}(S, \|\cdot\|_2, \epsilon)) \leq \mathbb{E}_{S \sim \mathbb{P}^m}(\sum_{i=1}^{|P|} I_i) \leq |P|(1 - (1 - \min((\frac{\epsilon}{r_i})^N, 1))^m)$$

$$\leq (\frac{2r_i + \epsilon}{\epsilon})^N (1 - (1 - \min((\frac{\epsilon}{r_i})^N, 1))^m)$$

$\square$

Before we get our result finally, an important lemma is needed.

**Theorem 12** (Theorem 26.5 of (Shalev-Shwartz & Ben-David, 2014); Algorithmic Foundations of Learning). *Assume that for all $z$ and $h \in \mathcal{H}$ we have that $|\ell(h, z)| \leq c$. Then,*

1. *With probability of at least $1 - \delta$, for all $h \in \mathcal{H}$,*

$$L_D(h) - L_S(h) \leq 2 \mathbb{E}_{S' \sim D^m} Rad(\ell \circ \mathcal{H} \circ S') + c\sqrt{\frac{2\ln(2/\delta)}{m}}.$$

*In particular, this holds for $h = ERM_\mathcal{H}(S)$.*

2. *With probability of at least $1 - \delta$, for all $h \in \mathcal{H}$,*

$$L_D(h) - L_S(h) \leq 2Rad(\ell \circ \mathcal{H} \circ S) + 4c\sqrt{\frac{2\ln(4/\delta)}{m}}.$$

*In particular, this holds for $h = ERM_\mathcal{H}(S)$.*

3. *For any $h^*$, with probability of at least $1 - \delta$,*

$$L_D(ERM_\mathcal{H}(S)) - L_D(h^*) \leq 2Rad(\ell \circ \mathcal{H} \circ S) + 5c\sqrt{\frac{2\ln(8/\delta)}{m}}.$$

4. *For any $h^*$, with probability of at least $1 - \delta$,*

$$L_D(ERM_\mathcal{H}(S)) - L_D(h^*) \leq 4\mathbb{E}_{S' \sim D^m}[Rad(\ell \circ \mathcal{H} \circ S')] + 2c\sqrt{\frac{2\log(1/\delta)}{m}}.$$

*Here, $z$ denotes the sample point, $S$ and $S'$ denotes the training set of size $m$, $\mathcal{H}$ denotes the hypothesis class, $\ell$ denotes the loss function, $h^*$ denote the hypothesis in $\mathcal{H}$ with the smallest generalization error and $ERM_\mathcal{H}(S)$ denotes the hypothesis in $\mathcal{H}$ with the smallest empirical error on $S$.*

Before we go further, we need a further lemma to apply Tonelli's theorem in the following proof.

**Lemma 11.** *The function $\Phi(S, v) = Cov(S, \|\cdot\|, v)$ is measurable w.r.t. the standard lebesgue measure, where $S \in (\mathcal{B}_{r_i})^m, v \in \mathbb{R}^+$. Here, $\mathbb{R}^+$ denotes all non-negative real numbers.*

*Proof.* Here, if suffices to prove that: for any $n \in \mathbb{N}$, the set $\{(S, v) : \Phi(S, v) \leq n\}$ is measurable. Let $S = (s_1, \cdots, s_m), X = (x_1, \cdots, x_n)$, we define $h(S, X) = \max_{i \in [m]} \min_{j \in [n]} \|s_i - x_j\|_2$, which is continuous in $S, X$. By Berge's theorem of maximum, $g(S) = \inf_{x_1, ..., x_n \in \mathcal{B}_{r_i}} h(S, X)$ is continuous in $S$. Hence $g(S) - v$ is continuous. Notice that:

$$\{(S, v) : \Phi(S, v) \leq n\} = \{(S, v) : g(S) - v \leq 0\}$$

Hence, this set is measurable, which indicates that $Cov(S, \|\cdot\|, v)$ is measurable w.r.t. $(S, v)$. $\square$

It's time to get our integral sample complexity result.

**Theorem 13** (estimation risk for graph level output). *For graph-level prediction tasks, let $\mathcal{A}_{L,N}$ be the hypothesis class of SOCP-GNNs defined above for a graph-level prediction task on the set $\mathcal{X}$ of SOCP instances whose valid parameters lie in $\mathcal{B}_{r_i}$, with outputs in $[-r, r]$ and loss functions that are $q$-Lipshitz and bounded by $p$. Then, for any training set $S$ of $m$ samples which are i.i.d. sampled from the uniform distribution $D$ over $\mathcal{X}$, let $h^*$ denote the hypothesis in $\mathcal{A}_{L,N}$ with the smallest generalization error and $ERM_{\mathcal{A}_{L,N}}(S)$ denotes the hypothesis in $\mathcal{A}_{L,N}$ with the smallest empirical error on $S$, we have: with probability of at least $1 - \delta$,*

$$L_D(ERM_{\mathcal{A}_{L,N}}(S)) - L_D(h^*) \le 4\mathbb{E}_{S' \sim D^m}[Rad(\ell \circ \mathcal{A}_{L,N} \circ S')] + 2p\sqrt{\frac{2\log(1/\delta)}{m}}$$

$$\le 4q \inf_{\epsilon \in [0, r/2]}[4\epsilon + \frac{12}{\sqrt{m}} \int_\epsilon^{r/2} \sqrt{(\frac{4Lr_i + v}{v})^N (1 - (1 - \min((\frac{v}{2Lr_i})^N, 1))^m) \log\left(\frac{2r}{v} + 2\right)} \, dv]$$

$$+ 2p\sqrt{\frac{2\log(1/\delta)}{m}}$$

*Proof.* By theorem 12 above, it suffices to notice that:

$\mathbb{E}_{S' \sim D^m}[Rad(\ell \circ \mathcal{A}_{L,N} \circ S')]$

$\le q\mathbb{E}_{S' \sim D^m}[Rad(\mathcal{A}_{L,N} \circ S')]$(lemma 4)

$\le q\mathbb{E}_{S' \sim D^m}[\inf_{\epsilon \in [0, r/2]} \left\{ 4\epsilon + \frac{12}{\sqrt{m}} \int_\epsilon^{r/2} \sqrt{\log Cov(\mathcal{A}_{L,N}, || \cdot ||_{2,S'}, v)} \, dv \right\}]$(lemma 6)

$\le q \inf_{\epsilon \in [0, r/2]} \mathbb{E}_{S' \sim D^m}[\left\{ 4\epsilon + \frac{12}{\sqrt{m}} \int_\epsilon^{r/2} \sqrt{\log Cov(\mathcal{A}_{L,N}, || \cdot ||_{2,S'}, v)} \, dv \right\}]$

$\le q \inf_{\epsilon \in [0, r/2]}[4\epsilon + \frac{12}{\sqrt{m}} \mathbb{E}_{S' \sim D^m} \int_\epsilon^{r/2} [\sqrt{\log Cov(\mathcal{A}_{L,N}, || \cdot ||_{2,S'}, v)}] \, dv]$

$\le q \inf_{\epsilon \in [0, r/2]}[4\epsilon + \frac{12}{\sqrt{m}} \mathbb{E}_{S' \sim D^m} \int_\epsilon^{r/2} [\sqrt{Cov(S', || \cdot ||_2, \frac{v}{2L}) \log\left(\frac{2r}{v} + 2\right)}] \, dv]$(lemma 8)

$\le q \inf_{\epsilon \in [0, r/2]}[4\epsilon + \frac{12}{\sqrt{m}} \int_\epsilon^{r/2} \mathbb{E}_{S' \sim D^m}[\sqrt{Cov(S', || \cdot ||_2, \frac{v}{2L}) \log\left(\frac{2r}{v} + 2\right)}] \, dv]$(Tonelli's Theorem)

$\le q \inf_{\epsilon \in [0, r/2]}[4\epsilon + \frac{12}{\sqrt{m}} \int_\epsilon^{r/2} \sqrt{\mathbb{E}_{S' \sim D^m}[Cov(S', || \cdot ||_2, \frac{v}{2L})] \log\left(\frac{2r}{v} + 2\right)} \, dv]$(Jensen Inequality)

$\le q \inf_{\epsilon \in [0, r/2]}[4\epsilon + \frac{12}{\sqrt{m}} \int_\epsilon^{r/2} \sqrt{(\frac{4Lr_i + v}{v})^N (1 - (1 - \min((\frac{v}{2Lr_i})^N, 1))^m) \log\left(\frac{2r}{v} + 2\right)} \, dv]$(lemma 10)

Here, $\sqrt{Cov(S', || \cdot ||_2, \frac{v}{2L}) \log\left(\frac{2r}{v} + 2\right)}$ is measurable since it's the square root of the multiplication of two finite-valued positive measurable functions. Since it's positive, we can exchange the order of integration by Tonelli's theorem. $\square$

**Theorem 14** (estimation risk for node level output). *For node-level prediction tasks, let $\mathcal{A}_{L,N}$ be the hypothesis class of SOCP-GNNs defined above for a node-level prediction task on the set $\mathcal{X}$ of SOCP instances whose valid parameters lie in $\mathcal{B}_{r_i}$, with outputs in $[-r, r]^n$ and loss functions that are $q$-Lipshitz and bounded by $p$. Then, for any training set $S$ of $m$ samples which are i.i.d. sampled from the uniform distribution $D$ over $\mathcal{X}$, let $h^*$ denote the hypothesis in $\mathcal{A}_{L,N}$ with the smallest generalization error and $ERM_{\mathcal{A}_{L,N}}(S)$ denotes the hypothesis in $\mathcal{A}_{L,N}$ with the smallest empirical error on $S$, we have: with probability of at least $1 - \delta$,*

$$L_D(ERM_{\mathcal{A}_{L,N}}(S)) - L_D(h^*) \le 4\mathbb{E}_{S' \sim D^m}[Rad(\ell \circ \mathcal{A}_{L,N} \circ S')] + 2p\sqrt{\frac{2\log(1/\delta)}{m}}$$

$$\le 4\sqrt{2}nq \inf_{\epsilon \in [0, r/2]}[4\epsilon + \frac{12}{\sqrt{m}} \int_\epsilon^{r/2} \sqrt{(\frac{4Lr_i + v}{v})^N (1 - (1 - \min((\frac{v}{2Lr_i})^N, 1))^m) \log\left(\frac{2r}{v} + 2\right)} \, dv]$$

$$+ 2p\sqrt{\frac{2\log(1/\delta)}{m}}$$

*Proof.* By theorem 12 above, it suffices to notice that:

$$\mathbb{E}_{S'\sim D^m}[Rad(\ell \circ \mathcal{A}_{L,N} \circ S')]$$

$$\leq \sqrt{2}nq\mathbb{E}_{S'\sim D^m}[Rad(\mathcal{A}_{L,N} \circ S')](\text{lemma } 5)$$

$$\leq \sqrt{2}nq\mathbb{E}_{S'\sim D^m}\Big[\inf_{\epsilon\in[0,r/2]}\Big\{4\epsilon + \frac{12}{\sqrt{mn}}\int_\epsilon^{r/2}\sqrt{\log\text{Cov}(\mathcal{A}_{L,N},||\cdot||_{2,S'},v)}\,dv\Big\}\Big](\text{lemma } 7)$$

$$\leq \sqrt{2}nq\inf_{\epsilon\in[0,r/2]}\mathbb{E}_{S'\sim D^m}\Big[\Big\{4\epsilon + \frac{12}{\sqrt{mn}}\int_\epsilon^{r/2}\sqrt{\log\text{Cov}(\mathcal{A}_{L,N},||\cdot||_{2,S'},v)}\,dv\Big\}\Big]$$

$$\leq \sqrt{2}nq\inf_{\epsilon\in[0,r/2]}\Big[4\epsilon + \frac{12}{\sqrt{mn}}\mathbb{E}_{S'\sim D^m}\int_\epsilon^{r/2}\Big[\sqrt{\log\text{Cov}(\mathcal{A}_{L,N},||\cdot||_{2,S'},v)}\Big]\,dv\Big]$$

$$\leq \sqrt{2}nq\inf_{\epsilon\in[0,r/2]}\Big[4\epsilon + \frac{12}{\sqrt{mn}}\mathbb{E}_{S'\sim D^m}\int_\epsilon^{r/2}\Big[\sqrt{n\text{Cov}(S',||\cdot||_2,\frac{v}{2L})\log\Big(\frac{2r}{v}+2\Big)}\Big]\,dv\Big](\text{lemma } 9)$$

$$\leq \sqrt{2}nq\inf_{\epsilon\in[0,r/2]}\Big[4\epsilon + \frac{12}{\sqrt{mn}}\int_\epsilon^{r/2}\mathbb{E}_{S'\sim D^m}\Big[\sqrt{n\text{Cov}(S',||\cdot||_2,\frac{v}{2L})\log\Big(\frac{2r}{v}+2\Big)}\Big]\,dv\Big](\text{Tonelli's Theorem})$$

$$\leq \sqrt{2}nq\inf_{\epsilon\in[0,r/2]}\Big[4\epsilon + \frac{12}{\sqrt{mn}}\int_\epsilon^{r/2}\sqrt{n\mathbb{E}_{S'\sim D^m}[\text{Cov}(S',||\cdot||_2,\frac{v}{2L})]\log\Big(\frac{2r}{v}+2\Big)}\,dv\Big](\text{Jensen Inequality})$$

$$\leq \sqrt{2}nq\inf_{\epsilon\in[0,r/2]}\Big[4\epsilon + \frac{12}{\sqrt{m}}\int_\epsilon^{r/2}\sqrt{(\frac{4Lr_\mathrm{i}+v}{v})^N(1-(1-\min((\frac{v}{2Lr_\mathrm{i}})^N,1))^m)\log\Big(\frac{2r}{v}+2\Big)}\,dv\Big](\text{lemma } 10)$$

$\square$

# E SOCP-BASED FORMULATION FOR OPF

- **Decision Variables**

$$w_i \in \mathbb{R}_+ \quad \text{voltage magnitude squared at bus } i \tag{11}$$

$$c_{ij}, s_{ij} \in \mathbb{R} \quad \text{cosine and sine terms for line } (i,j) \tag{12}$$

$$P_{g,i}, Q_{g,i} \in \mathbb{R} \quad \text{real and reactive power generation at bus } i \tag{13}$$

$$P_{ij}, Q_{ij} \in \mathbb{R} \quad \text{real and reactive power flows on line } (i,j) \tag{14}$$

where:

$$c_{ij} = w_i w_j \cos(\theta_i - \theta_j) \tag{15}$$

$$s_{ij} = w_i w_j \sin(\theta_i - \theta_j) \tag{16}$$

- **Objective Function** $\min \sum_{i \in \mathcal{G}} c_i \cdot P_{g,i}$

- **Second-Order Cone Constraints** For each line $(i,j) \in \mathcal{E}$, the rotated second-order cone constraint:

$$c_{ij}^2 + s_{ij}^2 \leq w_i w_j \tag{17}$$

can be directly reformulated as the following standard second-order cone constraint:

$$\left\| \begin{bmatrix} 2c_{ij} \\ 2s_{ij} \\ w_i - w_j \end{bmatrix} \right\|_2 \leq w_i + w_j \tag{18}$$

- **Power Flow Equations** Real power flow from bus $i$ to bus $j$:

$$P_{ij} = g_{ij} w_i - g_{ij} c_{ij} - b_{ij} s_{ij} \tag{19}$$

Reactive power flow from bus $i$ to bus $j$:

$$Q_{ij} = -b_{ij} w_i + b_{ij} c_{ij} - g_{ij} s_{ij} \tag{20}$$

where $g_{ij}$ and $b_{ij}$ are the conductance and susceptance of line $(i,j)$.

- **Nodal Power Balance** For each bus $i \in \mathcal{N}$:

$$P_{g,i} - P_{d,i} = \sum_{j \in \mathcal{N}(i)} P_{ij} + g_{ii} w_i \tag{21}$$

$$Q_{g,i} - Q_{d,i} = \sum_{j \in \mathcal{N}(i)} Q_{ij} - b_{ii} w_i \tag{22}$$

where $\mathcal{N}(i)$ is the set of buses connected to bus $i$, and $g_{ii}$, $b_{ii}$ are shunt elements.

- **Voltage Magnitude Limits**

$$(V_i^{\min})^2 \leq w_i \leq (V_i^{\max})^2 \quad \forall i \in \mathcal{N} \tag{23}$$

- **Generation Limits**

$$P_{g,i}^{\min} \leq P_{g,i} \leq P_{g,i}^{\max} \quad \forall i \in \mathcal{G} \tag{24}$$

$$Q_{g,i}^{\min} \leq Q_{g,i} \leq Q_{g,i}^{\max} \quad \forall i \in \mathcal{G} \tag{25}$$

- **Line Flow Limits**

$$\|(P_{ij}, Q_{ij})\|_2 \leq S_{ij}^{\max} \quad \forall (i,j) \in \mathcal{E} \tag{26}$$

## F  EXPERIMENT SETTINGS AND SUPPLEMENTARY RESULTS

### F.1  DATA GENERATION

#### F.1.1  GENERATION OF FEASIBLE SOCP INSTANCES

Following the SOCP generating scheme in CVXPY, we use the following steps to generate feasible and random SOCP instances, which admit at least an optimal solution.

(I) : Generate a secret point $x_s \in \mathbb{R}^n$ by sampling from a standard normal distribution, i.e., $x_s \sim \mathcal{N}(\mathbf{0}, \mathbf{I})$. Then generate the objective coefficient $e \sim \mathcal{N}(0, 0.25\mathbf{I})$.

(II) : Impose lower bounds and upper bounds on variables $l \leq x_s \leq r$ for the problem. Here $l = x_s - |\Delta_1| - 0.1, r = x_s + |\Delta_2| + 0.1$, where $\Delta_i$ are sampled i.i.d. from $\mathcal{N}(0, 0.25\mathbf{I})$ and $|\cdot|$ denotes component-wise absolute value.

(III) : Generate $F \in \mathbb{R}^{b \times n}$, whose nonzero entries are sampled i.i.d. from $\mathcal{N}(0, 0.01)$ and components are nonzero with probability 0.5. The vector $g$ is subsequently sampled by $g = Fx_s + |\Delta_3| + 0.1$, where $\Delta_3 \sim \mathcal{N}(0, 0.25\mathbf{I})$.

(IV) : For each conic constraint, randomly sample the cone dimension (the number of rows of $A_i, b_i$) in $[1, 7]$ with equal probability. Then, generate $A_i, c_i, b_i$, whose nonzero entries are sampled i.i.d. from $\mathcal{N}(0, 0.0025)$. Each component of the coefficient matrix $A_i, c_i$ is nonzero with probability 0.5. Then, generate $d_i = \|A_i x_s + b_i\|_2 - c_i^\top x_s + \epsilon$, where $\epsilon \sim \mathcal{U}(0.5, 1)$.

Step (II) ensures that the generated SOCP instances always have an optimal solution. Furthermore, the coefficients are intentionally sampled from distributions with different variances, introducing varying numerical scales to create more challenging test instances.

#### F.1.2  GENERATION OF (POSSIBLE) INFEASIBLE SOCP INSTANCES

We use the following steps to generate a (possible) infeasible SOCP instance with pre-determined probability $h \in [0, 1]$.

(I) : Sample a feasible SOCP instance by methods in Appendix F.1.1.

(II) : Execute step III-IV with probability $h$ and execute step V-VI with probability $1 - h$.

(III) : Sample a random integer $p$ in $[3, 20]$ and a scale coefficient $a \sim \mathcal{U}(0, 1)$. Then repeat step IV for $p$ times

(IV) : Randomly choose a type of constraint to break with equal probability. If the polyhedral constraint is chosen, we randomly choose one component of $g$ with equal probability, denoted by $g_i$, and then replace $g_i$ by $(Fx_s)_i - \delta - 3$. If the conic constraint is chosen, we randomly choose one with equal probability and then replace its corresponding $d_i$ by $\|A_i x_s + b_i\|_2 - c_i^\top x_s - \delta - 3$. Here $\delta \sim \mathcal{U}(0, a)$.

(V) : Sample a random integer $p$ in $[3, 20]$ and a scale coefficient $a \sim \mathcal{U}(0, 1)$. Then repeat step VI for $p$ times

(VI) : Randomly choose a type of constraint to enhance with equal probability. If the polyhedral constraint is chosen, we randomly choose one component of $g$ with equal probability, denoted by $g_i$, and then replace $g_i$ by $g_i + \delta$. If the conic constraint is chosen, we randomly choose one with equal probability and then replace its corresponding $d_i$ by $d_i + \delta$. Here $\delta \sim \mathcal{U}(0, a)$.

#### F.1.3  GENERATION OF OPF-SOCP INSTANCES

We use the following steps to generate feasible SOCP instances that admit an optimal solution (Here, initial problem settings are the same as Appendix E):

(I) : Read the reference problem in the IEEE test systems (Babaeinejadsarookolaee et al., 2019) which has the pre-determined number of buses.

(II) : Randomly remove one branch from the grid topology while making sure the resulting graph is still connected.

(III) : Apply multiplicative perturbations to the base real and reactive power demands, $P_{d,i}$ and $Q_{d,i}$, at each bus $i$, and to the linear generator cost coefficients, $c_i$. Each perturbation factor is drawn i.i.d. from a uniform distribution $\mathcal{U}[0.9, 1.1]$.

(IV) : Check if the problem has an optimal solution. If yes, then return the problem. Otherwise, repeat steps I-IV again.

### F.1.4 DATA GENERATION FOR PREDICTING OPTIMAL SOLUTIONS

We randomly generate 5000 feasible SOCP instances by methods in Appendix F.1.1 of size (50,10,10), (100,50,50), and (500,100,100) respectively. Each instance is solved in CVXPY to obtain a ground truth solution as the label. [9] Then, we divide these instances into training, validation, and test data classes by the ratio $8:1:1$.

To further validate our theorem on real-world situations, we use methods in section F.1.3 to randomly generate 1000 samples based on IEEE test systems. Then, we divide these instances into training, validation, and test data classes by the ratio $8:1:1$.

### F.1.5 DATA GENERATION FOR PREDICTING THE FEASIBILITY:

We randomly generate 5000 infeasible SOCP instances with probability $h = 0.5$ by methods in Appendix F.1.2 of size (50,10,10), (100,50,50) and (500,100,100) respectively. We use CVXPY to detect the feasibility of these instances as well. Then, we divide these instances into training class and validation class by the same ratio.

### F.2 IMPLEMENTATIONS AND TRAINING SETTINGS FOR PREDICTING THE OPTIMAL SOLUTION AND FEASIBILITY

For predicting the optimal solution, our SOCP-GNN is implemented with four message-passing layers. The learnable functions, denoted by $g_{l_1}^0$, $g_{l_2}^t$, $f_{l_3}^t$, and $f_{\text{out}}$ (where $l_1 \in \{1, \ldots, 4\}$, $l_2 \in \{1, \ldots, 6\}$, and $l_3 \in \{1, \ldots, 8\}$), are all parameterized by neural networks. Specifically, $g_{l_1}^0$ and $g_{l_2}^t$ are simple linear layers, while $f_{l_3}^t$ and $f_{\text{out}}$ are constructed with a single hidden layer containing 64 neurons. For comparison, our baseline FCNN is implemented with four hidden layers with residual connections, each containing 64 neurons. The other GNNs are implemented based on the basic message-passing method with the same embedding layer as SOCP-GNN, i.e. $h^{n,t+1} = \text{Update}(h^{n,t}, \text{Aggregate}(\{\{e_{nn'}, h^{n',t} \mid n' \in \mathcal{N}(n)\}\}))$. Here $h^{n,t}$ is the feature of node $n$ at the $t$-th message passing process, $e_{nn'}$ is the edge weight connecting node $n$ and its neighbor $n'$, and $\mathcal{N}(n)$ denotes the neighborhood of node $n$. In the vanilla MPNN, we first concatenate $e_{nn'}$ and $h^{n',t}$ to form $[e_{nn'}, h^{n',t}]$. This vector is processed by an MLP with two hidden layers (64 neurons each) to generate a message vector matching the dimension of $h^{n,t}$. We then compute the mean of these messages over the neighborhood of $n$, denoted as $\hat{h}^{n,t}$. Finally, a distinct MLP with an identical architecture maps the concatenation $[h^{n,t}, \hat{h}^{n,t}]$ to the updated feature $h^{n,t+1}$. For the Graph Isomorphism Network (GIN), we employ the same MLP architecture as in the vanilla MPNN to map $h^{n,t} + |\mathcal{N}(n)|\hat{h}^{n,t}$ to the updated feature $h^{n,t+1}$. Vanilla MPNNs and GINs both have five message passing layers. We use normalized MSE loss (Section 7) as the loss function.

For predicting the feasibility, our SOCP-GNN follows a similar structure to the one in solution prediction. Since the binary classification is simpler than the solution regression, we set the hidden layer with 16 neurons. For comparison, our baseline FCNN is implemented with three hidden layers, each containing 16 neurons. We use binary cross-entropy loss as the loss function.

All MLPs mentioned above use ReLU as the activation function. We use AdamW to optimize our learnable parameters for both FCNNs and GNNs with a maximum learning rate of $5 \times 10^{-4}$ and a batch size of 40. All experiments were conducted on an NVIDIA H200 GPU, with the exception of the inference time evaluation.

---

[9]We denote an SOCP instance by a tuple $(n, b, m)$, where $n$ represents the number of decision variables, $b$ denotes the number of polyhedral constraints, and $m$ indicates the number of second-order cone constraints.

## F.3 RESULTS FOR PREDICTING OPTIMAL SOLUTIONS AND FEASIBILITY

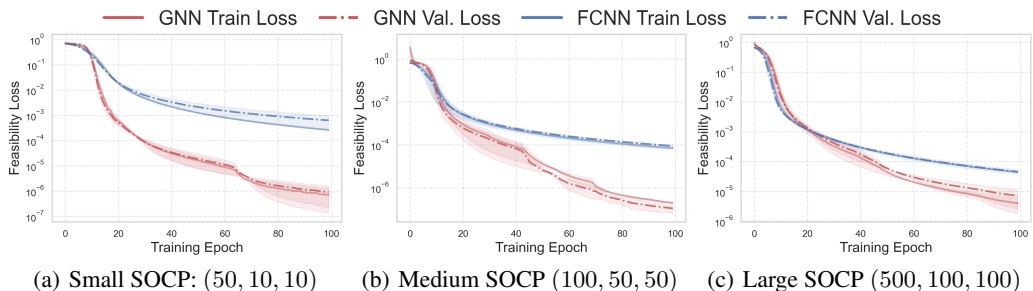

(a) Small SOCP: $(50, 10, 10)$     (b) Medium SOCP $(100, 50, 50)$     (c) Large SOCP $(500, 100, 100)$

Figure 7: Comparison between the proposed GNN and FCNN for feasibility classification for random SOCP instances. The GNN uses approximately 0.01M parameters across three scales, while the FCNN uses approximately 0.07M, 0.7M, and 7M parameters for all three problem scales, respectively.

As shown in Figures 4 and 7, the proposed SOCP-GNN surpasses the baseline FCNNs in both optimal solution prediction and feasibility classification tasks in both synthetic and real-world SOCP instances. Across all problem scales—small, medium, and large—the GNN achieves substantially lower relative MSE and binary cross-entropy loss compared to the FCNN baseline. This superior performance is particularly evident in its parameter efficiency: on large-scale problems, the GNN, with only approximately `0.35Mb` parameters, outperforms the FCNN, which requires `110Mb` parameters for the solution prediction task, representing a nearly 300-fold reduction in model complexity. We also observe similar trends in the feasibility classification tasks.

This dramatic improvement in both performance and efficiency validates the effectiveness of exploiting the inherent sparse geometric structure of optimization problems through graph representations and message passing. These results confirm the potential of our approach as a scalable, data-driven framework for solving complex optimization problems.

*Remark* 6. Since we have already proved that all target mappings are measurable, it follows that: FCNNs can provably approximate these target mappings within any given error tolerance. Hence, it's reasonable to use FCNNs as a baseline for comparison.

## F.4 EMPIRICAL STUDY ON SAMPLE/MODEL COMPLEXITY AND SIZE GENERALIZATION

As shown in Fig 5(e), we randomly generate 625, 1250, 2500, and 5000 synthetic training samples of size (50,10,10) and divide these instances into training, validation, and test data classes by the ratio $8 : 1 : 1$ respectively. We use four SOCP-GNNs with the hidden layer sizes 32, 64, 128, and 256, respectively. Then, we train these four models on the four different datasets, respectively, and then measure their training and validation losses. When the hidden layer size or the number of training samples increases, both the training loss and validation loss decrease. This demonstrates that: with a sufficient number of training samples, the SOCP-GNN can achieve near-zero approximation error and generalize effectively to unseen instances.

Moreover, we randomly generate 6000 synthetic training samples of size (10,5,5), (20,10,10), (40,20,20), (80,40,40) and (160,80,80) respectively. Then we divide these samples into training and test data classes by the ratio $5 : 1$. Then, we train the SOCP-GNN model with hidden layer size 64 on these datasets, respectively. Finally, each trained model was evaluated on all five test sets to measure its cross-size generalization performance, reported as test loss. The results are summarized in Fig 5(f). It's observed that: models trained on larger problem instances demonstrate superior generalization capabilities, particularly when tested on smaller, unseen problem sizes. Meanwhile, models trained entirely on smaller datasets also have the surprising ability to generalize to unseen larger datasets. This has validated the good size generalization probability of SOCP-GNNs, which motivates future research on efficient training of SOCP-GNNs leveraging this size generalization ability.

## F.5 EMPIRICAL STUDY ON THE LIPSCHITZ REGULARIZATION

To show how the Lipschitz assumption can be controlled in the experimental setting, we use projection-based methods to train our SOCP-GNN (Gouk et al., 2020). The method operates as follows: for

predefined constants $\lambda > 0$ and $p \geq 1$, we project each weight matrix $W$ onto an $\ell_p$-norm ball after each standard optimizer update. Specifically, at the end of every epoch, the weights are updated via the assignment:

$$W \leftarrow \frac{W}{\max\left(1, \frac{\|W\|_p}{\lambda}\right)}$$

To validate this approach, we conducted an experiment on the IEEE 118-bus dataset. The number of samples and the dividing rules are the same as above, The results are summarized in Table 1. In the table, p1_lambda0.5 refers to a configuration with $p = 1$ and $\lambda = 0.5$. The generalization gap is measured by the difference between train loss and true loss, approximating the difference between empirical risk and true risk. The Lipschitz coefficient $L$ is measured by randomly picking 10000 pairs of instances $x, y$ in dataset and then take the supremum of $\frac{\|f(x)-f(y)\|_2}{\|x-y\|_2}$. We repeat the experiment for 3 times and take the average.

Table 1: Training and test results for different configurations.

| Config | Train Loss | Test Loss | Gen Gap | Lip-L |
|---|---|---|---|---|
| baseline | 0.004885 | 0.005076 | 0.000191 | 0.5186 |
| p1_lambda0.5 | 0.008184 | 0.008208 | 0.000024 | 0.4011 |
| p1_lambda0.7 | 0.006707 | 0.006905 | 0.000199 | 0.4782 |

The experimental results indicate that selecting a smaller radius for the norm ball leads to a lower Lipschitz constant for the model. This suggests that the Lipschitz constant can be effectively controlled by constraining the norm of the weight matrices.

