# OpenReview forum: "On the Universality and Complexity of GNN for Solving Second-order Cone Programs"
_ICLR.cc/2026/Conference — ICLR 2026 Poster_

### Official Review · Reviewer_c5JN · 2025-10-27

**Soundness:** 3
**Presentation:** 3
**Contribution:** 3
**Rating:** 6
**Confidence:** 3

**Summary:**

This paper proposed a way to encode SOCP problems with graphs, which is an important extension of LP, QP, QCQP problems. The authors designed a message passing scheme on top. Besides, the separation power of WL on such graphs as well as generalization bounds are established. The empirical results also exhibited the sigfinicance of the work.

**Strengths:**

- The problem is well motivated. SOCP is an important extension of LP, QCQP, and this work is an important milestone towards more general convex cones.
- The paper is well written and easy to read.
- The design of the graph encoding for SOCP problems make a lot of sense, and faithfully encode all the information of an SOCP problem.
- The theoretical results are well established.

**Weaknesses:**

- I think the separation power of WL on SOCP is also an important point and should be mentioned in the main paper as well.
- The notations of graph nodes are a bit confusing, they are inconsistent in section 4.1 and figure 2.
- I understand there might be lack of baselines, but FCNN baseline is a bit too trivial.

**Questions:**

- What are the hardwares for comparing solving time of SOCP-GNN and MOSEK solver? It seems that the GNN has constant solving time with the growth of the instance size, so I guess it is on GPU. Did you also run MOSEK on GPU? If not that is not fare comparison.
- It is mentioned in the appendix that extending the work to more general $p$ cone only requires encoding $p$ parameter in the graph. It is plausible in the perspective of WL test, but does it hurt the generalization performance? For example, will there be such a family of problems, where different $p$ leads to the same solution?

---

> ### Author Response · Authors · 2025-11-22
> **Author's response 1**
>
> We thank Reviewer c5jN for the positive evaluation and for highlighting the importance of our work as a milestone towards more general convex cones.
>
>
>
> ---
>
> >`Weakness 1: I think the separation power of WL on SOCP is also an important point and should be mentioned in the main paper as well.`
>
> ---
>
>
>
> We agree this is an important point that deserves more prominence. The separation power of the WL test on our SOCP graphs is fundamental to the expressivity of our GNN model. We have now added a concise discussion of this in the main paper (Section 5.2), explaining how the WL test can distinguish non-isomorphic SOCP graphs that have different target properties, and we refer the reader to the appendix for the full proof.
>
>
>
> ---
>
> >`Weakness 2: The notations of graph nodes are a bit confusing; they are inconsistent in section 4.1 and figure 2.`
>
> ---
>
>
>
> Thank you for pointing out this inconsistency.
>
>
>
> - We apologize that the inconsistency is due to the different time of presenting the proofs using one notation and drawing the figure using the other notation.
>
> - To avoid confusion and consistent with Section 4.1, we have revised Figure 2 by explicitly changing the notation of nodes alongside the notation of node features.
>
>
>
> Thanks for your careful reading to help us improve clarity.
>
>
>
> ---
>
> >`Weakness 3: I understand there might be lack of baselines, but FCNN baseline is a bit too trivial.`
>
> ---
>
>
>
> We appreciate the reviewer’s concern regarding our choice of baselines. The Fully Connected Neural Network (FCNN) was initially included as a baseline to demonstrate the significant performance gained by explicitly encoding the SOCP’s structure into the model architecture via our graph representation.
>
>
>
> To address your concerns on need of more sophisticated baselines, we have incorporated two powerful message-passing architectures in the revised version: a  Message Passing Neural Network (**MPNN**) and a Graph Isomorphism Network (**GIN**).
>
>
>
> We note that no existing graph representation has been specifically designed for SOCP problems with non-linear constraints. Therefore, we compare all of those GNN-based baselines to the same graph structure proposed in this work. Notably, vanilla MPNN and GIN perform message passing based on the adjacency relationships, whereas our approach incorporates well-designed constraint-to-variable and variable-to-constraint message passing (Sec. 4.2).
>
>
>
> - The updated results, presented in section 7, reveal that all GNN-based models perform favorably against the FCNN, further validating the benefits of our problem-specific graph structure.
>
> - Moreover, our SOCP-GNN consistently outperforms the other two GNN baselines, highlighting the efficiency of our proposed three-sublayer message-passing mechanism. The experimental details can be found at Appendix F with highlights.

---

> ### Author Response · Authors · 2025-11-22
> **Author's response 2**
>
> ---
>
> >`Question 1:  What are the hardwares for comparing solving time of SOCP-GNN and MOSEK solver? It seems that the GNN has constant solving time with the growth of the instance size, so I guess it is on GPU. Did you also run MOSEK on GPU? If not that is not fare comparison.`
>
> ---
>
>
>
> This is a crucial question for ensuring a fair comparison.
>
>
>
> - Our SOCP-GNN model was executed on a GPU (NVIDIA H200), while the MOSEK solver was run on a CPU (AMD EPYC 7763 64-Core Processor). We acknowledge that this represents different hardware platforms.
>
> - To the best of our knowledge, MOSEK does not currently support GPU acceleration—it is specifically optimized for multi-core CPU execution using highly efficient sparse linear algebra routines. Meanwhile, GNN-based methods naturally leverage GPU architectures for parallel matrix operations. Therefore, each method was run on its suitable hardware platform.
>
> - **Additional direct comparison:** To address the reviewer’s concern and provide a more direct comparison, we have added results for our SOCP-GNN running on CPU in Section 7. These results show that even when running on the same CPU hardware as MOSEK, our approach delivers a 10-fold speedup on large-scale synthetic datasets and a remarkable 100-fold speedup on large-scale real-world data, with comparable accuracy. Furthermore, this speed advantage is significantly amplified when SOCP-GNNs are deployed on GPUs.
>
>
>
> ---
>
> >`Question 2: It is mentioned in the appendix that extending the work to more general p cone only requires encoding p parameter in the graph. It is plausible in the perspective of WL test, but does it hurt the generalization performance? For example, will there be such a family of problems, where different p leads to the same solution?`
>
> ---
>
>
>
> This is a very insightful question about the limits of generalization.
>
>
>
> We acknowledge your insights that, while encoding $p$ as a node feature allows the GNN to distinguish between different cone types from an expressivity (WL test) standpoint, this does not automatically guarantee good generalization. More precisely, encoding $p$ would introduce extra dimensions to the original problem. As a result, this would cause higher sample complexity to the generalization of SOCP-GNNs due to a large problem dimension (i.e., larger $N$), according to Theorem 2.
>
>
>
> There indeed exists such a class of problems where different $p$ leads to the same solution. Consider the following problems:
>
> $$
> \min_{x \in \mathbb{R}^n} \\|x - a\\|_p \quad \text{s.t. } \\|x\\|_p \leq c^T x
> $$
>
> where $a_i \equiv a' \geq 0, \forall i \in [n]$. If $c_i \geq 1, \forall i \in [n]$, we can see that: $a$ is an optimal solution since $\\|a\\|_p = n^{\frac{1}{p}} a' \leq n a' \leq c^T a$ holds for all $p \geq 1$ and $\\|x - a\\|_p \geq 0$. By adding the epigraph constraint, we can see that: these problems are p-order cone programs.
>
>
> When the distribution consists of problems whose solutions are insensitive/identical to $p$ (as in the above example), this does not necessarily harm generalization. If the model is trained on a dataset where optimal solutions are largely invariant to changes in $p$, it can learn to ignore this feature through supervised training. For instance, the MLPs in the initial embedding layer could learn to assign zero weights to connections involving $p$, making the GNN's output insensitive to $p$.

---

> > ### Comment · Reviewer_c5JN · 2025-11-25
> >
> > Thank you for the detailed feedback! I believe this is a theoretical and empirically strong paper and will make a good publication. The GNN and ML4CO community will benefit a lot from it. I would like to slightly raise my score.

---

### Official Review · Reviewer_tMNV · 2025-10-29

**Soundness:** 4
**Presentation:** 4
**Contribution:** 2
**Rating:** 4
**Confidence:** 4

**Summary:**

The author proposed a novel graph representation for SOCPs, that is a major class of convex optimization problems. Based on this representation, the author proposed a GNN structure called SOCP-GNN, that is simple enough to be effective. Additionally, the author analyzed the representation power and generalization capability of SOCP-GNN. Finally, the author provided the results a series of experiments to validate the proposed framework.

**Strengths:**

- The new graph representation extends to SOCP problems, which is more general than previously considered LP, QP, and QCQPs. And on the representation, the proposed structure is simple enough, while having enough representation power to express the inherent structure of SOCP instances.
- The author provides a framework for analyzing the generalization capability of SOCP-GNN or other structures with representation power guarantee established by WL-test and relevant tool(i.e. Lusin's theorem and Generalized Stone-Weierstrass Theorem).

**Weaknesses:**

- The previous representation on QCQPs **benefits from sparsity** to get better MP complexity and the SOCP-GNN(when applied on equivalent reformulated QCQPs) benefits from **the constraints being low-rank**. The author claimed that the proposed structure is less complex when used on equivalent reformulated convex QCQPs, but there is no argument about relation between sparsity of the original QCQP instance and the complexity of the reformulated instance.
- SOCPs do not cover **general** QCQPs, since only convex QCQPs can be directly reformulated to equivalent SOCPs, and SOCPs reformulated to special non-convex QCQPs (since the special combination of non-convex quadratic constraints and linear constraint makes it **implicitly convex** to be solved in polynomial time). So there is not too much improvement with respect to general non-convex QCQPs.

**Questions:**

- In the comparison of node complexity and MP complexity, the ranks of the cone constraints are denoted by $k_{i} $instead of $r_{i}$. $r_{i}$ is used to denote the rank of quadratic constraints, but here we are not discussing quadratic constraints.

---

> ### Author Response · Authors · 2025-11-22
> **Author's response 1**
>
> Thank you for your valuable feedback and for recognizing the strength of our simple but expressive SOCP-GNN and our analysis of its representation power and generalization capability. Below, we would like to clear up your confusion in this paper.
>
>
>
> ---
>
> >`Weakness 1: The previous representation on QCQPs benefits from sparsity to get better MP complexity and the SOCP-GNN (when applied on equivalent reformulated QCQPs) benefits from the constraints being low-rank. The author claimed that the proposed structure is less complex when used on equivalent reformulated convex QCQPs, but there is no argument about relation between sparsity of the original QCQP instance and the complexity of the reformulated instance.`
>
> ---
>
>
>
> Thank you for your detailed comments. After reviewing them carefully, we understand your comments as "(i) the previous GNN can benefit from the sparsity of QCQP, while our SOCP-GNN operates on the reformulated SOCP, which may lose the sparsity. (ii) a fair comparison between the previous GNN and our SOCP-GNN under different matrix coefficient structures."
>
>
>
> We first clarify that our claim "*the proposed structure achieves the same order of node and message passing complexity as state-of-the-art GNNs designed specifically for QCQP*" (lines 252-254) is concluded for general problem coefficients without specific structures, thus it is the worst-case comparison. If the quadratic matrix is low-rank, our worst-case complexity is lower than previous QCQP-GNN.
>
> However, for specific QCQP with a sparse quadratic coefficient matrix, the complexity may differ.
>
>
>
> - **Sparsity is not necessarily preserved when reformulating a QCQP to an SOCP:** We acknowledge your insight that the sparsity in QCQP (e.g., sparse quadratic matrix $Q$) may not be preserved in the reformulated SOCP, since matrix decomposition in the reformulation $Q = LL^T$ does not necessarily lead to the sparsity of $L$ in our SOCP reformulation.
>
> - **Sparsity can be also preserved for QCQP with certain sparse quadratic matrix $Q$:** For example, for symmetric positive definite banded matrices $Q$, we can always find a factor $L$ that has the same level of sparsity as $Q$ by Cholesky decomposition (e.g., Chapter 4.3.5 of [1]). Similar situations can also be found when $Q$ is block-diagonal. Furthermore, we note that preserving sparsity in matrix decomposition has been studied in matrix analysis literature, and there are numerous modern techniques to obtain a sparse factor $L$ (e.g., Chapter 4 of [2]).
>
>
>
> Thus, unless the concrete coefficient distribution of QCQP is specified, we cannot conclude the relationship between the sparsity of the quadratic matrix $Q$ and the sparsity of its factor $L$. Therefore, we focus on worst-case comparison and claim the efficiency of our methods in the manuscript.
>
>
>
> Thanks again for your insights. To avoid confusion, we have added more discussions in lines 255-262 to specify the worst-case complexity comparison and also discuss practical performance when the original QCQP problem is sparse.
>
>
>
> - [1] Golub, G. H., & Van Loan, C. F. (2013). Matrix computations. JHU press.
>
> - [2] Davis, T. A. (2006). Direct methods for sparse linear systems. Society for Industrial and Applied Mathematics.

---

> ### Author Response · Authors · 2025-11-22
> **Author's response 2**
>
> ---
>
> >`Weakness 2: SOCPs do not cover general QCQPs, since only convex QCQPs can be directly reformulated to equivalent SOCPs, and SOCPs reformulated to special non-convex QCQPs (since the special combination of non-convex quadratic constraints and linear constraints makes it implicitly convex to be solved in polynomial time). So there is not too much improvement with respect to general non-convex QCQPs.`
>
> ---
>
>
>
> We thank the reviewer for this comment and clarify the scope of our work.
>
>
>
> Our work focuses on GNNs for **convex optimization problems**. Within the domain of convex optimization, our contributions represent substantial advances over prior work:
>
>
>
> - Prior works [3,4] handle only convex QCQPs with positive semi-definite (PSD) quadratic matrices, as the PSD condition is necessary for their universality proofs. In contrast, our framework addresses the broader class of SOCPs, which covers all (implicitly) convex QCQPs—including those with non-PSD quadratic matrices in individual constraints. Despite the joint constraint set being convex (as the reviewer notes), previous GNN architectures theoretically fail on such instances. Thus, our work thus expands GNN capabilities to a wider range of convex optimization problems.
>
> - Moreover, we also extend our GNN design and universality analysis to p-order cone programs ([5,6]), which are convex optimization problems but are not covered by non-convex QCQP formulations.
>
> - Beyond expressivity results, we provide the first framework to analyze the generalization ability of WL-test-based GNNs for optimization, filling an important gap in the GNN for convex optimization literature.
>
>
>
> In summary, while we do not address general non-convex QCQPs, we significantly advance the state-of-the-art for GNNs on convex optimization.
>
>
>
>
> - [3] Chen, Z., Chen, X., Liu, J., Wang, X., & Yin, W. (2024). Expressive power of graph neural networks for (mixed-integer) quadratic programs. arXiv preprint arXiv:2406.05938.
>
> - [4] Wu, C., Chen, Q., Wang, A., Ding, T., Sun, R., Yang, W., & Shi, Q. (2024). On representing convex quadratically constrained quadratic programs via graph neural networks. arXiv preprint arXiv:2411.13805.
>
> - [5] Burer, S., & Chen, J. (2009). A p-cone sequential relaxation procedure for 0-1 integer programs. Optimization methods & software, 24(4-5), 523-548.
>
>
> - [6] Blanco, V., Magron, V., & Martínez-Antón, M. (2025). On the complexity of p-order cone programs. arXiv. https://doi.org/10.48550/arXiv.2501.09828
>
>
> ---
>
> >`Question 1: In the comparison of node complexity and MP complexity, the ranks of the cone constraints are denoted by $k_i$. Instead of $r_i$, $r_i$ is used to denote the rank of quadratic constraints, but here we are not discussing quadratic constraints.`
>
> ---
>
>
>
> Thanks for your question regarding the rank notifications. We make the following clarifications:
>
>
>
> - $k_i$ denotes the rows of matrix $A_i$ in the SOC constraint $\\|A_ix + b_i\\|_2 \leq c_i^T x + d_i$ in the general SOCP formulation at Sec. 3.
>
> - $r_i$ denotes the rank of the matrix $Q_i$ of convex QCQP, which shows in the objective function and constraint functions of QCQP at Sec 4.2.
>
>
> In the comparison of node complexity and MP complexity in Figure 3, we focus on convex QCQPs and their equivalent SOCP reformulations. In the SOCP reformulation, we have $k_i = r_i + 1$. Thus, we use $r_i$ in the node/MP complexity of SOCP-GNNs for a direct comparison with previous QCQP-GNNs.
>
>
> Thanks again for your help with the clarification.

---

### Official Review · Reviewer_wvA8 · 2025-10-29

**Soundness:** 3
**Presentation:** 3
**Contribution:** 4
**Rating:** 8
**Confidence:** 4

**Summary:**

The present manuscript addresses the solution of Second-Order Cone Programs (SOCPs) by means of Graph Neural Networks, specifically designed to process the variables-constraints graph arising from such problems, with a lightweight implementation that is competitive in terms of computational complexity.
The authors provide universal approximation results and generalization bounds for the proposed model, and evaluate their model against  traditional SOCP solvers.

**Strengths:**

The literature review is accurate and broad, and the related work is addressed (almost) properly.
The theoretical results are meaningful and derived with deep rigorousness (the Appendix has been checked).
All the needed mathematical concepts are clearly introduced and explained, and intuitive readings of the theoretical results are provided.
The experimental evaluation on synthetic data is well set and well conducted.

**Weaknesses:**

Major concerns:
- The literature review on VC dimension for GNNs is not complete, please check [1];
- at line 306, a Lipschitz assumption on GNN is stated; although, as noted by the authors, this is frequent in theoretical analysis of generalization capabilities of GNNs, I don't think that such property has been properly addressed when switching to the experimental framework;
- a similar concern arises about connecting the generalization bound with the experiments, i.e.  there are not connections between Theorem 2 and the experiments in Section 7;
- lastly, even if this paper is a proof of concept, the authors propose a benchmarking over AC-OPF solvers; in this case, I think that a comparison with actual deep learning OPF solvers is needed.

Minor concerns:
- line 104, "specially" -> "specifically"
- line 128, "representations" -> "representation"
- line 1845-1847: "Lipshitz" -> "Lipschitz"

[1] D’Inverno, G. A., Bianchini, M., & Scarselli, F. (2025). VC dimension of Graph Neural Networks with Pfaffian activation functions. Neural Networks, 182, 106924.

**Questions:**

- I would suggest the authors to revise the literature over generalization results for GNNs, as suggested above;
- I would suggest the authors make explicit connections between the experimental validation and the theoretical results;
- Lastly, I would strongly encourage the authors to insert a (possibly) fair comparison with deep learning-based OPF solvers.

---

> ### Author Response · Authors · 2025-11-22
> **Author's Response 1**
>
> We sincerely thank the time and effort you devoted to thoroughly reviewing our paper. Your detailed and constructive comments are greatly appreciated, and we will address your insightful concerns below.
>
>
>
> ---
>
> >`Concern 1: The Literature review for GNNs is not complete, please check [1]. [1] D’Inverno, G. A., Bianchini, M., & Scarselli, F. (2025). VC dimension of Graph Neural Networks with Pfaffian activation functions. Neural Networks, 182, 106924.`
>
> ---
>
>
>
> Thank you for bringing this recent and relevant work to our attention. This work derives upper bounds for the VC dimension of Graph Neural Networks using more general Pfaffian activation functions (like sigmoid and tanh), relating generalization capacity to network hyperparameters and the number of colors determined by the Weisfeiler-Lehman test. We have now incorporated and discussed the findings from D’Inverno et al. (2025) in our literature review on GNN generalization (line 100 and 919-922).
>
>
>
> ---
>
> >`Concern 2: at line 306, a Lipschitz assumption on GNN is stated; although, as noted by the authors, this is frequent in theoretical analysis of generalization capabilities of GNNs, I don’t think that such property has been properly addressed when switching to the experimental framework;`
>
> ---
>
>
>
> We thank the reviewer for highlighting this important connection between theory and practice. To address this, following the literature on Lipschitz neural networks [1], we implemented a projection-based Lipschitz regularization method for our SOCP-GNNs.
>
>
>
> The method operates as follows: for predefined constants $\lambda > 0$ and $p \geq 1$, we project each weight matrix $W$ onto an $\ell_p$-norm ball with radius $\lambda$ after each standard optimizer update. Specifically, at the end of every epoch, the weights are updated via the assignment:
>
> $$
> W \leftarrow \frac{W}{\max\left(1, \frac{\\|W\\|_p}{\lambda}\right)}
> $$
>
>
>
> To validate this approach, we conducted an experiment on the IEEE 118-bus dataset. The results are summarized in the following table. The detailed implementation is available at Appendix F.5 in the paper. In the table, `p1 lambda0.5` refers to a configuration with $p = 1$ and $\lambda = 0.5$. A smaller $\lambda$ represents stronger Lipschitz regularization.
>
> | Config       | Train Loss | Test Loss | Gen Gap  | Lip-L  |
> | ------------ | ---------- | --------- | -------- | ------ |
> | baseline     | 0.004885   | 0.005076  | 0.000191 | 0.5186 |
> | p1 lambda0.7 | 0.006707   | 0.006905  | 0.000199 | 0.4782 |
> | p1 lambda0.5 | 0.008184   | 0.008208  | 0.000024 | 0.4011 |
>
> The experimental results indicate that selecting a smaller radius ($\lambda$) for the norm ball leads to a lower Lipschitz constant for the model. This suggests that the Lipschitz constant can be effectively controlled by constraining the norm of the weight matrices.
>
>
> We include this experiment in Appendix F.5. and also discuss it in detail in Sec 7.3.
>
>
> - [1] Gouk, H., Frank, E., Pfahringer, B., & Cree, M. J. (2021). Regularisation of neural networks by enforcing lipschitz continuity. Machine Learning, 110(2), 393-416.

---

> ### Author Response · Authors · 2025-11-22
> **Author's response 2**
>
> ---
>
> >`Concern 3: a similar concern arises about connecting the generalization bound with the experiments, i.e. there are not connections between Theorem 2 and the experiments in Section 7;`
>
> ---
>
>
>
> Thank you for this insightful comment.
>
>
>
> - Theorem 2 establishes the sample complexity/generalization ability of the defined WL-test-based GNN, showing the generalization error is decreased with increasing of training samples, and dependence on the network complexity (i.e., Lipschitz).
>
> - In our original experiment, we show the training and validation loss of GNN with different parameter and trained with different samples, showing the trend of decreasing validation loss with increasing training samples.
>
> - Moreover, in the additional experiment on Lipschitz Regularization (Appendix F.5.), we note that if we restrict the matrix norm of the weight matrix to a ball in $\ell_p$ norm with radius $r$ in the training process, the generalization gap decreases as $r$ decreases (i.e. we decrease the Lipschitz constant $L$ of SOCP-GNN) as the train error increases. This enhances the tradeoff between the expressive power and generalization ability of SOCP-GNNs, as indicated in Theorem 2.
>
>
>
> ---
>
> >`Concern 4: lastly, even if this paper is a proof of concept, the authors propose a benchmarking over AC-OPF solvers; in this case, I think that a comparison with actual deep learning OPF solvers is needed.`
>
> ---
>
>
>
> Thanks for your interest in our experiment on OPF problems. We highly acknowledge your insight into the potential of our approach and analysis directed at non-convex AC-OPF problems. Below are our responses to your concern.
>
>
>
> - First, we would like to clarify that we solve the second-order cone relaxed AC-OPF problem; such relaxation provides a valid lower bound for the real AC-OPF solution due to potential constraint violations, making a direct comparison to existing deep AC-OPF solvers unfair.
>
> - Second, existing AC-OPF solvers use either FCNN or GNN to predict the optimal solution (with different loss designs). The GNN for AC-OPF operates on the power grid topology with physical node and edge parameters. While our SOCP-GNN operates on the graph from the variable-constraint relationship of the optimization problem itself. It makes previous GNN designs not directly applicable to our SOCP formulations. We have added a footnote to discuss this on page 8 of our revised manuscript.
>
> - Third, for a fair comparison with existing GNN solvers, we have conducted additional experiments, including Message Passing Neural Network (MPNN) and Graph Isomorphism Network (GIN) over the same SOCP graph structure (Sec. 7) to show the effectiveness of our GNN design. The results show that Our SOCP-GNN leverages the graph structure and three-sublayer message passing architecture to surpass other GNNs, such as GIN and standard GNNs.
>
>
>
> Furthermore, we have included additional references regarding the application of GNNs for Optimal Power Flow. (line 427-431 and 824)
>
>
>
> ---
>
> >`Minor concerns: (i) line 104, "specially" -> "specifically"; (ii) line 128, "representations" -> "representation"; (iii) line 1845-1847: "Lipshitz" -> "Lipschitz"`
>
> ---
>
>
>
> Thank you for catching these typos. We have corrected all of them in the revised manuscript with highlight. Thank you again for your time spending on reading papers carefully and your thoughtful advice that helps our paper become better.

---

> > ### Comment · Reviewer_wvA8 · 2025-11-26
> > **Response to the authors**
> >
> > I would like to thank the authors for carefully revising and enhancing the manuscript.
> > They have addressed all my concerns in a thorough and exhaustive manner, therefore I confirm my score and suggest acceptance.

---

### Official Review · Reviewer_5AzP · 2025-11-01

**Soundness:** 3
**Presentation:** 3
**Contribution:** 3
**Rating:** 6
**Confidence:** 3

**Summary:**

This ICLR submission proposes a graph representation and Graph Neural Network (GNN) architecture, termed SOCP-GNN, for solving Second-Order Cone Programs (SOCPs). The graph representation has variable, polyhedral, minor conic, and major conic nodes, enabling message-passing GNNs to predict key properties like feasibility and optimal solutions. They prove universal approximation capabilities using Weisfeiler-Lehman-based expressivity and derive sample complexity bounds via Rademacher complexity for generalization. The approach generalizes to p-order cone programs and demonstrates empirical superiority on synthetic SOCPs and real-world power grid problems, using fewer parameters than fully connected networks.

**Strengths:**

1. The paper is in general well written, and the main results are easy to follow.

2. The paper provides the generalization analysis beyond WL-test-based GNN analysis, deriving sample complexity bounds via Rademacher complexity.

**Weaknesses:**

The WL-test-based universal approximation analysis is not completely new; it uses ideas from (Chen et al., 2022b), as the authors explicitly noted.

**Questions:**

Is it possible to add quadratic terms in the objective function, as in (Wu et al., 2024; Chen et al., 2024b)?

---

> ### Author Response · Authors · 2025-11-22
>
> We thank the reviewer for the thoughtful feedback and for recognizing the clarity of our writing and the value of our generalization analysis. We address your concerns below, point by point.
>
>
>
> ---
>
> >`Weakness 1: The WL-test-based universal approximation analysis is not completely new; it uses ideas from (Chen et al., 2022b), as the authors explicitly noted.`
>
> ---
>
>
>
> We acknowledge that our theoretical framework builds upon the foundation laid by [1], which establishes the Weisfeiler-Lehman (WL) test-based universal approximation analysis for GNNs. However, our work makes several non-trivial theoretical contributions that significantly extend beyond this prior work:
>
>
>
> - We extend GNN universality from LP [1] and convex QCQP [2,3] to second-order cone programming (SOCP), which subsumes both as special cases.
>
> - Such an extension is non-trivial, which needs both a new graph design for encoding non-linear SOC constraints (Section 4) and a new technical proof leveraging the structure property of the SOC constraint (e.g., Lemma 3 in Appendix C). We have discussed those difficulties in section 1 and 3 in the manuscript.
>
> - Moreover, we extend our universality analysis to more complex p-order cone constraints, further broadening the capability of GNN for non-linear optimization problems [4,5].
>
>
>
> These contributions—the SOCP extension, novel graph encodings, new technical lemmas, and p-order cone generalization—represent substantial advances beyond the methodological foundation of [1].
>
>
>
> - [1] Chen, Z., Liu, J., Wang, X., Lu, J., & Yin, W. (2022). On representing linear programs by graph neural networks. arXiv preprint arXiv:2209.12288.
>
> - [2] Chen, Z., Chen, X., Liu, J., Wang, X., & Yin, W. (2024). Expressive power of graph neural networks for (mixed-integer) quadratic programs. arXiv preprint arXiv:2406.05938.
>
> - [3] Wu, C., Chen, Q., Wang, A., Ding, T., Sun, R., Yang, W., & Shi, Q. (2024). On representing convex quadratically constrained quadratic programs via graph neural networks. arXiv preprint arXiv:2411.13805.
>
> - [4] Burer, S., & Chen, J. (2009). A p-cone sequential relaxation procedure for 0-1 integer programs. Optimization methods & software, 24(4-5), 523-548.
>
> - [5] Blanco, V., Magron, V., & Martínez-Antón, M. (2025). On the complexity of p-order cone programs. arXiv. https://doi.org/10.48550/arXiv.2501.09828
>
>
> ---
>
> >`Question 1: Is it possible to add quadratic terms in the objective function, as in (Wu et al., 2024; Chen et al., 2024b)?`
>
> ---
>
>
>
> We clarify that our SOCP-GNN can indeed be applied to optimization problems with a convex quadratic objective function (as discussed in Remark 3 in the original manuscript). The convex quadratic objective can be equivalently converted to a linear objective by adding an additional convex quadratic constraint, and then we can reformulate the convex quadratic constraint into a second-order cone constraint and apply our GNN accordingly.
>
>
>
> - In detail, we consider an SOCP with a convex quadratic objective:
> $$
> \min_{x}  x^T Q x + q^T x \quad \text{s.t. } x \in \cap_{i=1}^m C_i
> $$
>
> where $Q \in S^n_+$ is positive semi-definite and $C_i$ is the feasible region determined by the $i$-th SOC constraint.
>
>
>
> - This problem is equivalent to the following problem via epigraph constraint:
> $$
> \min_{x,\gamma} \gamma \quad \text{s.t. } x \in \cap_{i=1}^m C_i, \quad  x^T Q x + q^T x \leq \gamma
> $$
>
>
>
> - Since $Q \in S^n_+$ is positive semi-definite, we can decompose $Q$ as $Q = LL^T$. The problem can be further reformulated SOCP (as discussed in Remark 3):
>
> $$
> \min_{x,\gamma} \gamma \quad \text{s.t. } x \in \cap_{i=1}^m C_i, \quad \left\\| \begin{pmatrix} \frac{1-\gamma+q^T x}{2} \\\\ L^T x \end{pmatrix} \right\\|_2 \leq \frac{1 + \gamma - q^T x}{2}
> $$
>
> The equivalently reformulated problem is also an SOCP with a linear objective and one additional SOC constraint, which can be modeled and solved by SOCP-GNNs, according to our theorem. (We also add the footnote and remark 3 on page 4 to clarify this.)

---

### Author Response · Authors · 2025-12-04
**Author's brief summary**

Dear Reviewers, Area Chairs, and Program Chairs,

We sincerely thank all the reviewers for their insightful and constructive comments, and thank the area chairs and program chairs for their extra efforts spent on our paper.  We are deeply saddened by the unexpected accident that affected our community, and hope that everyone's efforts will pay off.

---

We first thank the reviewers for their acknowledgment of our novelty and contributions:

1. **Architecture Contributions:**
    * Reviewer c5JN: *"The design of the graph encoding for SOCP problems make a lot of sense, and faithfully encode all the information of an SOCP problem."*
    * Reviewer tMNV: *"... And on the representation, the proposed structure is simple enough, while having enough representation power to express the inherent structure of SOCP instances."*
2. **Theoretical Contributions:**
    * Reviewer tMNV: *"The author provides a framework for analyzing the generalization capability of SOCP-GNN or other structures with representation power guarantee established by WL-test and relevant tool(i.e. Lusin's theorem and Generalized Stone-Weierstrass Theorem)."*
    * Reviewer 5AzP: *"The paper provides the generalization analysis beyond WL-test-based GNN analysis, deriving sample complexity bounds via Rademacher complexity."*
    * Reviewer wvA8: *"The theoretical results are meaningful and derived with deep rigorousness (the Appendix has been checked)."*
    * Reviewer c5JN: *"The theoretical results are well established."*
3. **Experimental Strengths:**
    * Reviewer wvA8: *"The experimental evaluation on synthetic data is well set and well conducted."*
    * Reviewer c5JN: *"I believe this is a theoretical and empirically strong paper and will make a good publication. The GNN and ML4CO community will benefit a lot from it."* (Reply on Nov.25)
4. **Well Motivation:**
    * Reviewer c5JN: *"The problem is well motivated. SOCP is an important extension of LP, QCQP, and this work is an important milestone towards more general convex cones."*
    * Reviewer tMNV: *"The author proposed a novel graph representation for SOCPs, that is a major class of convex optimization problems."*

---

We have made the following improvements that directly address the major concerns of reviewers:

- We have incorporated two additional GNN baselines (vanilla MPNNs and GINs) to further demonstrate the effectiveness of our graph representations and proposed message passing mechanisms (Reviewer c5JN and wvA8).
- We have conducted a new projection-based Lipschitz regularization experiment to empirically validate the Lipschitz assumption, while bridging the gap between our theoretical generalization results and experimental observations. (Reviewer wvA8).
- We clarified the hardware settings and added new CPU-based GNN experiments, demonstrating that our method maintains a significant speed advantage (up to 100x) even on CPUs. (Reviewer c5JN).
- We extended the original discussion regarding worst-case complexity when restricted to QCQPs, and further discussed the practical performance in sparse settings. (Reviewer tMNV)
- We show that SOCP-GNNs can handle the situation when the objective function is quadratic. (Reviewer 5AzP)

---

We also clarified the contributions and made minor revisions to address other comments of reviewers:

1. Correct typos and include missing citations that stem from and extend the literature previously discussed (Reviewer wvA8).
2. Clarify our contributions and novelties. (Reviewer 5AzP, tMNV)
3. Add a concise discussion about the separation power of the modified WL-test. (Reviewer c5JN)
4. Include a comprehensive discussion regarding the inherent difficulties in benchmarking deep learning-based OPF solver. (Reviewer wvA8)
5. Correct the inconsistency between the figure and the text. (Reviewer c5JN)

---

Two reviewers (wvA8 and c5JN) responded to our rebuttals prior to the incident. Both confirmed that their concerns were thoroughly addressed, acknowledging the paper's theoretical and empirical strengths and its significant value to the GNN and L2O communities. We believe that our comprehensive responses will address the concerns of the other two reviewers as well.

Once again, we sincerely thank the reviewers, AC, and SAC for their thoughtful feedback and engagement, which have substantially improved the clarity and scope of our work.

Best regards,

The Authors

---

### Meta-Review · Area_Chair_X3Q2 · 2026-01-06

**Summary:**

The submission is technically sound with a meaningfully new framework, especially the SOCP-specific graph encoding and the accompanying expressivity and generalization analyses. Some disagreement in novelty, with some viewing the universality argument as largely an adaptation of prior WL-based templates, while others consider the SOCP extension and supporting lemmas a substantial step. Practical concerns about baseline strength and runtime fairness were prominent initially but were materially reduced in the rebuttal through stronger GNN baselines and clarified CPU vs GPU comparisons. Remaining debate is mostly about positioning and scope of comparisons rather than correctness.

**Reviewer Concerns:**

Novelty. Incremental vs substantive extension beyond prior WL-based universality work? What is new vs integration.

Scope. Convex focus only; limits relative to general nonconvex QCQP.

Practical applicability. Lipschitz assumption and how it relates to the implemented model; weak linkage between generalization bounds and experiments.

Experiments. Initially trivial FCNN baseline; hardware fairness for runtime; OPF comparison to deep learning OPF solvers or justification for non-comparability.

Complexity. Reformulation may lose sparsity, so comparisons can be unclear in structured regimes.

**Reviewer Scores:**

Slight increase as indicated by the reviewers. Already past threshold for acceptance.

---

### Decision · Program_Chairs · 2026-01-26

Accept (Poster)